# Robust odor identification in novel olfactory environments in mice

Yan Li[1], Mitchell Swerdloff[1], Tianyu She[1], Asiyah Rahman[1], Naveen Sharma[1], Reema Shah[1], Michael Castellano[1], Daniel Mogel[1], Jason Wu[1], Asim Ahmed[1], James San Miguel[1], Jared Cohn[1], Nikesh Shah[1], Raddy L. Ramos[1] & Gonzalo H. Otazu[1] ✉

Relevant odors signaling food, mates, or predators can be masked by unpredictable mixtures of less relevant background odors. Here, we developed a mouse behavioral paradigm to test the role played by the novelty of the background odors. During the task, mice identified target odors in previously learned background odors and were challenged by catch trials with novel background odors, a task similar to visual CAPTCHA. Female wild-type (WT) mice could accurately identify known targets in novel background odors. WT mice performance was higher than linear classifiers and the nearest neighbor classifier trained using olfactory bulb glomerular activation patterns. Performance was more consistent with an odor deconvolution method. We also used our task to investigate the performance of female *Cntnap2*[-/-] mice, which show some autism-like behaviors. *Cntnap2*[-/-] mice had glomerular activation patterns similar to WT mice and matched WT mice target detection for known background odors. However, *Cntnap2*[-/-] mice performance fell almost to chance levels in the presence of novel backgrounds. Our findings suggest that mice use a robust algorithm for detecting odors in novel environments and this computation is impaired in *Cntnap2*[-/-] mice.

Wild-type (WT) mice in their natural environment need to identify target odors in the presence of novel background odors. Odorants activate olfactory receptors expressed in the olfactory epithelium. These receptor neurons project to specific glomeruli in the olfactory bulb. Glomerular activation is processed by the olfactory bulb circuitry and is conveyed by mitral and tufted cells to different brain areas[1–3]. Neural representations of odor mixtures with different weak targets but the same strong background odor could be dominated by the background odor[4], resulting in a similar neural representation. The olfactory bulb circuitry can separate similar patterns[5]; however, mixtures that include novel background odors do not have previously learned associations, and mice need to generalize to produce an appropriate action—a function that is affected in Autism Spectrum Disorders (ASD). Novel odors produce larger mitral cell activation

compared to familiar odors[6] suggesting that novel background odors might be particularly effective at masking target odors of interest.

WT mice can be trained over hundreds of trials to detect target odors in the presence of familiar background odors[7]. However, we do not know if WT mice can generalize and successfully recognize a known target odor on the first presentation with a novel background odor, nor what algorithm WT mice might employ.

Here we have developed a novel behavioral paradigm to study odor identification in novel backgrounds in the *Cntnap2*[−/−] mouse model of autism. In the first part, we characterize the behavioral strategy used by WT mice to identify target odors in novel background odors. In the second part, we compare behavioral performance with previously proposed algorithms of odor identification using intrinsic imaging of the olfactory bulb. In the third part, we analyze the behavior

[1]Department of Biomedical Sciences, New York Institute of Technology, College of Osteopathic Medicine, Northern Boulevard, PO Box 8000 Old Westbury, NY 11568, USA. ✉e-mail: gotazual@nyit.edu

of the *Cntnap2*[−/−] mouse model of autism in odor recognition in the presence of novel backgrounds.

## Results

### Odor identification in novel backgrounds (olfactory CAPTCHA) in WT mice

In order to test responses to novel background odors, we used the olfactory equivalent of the visual CAPTCHA[8] employed for human verification tasks, which also serves as a benchmark for testing recognition algorithms[9]. CAPTCHA requires a user to identify letters (known targets) superimposed on distracting stimuli (novel backgrounds). Mice exposed to a novel background as part of an olfactory CAPTCHA task have to identify the target odors present permitting direct evaluation of olfactory function. In contrast, it is not straightforward to evaluate the olfactory function of animals freely exploring novel background odors[10,11] because there is no task assigned. We hypothesized that WT mice will be able to identify known odors in the presence of novel background odors since this situation mimics what they experience in the wild and that a mouse model of autism will not because CAPTCHA requires generalization and robust responses in the presence of novelty, which are affected in ASD.

Mixtures of target and background odors were divided into a training set and a test set (see Fig. 1A). Water-deprived mice were trained with the training set and their performance was evaluated with the test set. The training set consisted of 16 mixtures of 3 odors, 1 target odor that solely determined reward availability (4 possible odors), 1 contextual background odor (4 possible odors), and 1 fixed background, (s)-(−)-limonene. Both (s)-(−)-limonene and the contextual background odors were presented at a higher concentration (0.1% of saturated vapor) than the target odors (0.025%). The contextual background odors and the (s)-(−)-limonene were the known background odors.

The test set also consisted of mixtures of three odors that included the same target odors and contextual background odors, but the (s)-(−)-limonene was replaced by one of 11 possible novel background odors, which were also presented at a higher concentration (0.1%). We used a go/no-go design (see Fig. 1B), with two of the target odors indicating the presence of a water reward upon licking the water tube with odors delivered using an automated serial air-dilution odor machine[12]. The differences in performance between the training set and the test set can be solely ascribed to the novel background odor because both sets include the same mixtures of target and contextual background odors.

We chose a relatively complex training set (eight go mixtures and eight no-go mixtures) to promote generalization. We chose a go/no-go licking task because it is relatively easy for WT mice to learn[13] and it can also be learned by the *Cntnap2*[−/−] mouse model of autism[14]. We did not use odors that are known to trigger innate responses[15–17] because we wanted to study a general mechanism for target detection that did not rely on specialized selective odor receptors.

### Detection of weakly activated glomeruli using intrinsic imaging

We built a triple serial air-dilution odor machine (see Supplementary Fig. 1 for an odor machine schematic) to deliver odors at low concentrations in a reliable manner (see the "Methods" section and Supplementary Fig. 2). We used intrinsic optical imaging (see Fig. 1C, D) to quantify the pattern of glomerular activation of the target and background odors at the same concentrations used during the behavior. Intrinsic imaging[18] was performed in five naive WT mice that were awake but passively exposed to 9 s pulse odors. We quantified the odor evoked responses as *z*-scores using the 5 s air interval preceding the odor presentation as baseline. The average odor response was calculated as the mean value, averaged over repeats, of the *z*-score during a 7 s window that started 2 s after odor onset. ROIs were drawn over glomeruli with detectable negative values on their calculated *z*-score response for at

least one of the presented odors (see Fig. 1E, F). Nearby ROI (center distance < 50 μm) responded to different sets of odors consistent with ROI signals originating from different glomeruli (see Supplementary Note 1: Physical characteristics of drawn ROIs). We recorded from 155 ± 38.1 ROI (mean ± s.d.) per mouse (775 glomeruli total).

We could detect odor-evoked responses on individual odor presentations using intrinsic imaging (see Fig. 1G, H) as previously shown[18]. The imaged *z*-score response of an individual glomerulus changed from trial to trial. These variations in the imaged *z*-score reflect both real variations in the glomerular responses as well as imaging noise in the intrinsic signal. Variable glomerular responses could potentially affect odor recognition in mice whereas the imaging noise mostly affects our experimental capability to detect glomeruli that were weakly activated by an odor. Imaging noise can be reduced by averaging over multiple odor repeats whereas the mice need to make decisions based on single odor presentations.

To quantify these two variability sources, we plotted the standard deviation of the trial by trial glomerular responses against the average glomerular response (see Fig. 1I). The standard deviation increased with larger average responses, consistent with a previous report using glomerular calcium imaging[19] that showed that the standard deviation $\sigma$ was proportional to $\mu$, the average glomerular response, with the proportionality constant given by the coefficient of variation (CV). This model would predict zero variation for a glomerulus that was not activated by an odor. However, there was a measurable variance $\sigma^2_{noise}$ in the intrinsic glomerular response even in the absence of an average odor evoked response which originated from the imaging noise. Using the data from 2684 ROI-odor pairs from 3 WT mice, we estimated the coefficient of variation CV and the $\sigma^2_{noise}$ by fitting the function $\sigma^2(\mu) = \sigma^2_{noise} + CV^2\mu^2$. The estimated coefficient of variation (CV) was 0.34 (95% CI: 0.30–0.37) which is similar to the value of 0.37 ± 0.07 (mean ± SD) estimated using calcium imaging in anesthetized mice[19]. This coefficient of variation includes both uncorrelated fluctuations of individual glomerular responses as well as fluctuations that are correlated across the whole glomerular population. WT mice performance might be mostly limited by the uncorrelated variability because, by having access to all set of glomeruli, mice could compensate for the correlated fluctuation of the whole population[19]. Therefore, we calculated a coefficient of variation $CV_{uncorr}$ for the uncorrelated fluctuations (see Supplementary Fig. 3) after subtracting the population response fluctuations. $CV_{uncorr}$ was 0.25 (95% CI: 0.23–0.27) which is larger than the $CV_{uncorr}$ of anesthetized animals (0.099 ± 0.019, mean ± SD). In contrast to anesthetized mice where the correlated variability dominated, in awake mice correlated variability was less dominant.

The estimated trial-to-trial variability associated with the imaging noise $\sigma_{noise}$ was 1.59 (95% CI: 1.51–1.66). In order to determine a threshold for glomeruli activation that was reliably different from zero, we averaged the odor-evoked responses over $n$ trials resulting in a standard error of the mean of the glomerular activation of $\sigma_{noise}/\sqrt{n}$. We used at least $n = 16$ trials so the standard error of the mean was <0.42. Glomeruli-odor responses that had an average *z* score of −0.42 or larger were set to zero. This threshold was smaller than the responsive ROI−odor responses (−1.37 ± 1.17 *z*-score, mean ± s.d., $n = 6477$ ROI−odor pairs, 5 WT mice).

To test whether low levels of intrinsic glomerular activation propagated to structures downstream from the glomeruli, we performed fluorescent and intrinsic imaging in a Thy1-GCaMP6F mouse[20] that expresses GCamP6F in the mitral and tufted cells. This mouse line permits the measurement of the output signal from the olfactory bulb[21]. Glomeruli identified with intrinsic imaging colocalized to glomeruli identified using the fluorescence signal (see Supplementary Fig. 4). Even relatively small deflections in intrinsic signals in the individual glomerulus in response to odors (*z*-scores between 0 and −0.3) were correlated with statistically significant increases in fluorescent signal in the output of the bulb. This confirmed that our threshold of

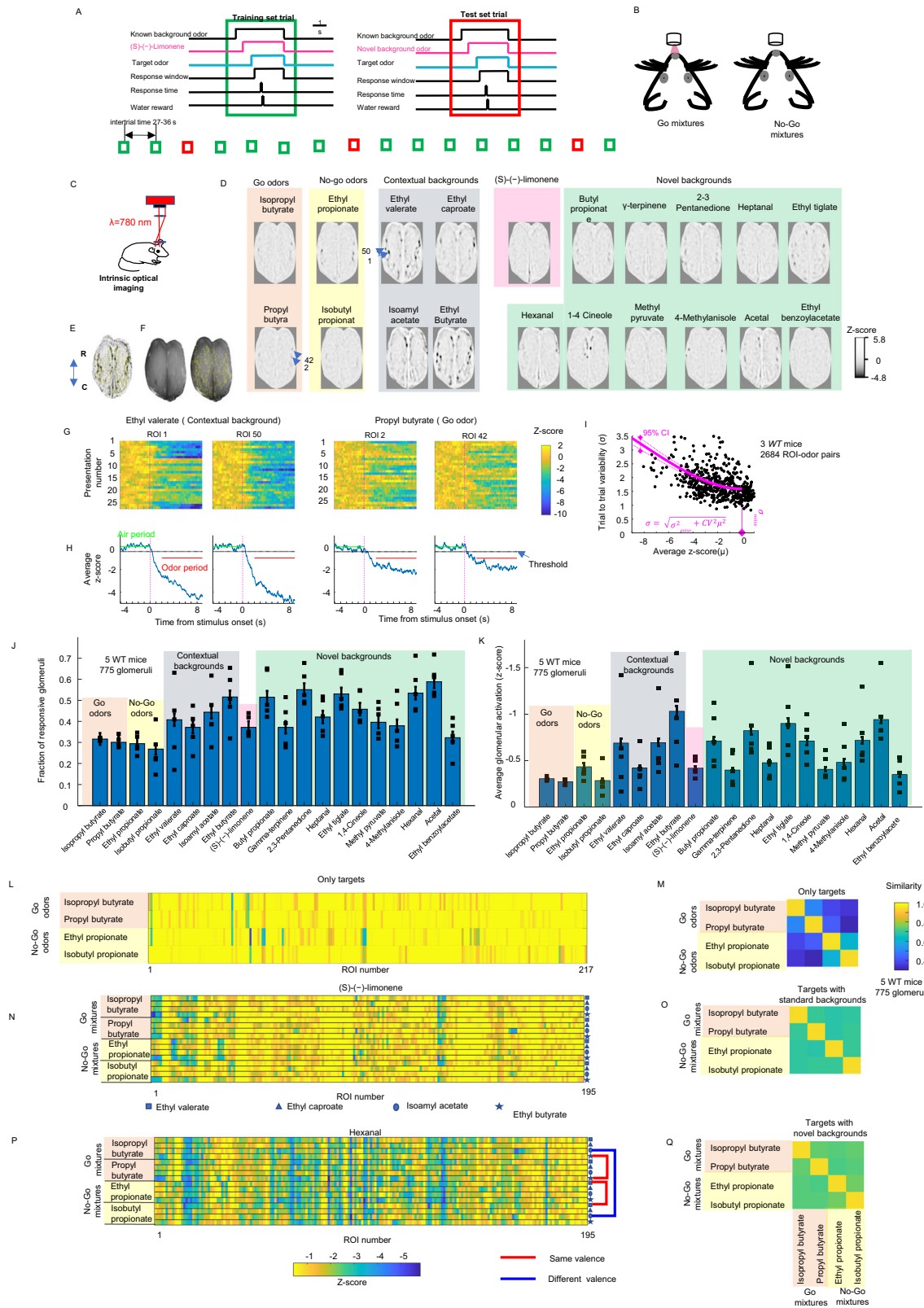

−0.42 in z-score conservatively identified glomerular responses that activated olfactory bulb output neurons.

## Glomeruli activated by target odors were also activated by background odors

We imaged responses to the four targets and the 16 background odors at the concentrations used for behavioral measurement from 5 WT mice with 155 ± 38.1 (mean ± s.d.) ROIs recorded per animal (775 ROIs total). The four target odors induced activity that exceeded our detection threshold in a smaller fraction of the available glomeruli per animal (29.5% ± 2.0%, mean ± s.d. n = 4 odors × 5 mice = 20 animal odor pairs, see Fig. 1J) compared to the four contextual background odors (43.5% ± 6.2%, mean ± s.d. n = 20 animal odor pairs, p = 0.005, t-test) and the 11 novel background odors (46.1% ± 8.8%, mean ± s.d., n = 55

**Fig. 1 | Glomerular responses of odors used for behavioral testing. A** Stimuli used during training consisted of mixtures of three odors: a contextual background odor, (s)-(−)-limonene, and a target background odor. Test stimuli were identical except that (s)-(−)-limonene was replaced by one out of 11 novel background odors. Test set trials were separated by 4–6 training set trials. **B** Head-fixed mice got rewarded with water for licking after the go target odor onset. **C** Intrinsic optical imaging was used to measure glomerular activation in response to odors used during the behavior. Minimal projection of the average *z*-score image. Activated glomeruli appeared as reductions in reflectance. **D** Average images of the *z*-score of activated glomeruli for the odors used in the behavior at the concentrations used with the test set. **E** Minimal projection of the odor responses to all presented odors. Yellow contours are the drawn ROIs. **F** Brain surface illuminated with white light. Drawn ROIs were located away from major blood vessels. **G** Single trial responses for individual odors. **H** Average *z*-score indicating the periods that were used to quantify the odor response. The air baseline period is also indicated as well as the *z*-score threshold (−0.46) used to detect glomerular responses. **I** Average odor response versus trial-to-trial variability. Trial-to-trial variability was the combination of a component that scaled with the average odor response plus a constant. Purple line indicates mean fitted trial-to-trial variability and dotted lines are the 95% confidence intervals. **J** Average fraction of glomeruli activated by odors. The error bars are 95% confidence interval, *n* = 775 ROI. Symbols correspond to individual WT mice. **K** Average *z*-score response to an odor. Error bars are s.e.m., *n* = 775 ROI. **L** Example of the average glomerular response of one WT mouse to the two go-target odors and the two no-go target odors. **M** Similarity matrix between the target odors. **N–P** Examples of the odor responses for a WT mouse to the 16 mixtures used in the training set (**N**) and to the 16 test set mixtures where the novel background odor was hexanal (**P**). **O–Q** Average similarity matrix from 6 WT mice.

animal odor pairs, *p* = 0.0029, *t*-test). The four target odors had also lower levels of average glomerular activation per animal (−0.32 ± 0.07 *z* score, mean ± s.d. *n* = 20 animal odor pairs, see Fig. 1K) compared to the four contextual background odors (−0.70 ± 0.25 *z* score, mean ± s.d., *n* = 20 animal odor pairs. *p* = 0.025, *t*-test) and the 11 novel background odors (−0.63 ± 0.25 *z* score, *n* = 55 animal odor pairs, mean ± s.d., *p* = 0.0168, *t*-test).

For the test set, (s)-(−)-limonene was replaced by one of the 11 novel background odors. (S)-(−)-limonene activated 37% of the glomeruli and this fraction was not significantly different from the 11 novel background odors (*p* = 0.35, *t*-test). (S)-(−)-limonene average glomerular activation level was −0.41 *z*-score and it was also not significantly different from the average activation level of the 11 novel background odors (*p* = 0.36, *t*-test).

Target detection in background depends on the overlap between the background and the target[7]. We assessed the overlap between the targets and the backgrounds by determining the fraction of the significantly activated glomeruli for a target odor that was also significantly activated by a background odor. A large fraction of the glomeruli that responded to a target also responded to individual contextual backgrounds (47.8 ± 8.3%, mean ± s.d., 16 target–background combinations), with 85.6 ± 3.4% (mean ± s.d., 4 target odors) of the target responding glomeruli also responding to at least one of the contextual background odors. A large fraction of the target responding glomeruli also responded to individual novel background odors (51.4 ± 10.6%, 44 target-background combinations), with 96.9 ± 2.1% (mean ± s.d., four target odors) of the target activated glomeruli responding to at least one of the novel background odors. Almost all the glomeruli that responded to the targets (99.4 ± 6%, mean ± s.d., 4 targets) would also respond to at least one of the contextual or novel background odor. The large overlap between the target representation and the background representation suggests a relatively difficult task caused by the background odors.

### Background odors increased the similarity between odor mixtures

In order to estimate the difficulty of target identification without any background odors, we calculated the similarity between the go target odors and the no-go target odors. We defined the similarity as the dot product between normalized glomerular patterns. A value of 1 indicates that the shapes of glomerular activation produced by a pair of odors were the same, whereas a value of zero indicates that the patterns were orthogonal. In order to simulate an instantiation of glomerular response to individual odor presentation, we added a level of noise proportional to that glomerular average activation level, that is

$$s_{i,j}(t) = \overline{s_{i,j}} + \overline{s_{i,j}} \mathrm{CV} \sigma \qquad (1)$$

where $s_{i,j}(t)$ is the *z*-score response of the *i*th glomerulus to *j*th odor at odor presentation *t*, $\overline{s_{i,j}}$ is the average *z*-score odor response, $\sigma$ is a

random gaussian variable from a distribution of mean 0 and variance equal to 1, and CV is the coefficient of variation. We measured the average glomerular response of the targets from 5 WT mice (775 glomeruli, see Fig. 1L, M). We simulated 100 instantiations of each average target odor response pair (500 instantiations total in 5 WT mice) using CV = $\mathrm{CV_{uncorr}}$ = 0.25. The similarity between the two go-target odors was 0.51 ± 0.17 (mean ± s.d, correlation from 500 instantiation pairs) and 0.60 ± 0.08 between the two no-go target odors. The go and no-go target odors were different, with a lower similarity of 0.37 ± 0.08 (mean ± s.d., correlation from 500 instantiation pairs). The relative difference in glomerular patterns between the go-target odors and the no-go target odors would suggest that these stimuli could be easily discriminated by WT mice.

The large glomerular activation of the background odors and the large overlap between the target and background odors increased the similarity between the go and no-go mixtures, compared to the go and no-go targets without backgrounds. In order to compute the similarity between the mixtures that included the background odors, we recorded glomerular responses from the training set and test set mixtures. We performed 32 recording sessions from 6 WT mice (161.3 ± 39.8 glomeruli per session, mean±s.d, 5163 glomeruli) where we recorded training set mixtures that included (s)-(−)-limonene and the corresponding mixtures that were part of the test set, where (s)-(−)-limonene was replaced by one of the 10 novel background odors. We simulated 100 instantiations of each odor mixture response. We calculated the similarities between mixtures that were composed of the same background odors but differed only on the target odor.

The similarity between the go mixtures (same valence) with standard backgrounds was 0.70 ± 0.13 (mean ± s.d., correlation from 7200 instantiation pairs, see Fig. 1N, O) and 0.70 ± 0.11 (mean ± s.d., correlation from 7200 instantiation pairs) for the no-go mixtures. The similarity between go and no-go mixtures (different valence) with standard backgrounds was high, 0.69 ± 0.13. In the case of mixtures that included novel background odors, the similarity between the go mixtures (same valence) was 0.75 ± 0.11 (mean ± s.d., correlation from 7200 instantiation pairs) and 0.75 ± 0.09 (mean ± s.d., correlation from 7200 instantiation pairs) for the no-go mixtures. The similarity between the go mixtures and the no-go mixtures (different valence, see Fig. 1P, Q) with novel background odors was relatively high (0.73 ± 0.12). The background odors increased the glomerular representation similarity of the mixtures that the animals needed to discriminate but within the range of previously used stimuli in rodent tasks[22–24]. The presence of background odors also increases the perceptual similarity between the mixtures[25] potentially increasing the task difficulty.

### Single glomeruli could not reliably identify targets in novel backgrounds

Which strategies could mice use to identify target odors in novel background odors? Single olfactory receptors can determine

mouse behavior in response to certain odors[17]. A very simple strategy is to identify the individual glomerulus that best discriminates the go-stimuli from the no-go stimuli from the training set and to use that best glomerulus when a mixture with a novel background odor is present. This strategy is appropriate if the glomeruli that are good discriminators for the training set are also good discriminators for the test set. We analyzed glomerular

responses from the training set and test set mixtures (32 recording sessions from 6 WT mice).

We quantified the discriminability between go stimuli and no-go stimuli of a single ROI using the area under response operating curve (auROC) calculated for both the training set and the test set. However, the best ROI for the training set did not perform as well for the test set (see Fig. 2A, B for an example). There was only a weak

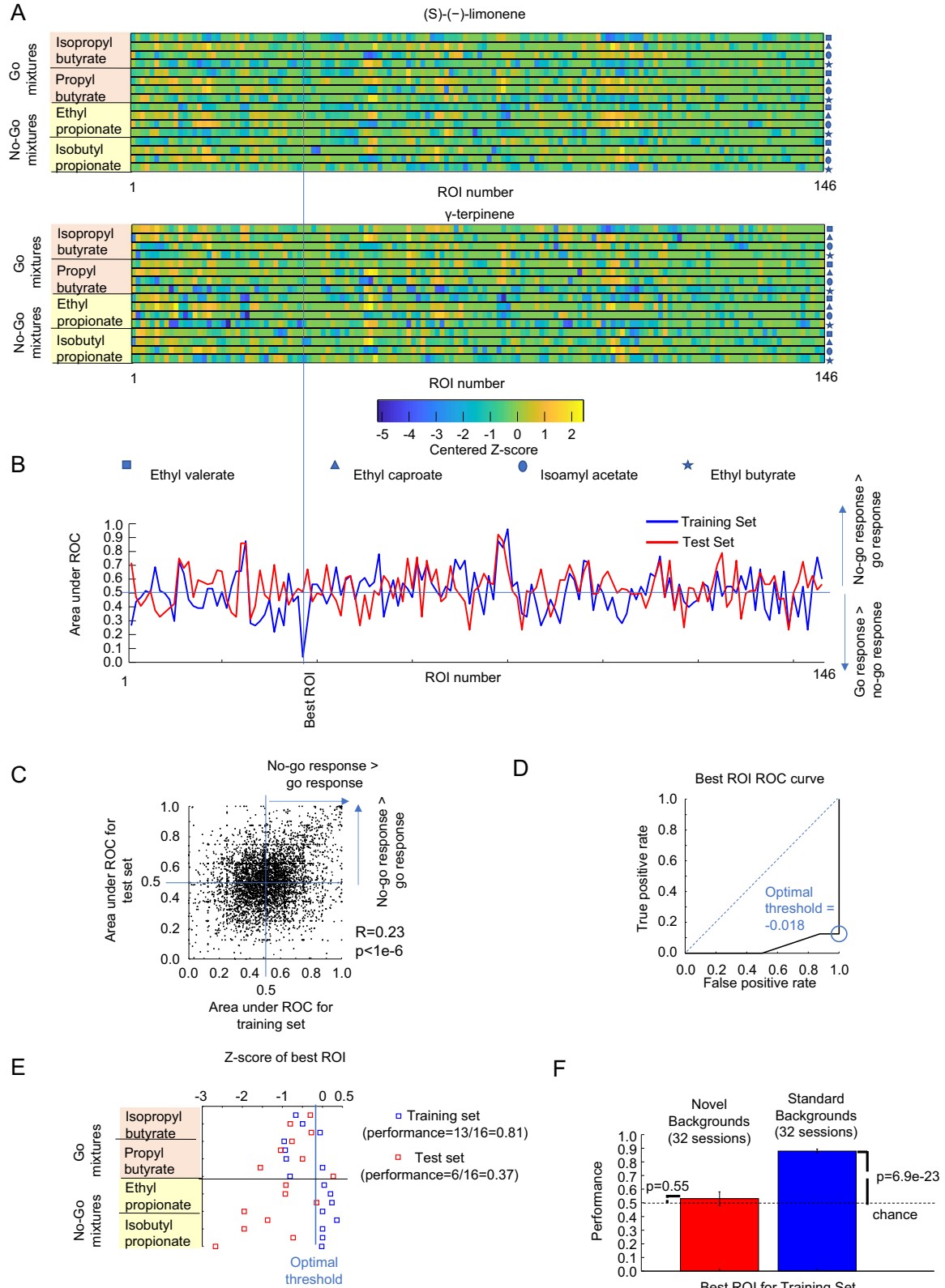

**Fig. 2 | Odor responses from individual glomeruli could not identify target odors in novel environments. A** Example of the average glomerular responses (as *z*-scores) of a WT mouse to the training set (16 mixtures) and responses to the test set with γ-terpinene as the novel background. The vertical line marks the ROI that best discriminated between the go mixtures from the no-go mixtures from the training set calculated using the auROC. **B** AuROC for the training set and the test set for all the ROIs of the example marking the position of the best ROI for the training set. **C** AuROC for the training set plotted against the auROC for the test set calculated from 32 recording sessions from 6 animals using real mixtures (5163 glomeruli). The value of *r* corresponds to the Pearson linear correlation. **D** ROC for the best glomerulus calculated from the training set for the above example. The circle indicates the optimal threshold to differentiate between the go mixtures and the no-go mixtures from the training set. **E** Example of the *z*-score responses of the best ROI, determined by the training set, to the 16 mixtures of the training set and the 16 mixtures of the test set. Each row represents the *z*-score of the best ROI for a given combination of target odor and contextual background odor from the training set (blue square) or the test set (red square). The blue vertical line represents the optimal threshold (*z*-score = −0.018) calculated from the training set. *Z*-score responses that exceed this threshold corresponded to no-go stimuli and responses below the threshold corresponded to go-stimuli. The performance for the test set was determined by the fraction of test set responses that were correctly discriminated by this threshold. **F** Performance of the best glomerulus for the training set and the test set for 6 WT mice, 32 recording sessions. Error bars represent the standard error of the mean. *p*-values were calculated using a *t*-test.

correlation between how well a single glomerulus activity separated the go mixtures from the no-go mixtures for the training set and how well it separated them in the test set. The linear correlation between the auROC for the test set and the auROC for the training set was low albeit significant (32 recording sessions from 6 animals, 5163 glomeruli, $r = 0.23$, $p < 1e-6$, Pearson linear coefficient, see Fig. 2C).

To determine the performance of an individual glomerulus strategy on the test set, we identified the glomerulus that was the best discriminator between the go mixtures and the no-go mixtures from the training set for each recording day. We calculated the discriminability as abs(auROC −0.5)+0.5. The discriminability is 1 if a glomerulus perfectly discriminates the go stimuli from the no-go stimuli and it has a value of 0.5 if a glomerulus does not distinguish them. For the best-discriminating glomerulus, we also determined the optimal threshold to distinguish between the go-stimuli from the no-go stimuli by finding the tangent to the ROC curve to a line with a slope of 45° (see Fig. 2D). Both the auROC and the optimal threshold were calculated using the function *perfcurve* from Matlab R2017b. We applied the optimal threshold to the responses of the optimal glomeruli to the test set that included the novel background odor (see Fig. 2E). We generated 100 instantiations of the activations of each glomerulus using Eq. (1). We used a value of CV = 0.34 that included both the correlated and the uncorrelated variability because we are looking at individual glomerular responses and there is no mechanism to subtract the correlated variability without information from other glomeruli.

The best glomeruli of each recording day produced a performance for the training set of 87.9 ± 1.4% (mean ± s.e.m., $n = 32$ recording sessions, 6 animals) that was significantly different from the chance level of 0.5($p = 6.9e-23$, $n = 32$ recording sessions, *t*-test). However, the performances of these optimal glomeruli were much lower for novel background odors (53.0 ± 5.1%, mean ± s.e.m., 32 recording sessions, 6 WT mice, see Fig. 2F) and they were not significantly different from chance level ($p = 0.55$, $n = 32$ recording sessions, *t*-test). In our task, a single glomerulus could not be used to detect targets in the presence of novel background odors.

### Linear classifiers and nearest neighbor classifier could identify target odors in novel background odors
The mouse olfactory system could combine information from multiple glomeruli and implement simple supervised algorithms using the feedforward connections between the olfactory bulb and its targets, including the olfactory tubercle, anterior olfactory nucleus, and piriform cortex[26]. These supervised algorithms can be trained to classify mixtures of background odors and target odors. Linear classifiers are simple algorithms, which can be implemented by a single output neuron[19] that receives glomerular activation as input and have been previously shown to match the performance of WT mice[7] in identifying target odors in known background odors[19]. The synaptic weights are adjusted, based on training examples, to detect the presence of the target odors. After training, the output neuron responds to all mixtures that include a go-target odor.

We used our imaging data for the training and test set from 32 recording sessions from 6 WT mice. We trained two types of linear classifiers: SVM and logistic regression models (see Fig. 3A, B). The logistic regression model produces a better fit for the training data, whereas the SVM is less prone to outliers as it focuses most on the data points that are closer to the separation boundary between the go-stimuli and the no-go stimuli. We trained the algorithms with the average glomerular responses of the training set and evaluated using 100 instantiations of each of the glomerular responses to the test set using Eq. (1) with CV = $CV_{uncorr}$ = 0.25. The logistic regression model generalized to novel background odors with a performance that was significantly better than chance (see Fig. 3C, D, 70.1 ± 2.5%, $p = 3.3e-9$, *t*-test). The SVM also generalized to a similar level for the novel background odors (see Fig. 3E, F, 70.1 ± 2.2%, mean ± s.e.m, with $p = 1.02e-9$, *t*-test).

The expansion in the number of neurons between the olfactory bulb and the piriform cortex could be used to implement the nearest neighbor classifier (NNC)[27]. The NNC determined which of the average glomerular responses of the training set mixtures was the best match to an instantiation of the test set mixture created using Eq. (1) (see Fig. 3G). The classification was considered a success if both the best-match mixture and the observed mixture had the same valence (go or no-go) for the target odor. We trained the NNC using the same imaging data as used for the linear classifiers. Although the NNC is not a linear classifier, NNC performance for the novel background odors was also very similar to the linear classifiers and it was also significantly higher than the chance level (see Fig. 3H, 70.8 ± 2.9%, $p = 3.3e-9$, *t*-test). The performance of the linear classifiers and the NNC were robust to changes in the *z*-score used as threshold (see Supplementary Note 2: Changes in odor-evoked *z*-score detection threshold did not affect algorithms performance and Supplementary Fig. 6). Our results show that odor identification in novel environments is a relatively complex behavior that could be solved using information from multiple glomeruli.

### Plateau performance was achieved with a small fraction of the glomeruli
Individual glomeruli that separated the go stimuli from the no-go stimuli from the training set did not generalize to mixtures where the targets were embedded with novel background odors. The linear classifiers and the NNC could generalize to a performance of 70% by combining the information from different glomeruli. We wondered what was the minimal number of glomeruli necessary for successful generalization for these algorithms.

To determine how the performance of the linear classifiers changed as we reduced the number of glomeruli used, we used a sparseness constraint. The regularization constant λ determines the number of glomeruli that are assigned a non-zero weight in the classifier (see Fig. 4A–D). Larger values of λ produced solutions that used fewer glomeruli. We tested 21 values of λ between 0 (all available glomeruli included) and 1 (few glomeruli included). The performance of the SVM classifier increased as more glomeruli were included and

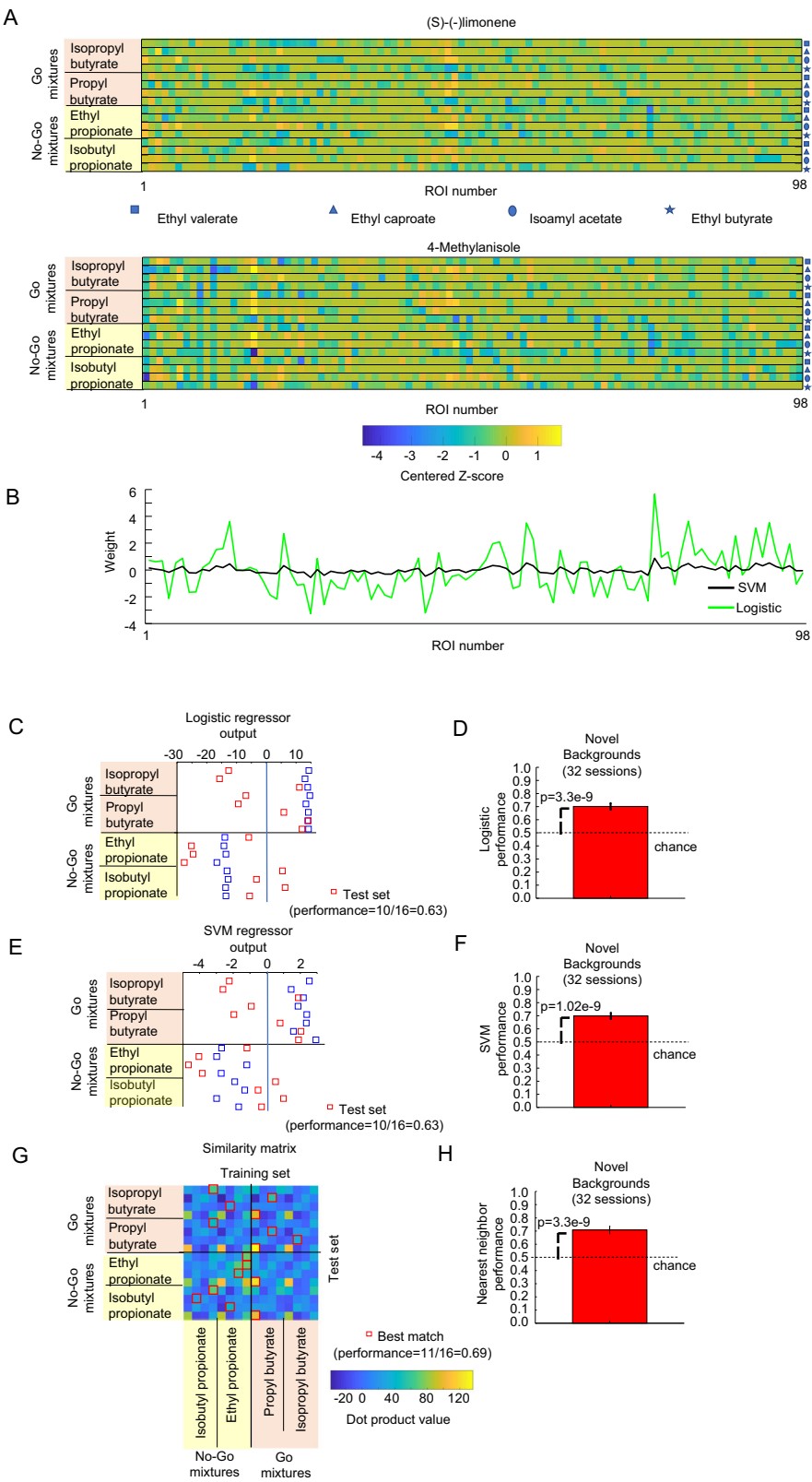

reached a peak value of 73.7 ± 2.2% (mean ± s.e.m., $n$ = 32 sessions) at a value of $\lambda = 0.25$, which corresponded to an average number of only 24.5 ± 2.2 glomeruli. The peak performance was not significantly different ($p$ = 0.23, $t$-test) from the performance using the full set of available glomeruli (70.1 ± 2.2%, 161.3 ± 39.8 glomeruli, mean ± s.d., 32 recording sessions). The logistic regression model had a similar behavior as the SVM model, increasing the performance with more

glomeruli and reaching a peak performance (73.1 ± 2.2%) at $\lambda = 0.1$ with 36.3 ± 2.9 glomeruli (see Fig. 4F). The performance was not different from the performance using the whole set of available glomeruli ($p$ = 0.33, $t$-test, 70.1 ± 2.5%, 32 recording sessions).

We also systematically reduced the number of glomeruli used to calculate the NNC by increasing the discriminability threshold for the selection of the included glomeruli. The selectivity was based only on

**Fig. 3 | Linear classifiers and NNC could identify odors in novel environments. A** Example of the average glomerular responses of a WT mouse to the 16 mixtures of the training set and responses to 16 mixtures of the test set with 4-methylanisole as the novel background. **B** Estimated weights of SVM with linear kernel and logistic linear classifiers trained using the training set for the example. **C** Output produced by multiplying the weights $w_i$ of the logistic linear classifier and adding the constant bias term $w_0$ with the glomeruli activation. Each row represents the output produced by a given combination of target odor and contextual background odor from the training set (blue squares) or the test set (red squares). Correct performance consisted of positive responses for go stimuli and negative responses for no-go stimuli. **D** Performance of the logistic regressor calculated on data from 6 WT mice, 32 recording sessions on the test set, with the performance of each recording day using 100 instantiations of each of the responses of the test set using Eq. (1).

Error bars are s.e.m. and $p$-values were calculated using a two-tailed $t$-test. **E** Same as **C** but using the SVM weights. **F** Performance of the SVM for the data from 6 WT mice, 32 recording sessions. Data are presented as mean values ± s.e.m. and $p$-values were calculated using a two-tailed $t$-test. **G** Example of a matrix of dot products between the 16 training set mixtures and the 16 test set mixtures used for calculating the NNC. Each target odor appeared mixed with each of the four contextual background odors. For each mixture in the test set, the red squares indicate the location of the most similar mixture from the training set. If the valence (go or no-go) of the target odor in the test mixture matched the valence of the most similar mixture (Nearest Neighbor), the trial was considered correct. **H** The performance of the NNC for novel background odors for 6 WT mice, 32 recording sessions. Data are presented as mean values ± s.e.m. and $p$-values were calculated using a two-tailed $t$-test.

the responses to the training set and was calculated using the above formula abs(auROC −0.5)+0.5 (see Fig. 4E). We tested 10 discriminability threshold values between 0.5 (all glomeruli included) and 1 (only glomeruli that perfectly discriminate the training set included). The nearest neighbor classifier (see Fig. 4F) had a similar response as the linear classifiers with performance increasing with more included glomeruli, reaching a peak performance of 78.0 ± 2.4% at selectivity = 0.625, which corresponded to 27.1 ± 3.6 glomeruli. The peak performance was not significantly different from the performance that included all the glomeruli (70.8 ± 2.9%, $p = 0.0571$, $t$-test, 32 sessions). Thus, linear classifiers and the NNC reached a plateau in performance with few glomeruli.

### WT mice could identify target odors among novel background odors

Four head-fixed, water-deprived WT mice were trained to detect the presence of target odors and were rewarded with water if they licked in response to the target go-odors (hits). If they licked before the target go-odors appeared (early licks) or if they licked in response to no-go target odors (false alarms), odor delivery was stopped and the mice were given a time-out. Mice were trained for ~9 days (see training protocol) to detect targets among known backgrounds. Once animals displayed consistent performance at the final concentration (>80% correct for more than 50 trials), trials with novel background odors were introduced. Presentation of the novel background odor and the contextual background odor preceded the onset of the target odor by 0.75 s and 1.5 s respectively (see Supplementary Fig. 7 for a detailed description of task timing). This asynchronous odor presentation emulates a situation where the animal is in a novel environment when the target appears. We used a long inter-trial interval (30.4 ± 3.3 s, mean ± s.d.) to avoid receptor adaptation effects and to permit the airflow to clean the odor delivery system.

WT mice performance for known background odors was high (87.4%, $n = 1563$ trials), and it was significantly higher than the 50% chance level ($p = 1.24e{-}215$, binomial test). The identity of the contextual background odors had a very small effect on performance with the lowest performance being 84.6% for ethyl butyrate and the highest performance being 88.9% for isoamyl acetate. There was little variation in the performance for the individual go-target odors: 93.9% for isopropyl butyrate and 93.0% for propyl butyrate. There was also not much variation in the performance for the individual no-go target odors: 84.6% for ethyl propionate and 78.5% for isobutyl propionate. WT mice almost never failed to detect the go targets (only 0.7% of the trials were misses), as previously described in other go/no-go behavioral tasks[7,28].

The trials with different novel background odors were separated by five or six trials with known background odors. These trials with known background odors were included to maintain the animal's motivation with an easier task and to keep reinforcing the target odors. Each trial of a novel background odor was separated by another trial with the same background odor by ~25 min. To prevent learning of the

novel background odors, each novel background odor was presented, at most, four times per day on two separate days, giving a maximum of eight trials per novel background odor per animal.

WT mice were able to identify targets in the presence of novel background odors at higher than chance levels (76.9%, $n = 334$ trials for the novel background trials, $p = 3.08e{-}24$, binomial test, see Fig. 5A). Interestingly, when the novel background odors were used, WT mice had an increase in the number of misses to 5% (17/334 of trials, see Fig. 5B) from 0.7% with known background odors, which was significant ($p = 2e{-}6$, Fisher exact test adjusted using the Bonferroni correction) suggesting increased difficulty in detecting the target odor in novel backgrounds compared to known backgrounds. WT mice also significantly increased the number of early licks for novel backgrounds (from 3.9% of 1563 trials to 7.7% of 334 trials, $p = 0.011$, Fisher exact test adjusted using the Bonferroni correction). The fraction of false alarms was not significantly changed (10.1% of 334 trials for novel backgrounds, 7.9% of 1563 trials for known background odors, $p = 0.57$, Fisher exact test adjusted using the Bonferroni correction).

WT mice were able to identify target odors among novel background odors, even on the first presentation (see Fig. 5C), with a performance at 79% ($p = 6.4e{-}5$, binomial test, $n = 44$ trials, 4 animals). There was no systematic increase in performance as the animals familiarized themselves with a novel background odor, and there was no significant positive linear correlation between a background odor presentation number and performance ($r = -0.48$, $p = 0.22$, Pearson linear correlation, $n = 8$). To directly test whether familiarity with the novel background odors increased performance, we trained a different cohort of five WT mice in the same task (see Supplementary Note 3: Familiarity with background odor did not increase target recognition in novel backgrounds for WT mice). This new cohort had been exposed to 5 of the 11 novel background odors in their home cages for 30 min over 6 days before behavioral training started. The performance of this cohort in the novel background task in response to the previously exposed 5 odors was 79.6 % (211 trials), which was not significantly different ($p = 0.89$, Fisher exact test) from the original group performance on these odors (78.8%, 132 trials, 4 animals). Both exposures to the same novel background odor during the task and longer exposure times outside the task context did not produce systematic performance increases, indicating that WT mice employed a robust algorithm that did not require previous knowledge of the background odor to detect target odors.

### WT recognized the novel background odors after a single presentation

We wanted to determine how quickly the WT mice familiarized themselves with a novel background odor by measuring the sniffing response. Mice respond to novel odors by increasing their sniff rate[29–31] as well as orienting their nostrils to the location of the novel odor[32]. We monitored non-invasively animal sniffing during the olfactory CAPTCHA task using a flow sensor attached to the odor delivery tube[33] (see Fig. 5D, E). Animals had a base sniff rate of 2.18 ± 0.03 sniffs

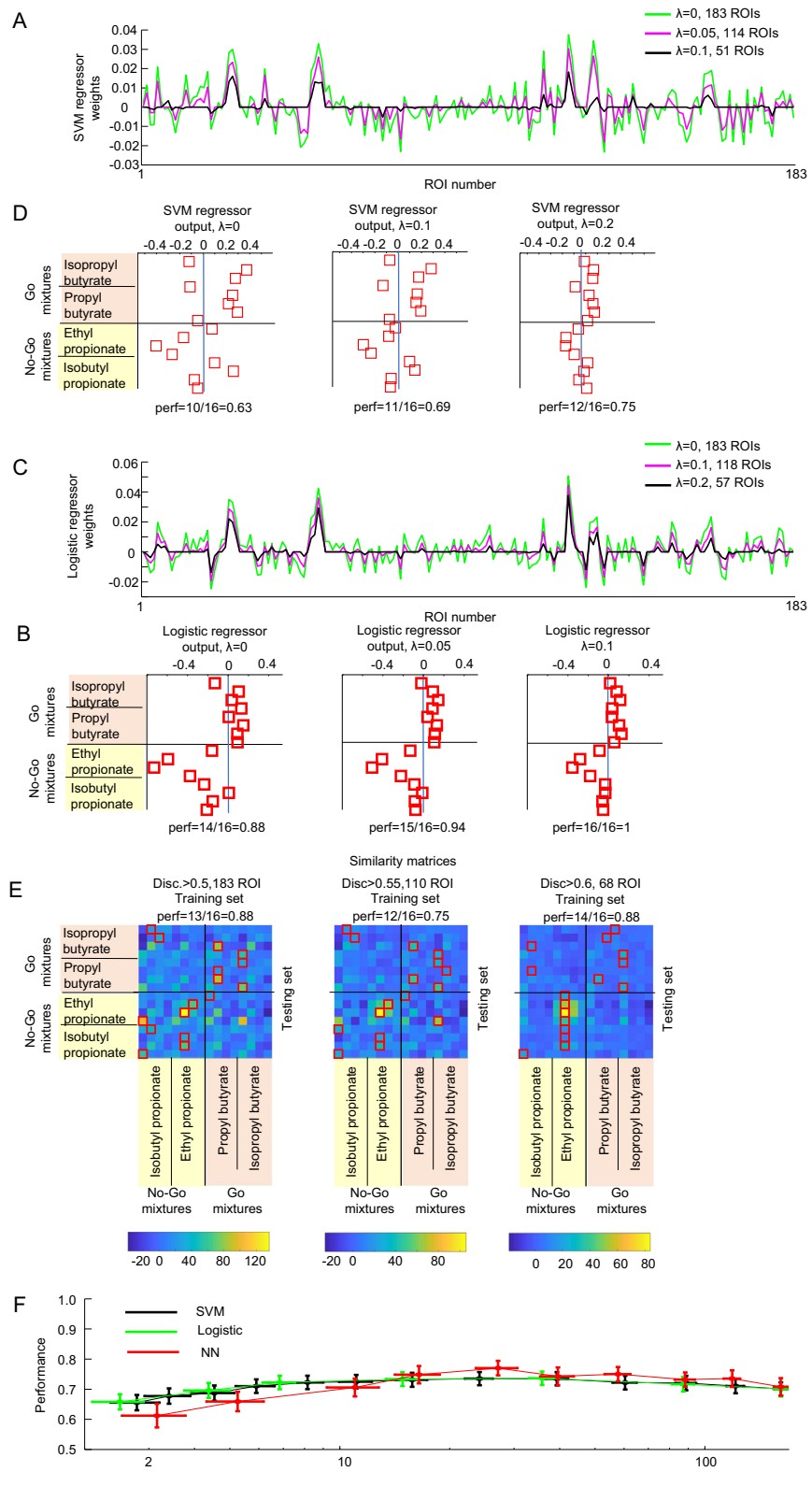

**Fig. 4 | Linear classifiers and the NNC did not require the use of all available glomeruli to reach plateau performance. A** Example of regressor weights for the linear SVM as the sparseness constraint is changed. **B** The SVM regressor weights were applied to test set mixtures where methyl pyruvate was the novel background odor. **C** and **D** Logistic regressor weights were also calculated from the training set and were applied to the test set. **E** Number of glomeruli used in the NNC was changed by thresholding based on the auROC of the training set of individual

glomeruli. Similarity matrices changed as the threshold was changed. The red squares indicate the best match to the training set. **F** Performances of the linear classifiers and the NNC as a function of the number of ROI included calculated for 32 recording sessions, 6 WT mice. Vertical error bars are the s.e.m. of the performance of the classifiers and horizontal error bars are the s.e.m. for the number of glomeruli used for the classifiers.

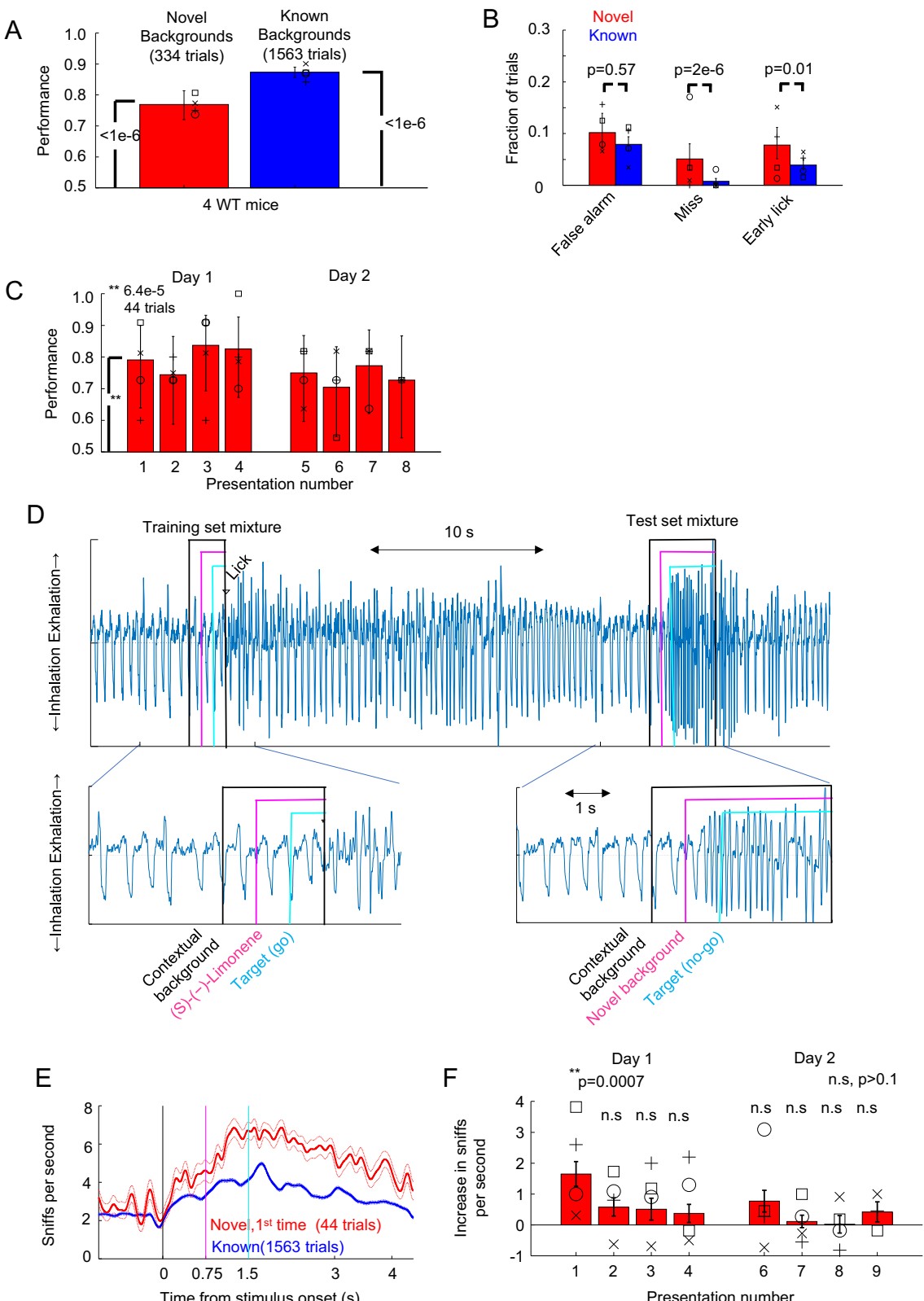

per second (mean ± s.e.m., $n = 1563$ trials, 4 animals) just before the onset of the odors. As the contextual background arrived, the sniff rate increased to $3.02 ± 0.05$ sniffs per second ($n = 1563$ trials, 4 animals) and it further increased with the onset of (s)-(−)-limonene to $3.93 ± 0.06$ sniffs per second ($n = 1563$ trials, 4 animals). After the onset of the target, the sniff rate further increased to $4.23 ± 0.05$ sniffs per second. On the first presentation of the novel background odors,

the sniff rate in the first second following the onset of the novel background odor increased compared to the preceding trials with known background odor ($1.65 ± 0.40$ extra sniffs per second, mean ± s.e.m., 44 trials, 4 animals, $p = 0.0007$ Bonferroni corrected $t$-test) indicating that the novel odors were perceived as a novel. Interestingly, as previously shown for single odors[29,30,32], average sniff rates for novel odors dropped to rates similar to known background trials

**Fig. 5 | WT mice successfully solved olfactory CAPTCHA (asynchronous case) and explored novel background odors. A** Group performance of four WT mice for known (1563 trials) and novel (334 trials) background odors. Individual WT mice performances are represented by black symbols. *p*-values were calculated using a two-tailed binomial test and error bars are the 95% confidence interval. **B** Fraction of trials of types of errors with the 95% confidence interval for known and novel background odors. *p*-values were calculated using a two-tailed Fisher exact test adjusted using the Bonferroni correction. **C** Performances with the 95% confidence interval for the novel background odor as a function of the novel background odor presentation (8 presentation total over 2 days). Symbols represent individual animals. *p*-value was calculated for the first presentation to determine whether the

performance was different from 50% (chance) using a two-tailed binomial test. **D** Example respiratory signal with inserts for standard odor presentation (left) and novel odor presentation (right). **E** Mean ± s.e.m. respiration signal for four WT mice for the first presentation of novel background odors compared to the presentation of the interleaved trials with standard background odors. **F** Mean ± s.e.m. of the increase in sniff rate for novel odor compared to the preceding trial with (s)-(−)-limonene as a function of the novel background odor presentation number (8 presentations total over 2 days), *n* = 44 trials per presentation. *p*-values were calculated using the Bonferroni-corrected two-tailed *t*-test. Symbols represent average increases of individual WT mice.

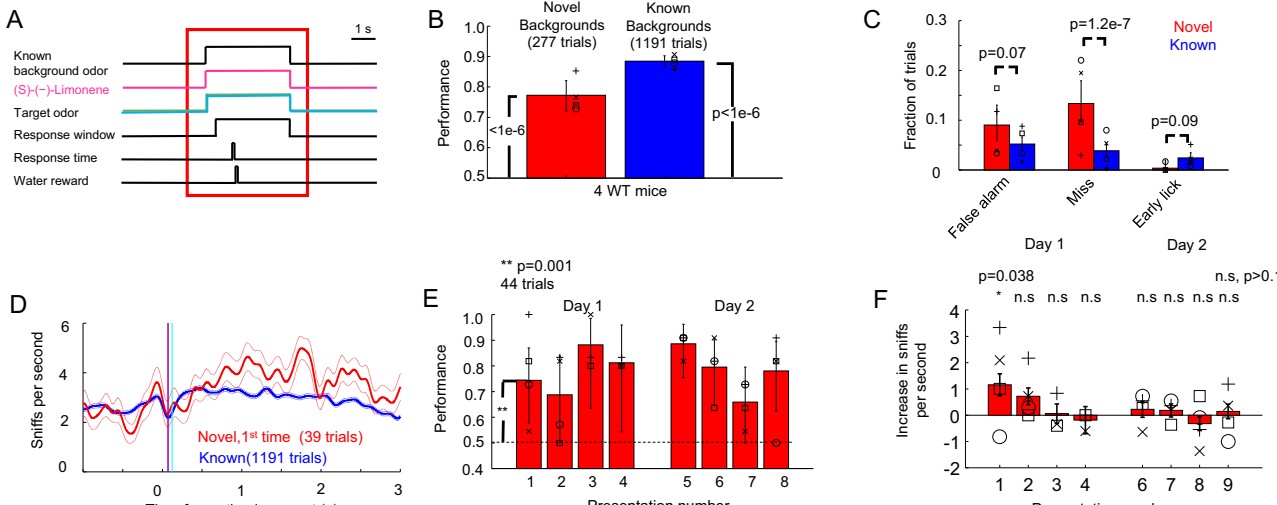

**Fig. 6 | WT mice successfully solved olfactory CAPTCHA (synchronous case). A** Stimuli were the same as in the asynchronous case but the target appear only 50 ms after the onset of the background odors. **B** Group performance of four WT mice for known and novel background odors in the synchronous case. Error bars are the 95% confidence interval (*n* = 277 trials for novel backgrounds and *n* = 1191 trials for known backgrounds) and symbols are the performance of individual WT mice. *p*-values were calculated using the binomial test. **C** Fraction of types of errors and 95% confidence interval for known and novel background odors. *p*-values were calculated using the Fisher exact test adjusted using the Bonferroni correction.

**D** Mean ± s.e.m. respiration signal for four WT mice. **E** Performance and 95% confidence interval for the novel background odor as a function of the presentation number. *P*-value was calculated for the first presentation to determine whether the performance was different from 50% (chance) using a binomial test, *n* = 44 trials. **F** Mean ± s.e.m. of the increase in sniff rate for novel background odor compared to the preceding trial with (s)-(−)-limonene as a function of the novel background odor presentation number, *n* = 39 trials. *p*-values were calculated using the Bonferroni corrected *t*-test. Symbols represent average increases of individual WT mice.

(*p* > 0.1, Bonferroni corrected *t*-test) on further exposures, indicating that the animals recognized the novel background odor (see Fig. 5F). This was quite remarkable, because the novel background odors were presented with different targets and contextual background odors, and subsequent presentations of the same novel background odor were separated by at least 25 min.

We also quantified an animal's familiarity with the novel odors using the number of sniffs that it took from target onset to a correct lick response. During the known background trials, the animals took 2.44 ± 0.05 sniffs (718 responses). Animals took a larger number of sniffs (3.46 ± 0.64, 13 responses) on the first exposure to a novel background and it was significantly higher than the number of sniffs taken for known backgrounds (Bonferroni corrected *t*-test, *p* = 0.0374, see Supplementary Fig. 8). Consistent with increasing familiarity after the first exposure and reduction of exploratory behavior, the number of sniffs before licking fell after the first presentation of a novel odor to a value that was similar to the number of sniffs in the presence of known backgrounds (Bonferroni corrected *t*-test, *p* > 0.5). WT mice increased their sniff rate and number of sniffs in the presence of novel odors but this increase disappeared after a single exposure. Surprisingly, there was no increase in performance even when WT mice recognized the novel background odor, suggesting that knowledge of the background odor, demonstrated through their sniff rate, was not necessary for target recognition.

## Synchronous presentation of the target and backgrounds increased baseline sniff rates

In the first set of experiments, the target odor appeared 0.75 s after the onset of the novel background odor. Unsupervised algorithms[34] can use differences in the temporal profile between the background odor and target odor to segment them. These differences could also be onset delays[35] between the target and the background[30]. In order to directly test the contribution of this asynchrony in the onset between novel background and target for target identification, we trained a different group of five WT mice in a condition where the target and background odors were presented synchronously (synchronous case), rather than the target odor appearing after the novel background odor (asynchronous case, see Fig. 6). This synchronous odor presentation emulates a situation where the animal is suddenly presented with a mixture that includes a novel odor. One mouse's performance on known background odors was close (75.4 ± 1%), but did not reach our threshold of 80% for the performance of known background odors on none of the sessions with novel background odors and was excluded from further analysis. For the known background odors, WT mice in the synchronous case performed significantly higher than the chance level (88.5%, *n* = 1191 trials, *p* = 4e−176, binomial test, 4 animals). The identity of the contextual background odor resulted in only small variations in performance, with the lowest performance for the contextual background being 86.5% for ethyl valerate and the highest one

being 91.8% for ethyl caproate. There was not much variation in the performance for the individual go-target odors: 92.0% for isopropyl butyrate and 86.2% for propyl butyrate. There was also not much variation in the performance for the individual no-go target odors: 91.2% for ethyl propionate and 84.4% for isobutyl propionate.

In the synchronous odor presentation, the target odor started at the stimulus onset, making it easier to be missed by the animals compared to the asynchronous case where the target started 1.5 s after stimulus onset. Compared to the asynchronous case, there was a small but significant increase in the number of misses in the synchronous case compared to the fraction of misses in the asynchronous case (from only 0.7% (12/1563) to 3.9% (46/1191), $p = 2.1e-8$, Fisher exact test, see Fig. 6C). WT mice increased their baseline sniffing during synchronous odor presentation compared to the asynchronous case suggesting an enhanced level of alertness in order not to miss the target. The base rate of sniffing before odor onset was higher for WT mice that were doing the synchronous task ($2.62 \pm 0.04$ sniffs per second, $n = 1191$ trials, $n = 4$ animals) compared to WT mice doing the asynchronous task ($2.18 \pm 0.03$, $n = 1563$ trials, $p = 6.9e-19$, $t$-test, $n = 4$ WT mice). These differences were not due to differences in animal batches. The same WT mice that did the synchronous task had also significantly lower baseline sniff rates ($2.07 \pm 0.04$ sniffs per second, $n = 1209$, $n = 4$ WT mice) when they were performing the asynchronous task with known background odors during their training process, compared to when they performed the synchronous task ($p = 5.99e-23$, $t$-test). The synchronous task seems to require an increased baseline attention level given the unpredictable appearance of the target.

Once odors started, the increase in sniff rates was smaller in the synchronous case compared to the asynchronous case as animals were already at a higher baseline sniff level (see Fig. 6D). The increase in sniff rate with respect to baseline during the presentation of the target for WT mice performing the synchronous behavior ($0.59 \pm 0.050$ extra sniffs per second, 1191 trials) was significantly smaller than the increase seen in WT mice performing the asynchronous behavior ($2.06 \pm 0.05$ extra sniffs per second, 1563 trials, 4 animals, $p = 3.31e-85$, $t$-test).

## WT mice could detect targets with synchronous presentation with novel background odors

WT mice were able to identify the target odors with the synchronously presented novel background odors at higher than chance levels (77.2%, $n = 277$ trials, 4 animals, $p = 7.4e-21$, binomial test, see Fig. 4B). Similar to the asynchronous case, the WT mice had a significant increase of not licking for the go-stimulus (misses) with novel backgrounds compared to mixtures with known backgrounds (13.3%, 37/277 trials of novel background trials, versus 3.8%, 46 of 1191 trials of known background odors, Bonferroni corrected Fisher exact test, $p = 1.2e-7$, see Fig. 6C). The total performance for the synchronous presentation (77.2%, $n = 277$ trials, 4 animals) was almost identical (Fisher exact test, $p > 0.9$) to the asynchronous case performance of the previous group of 4 WT mice (76.9%). WT mice also identified the target odors among novel backgrounds at higher than chance levels (74.3%, 39 trials, $p = 0.001$, binomial test), even on the first presentation of a novel background odor (see Fig. 6E). Thus, the difference in the temporal profile between the novel background odor and the target odor did not contribute significantly to the performance of our task.

WT mice during the synchronous task also reacted to the first presentation of a novel background odor by increasing their sniff rate in the first second following the onset of the stimulus ($1.15 \pm 0.42$ extra sniffs per second, mean $\pm$ s.e.m., 39 trials, 4 animals, $p = 0.038$ Bonferroni corrected single tailed $t$-test) with respect to the same period in the known background odor (see Fig. 6D). Although this increase in sniff rate in response to novel odors was higher in the asynchronous case ($1.65 \pm 0.4$ extra sniffs per second) compared to the synchronous case ($1.15 \pm 0.42$ extra sniffs per second), the difference was not significant ($p = 0.39$, $t$-test). The increased sniff rate in response to a novel

background (see Fig. 6F) also adapted after a single presentation to a value that was not significantly different from the sniffing to known backgrounds ($p > 0.5$, Bonferroni corrected single-tailed $t$-test). Animals took $2.69 \pm 0.07$ sniffs (519 trials, 4 animals) before licking for the go odor (hit) for known backgrounds. The number of extra sniffs before a correct lick response was significantly increased (see Supplementary Fig. 9, $p = 0.02$, Bonferroni-corrected $t$-test) only for the second odor presentation ($4.6 \pm 0.9$ sniffs, $n = 6$ trials) for novel background odors and it was not significantly different from the known background odor number of extra sniffs for the other presentations ($p > 0.5$, Bonferroni corrected $t$-test). There was also no significant correlation between the odor presentation number and the performance of the WT mice ($r = -0.02$, Pearson linear correlation, $n = 8$, $p = 0.95$); hence, as with the asynchronous case, there was no improvement in performance with further exposures to a novel background odor.

We wondered whether rapid sniff rates would correlate with increased performance as described for other olfactory tasks[36]. When animals increased their sniff rates, inhalations become shorter[31]. Brief inhalations correlated with higher performance only on novel background odors, suggesting that rapid inhalation-induced adaptation[30] might contribute to odor identification in novel environments (see Supplementary Note 4: Briefer inhalation widths were correlated with increased WT mice performance in novel environments and Supplementary Fig. 10); however, the similar performance for the synchronous and asynchronous case showed that temporal desynchronization between background and target is not the only mechanism used in odor identification in novel olfactory environments.

## Classifiers that included more glomeruli had higher correlation with animal behavior

Linear classifiers and nearest neighbor classifiers could identify odors in novel environments at similar levels that approached WT mice performance. Classifiers that used a small fraction of the available glomeruli reached a performance similar to classifiers that used all available glomeruli. To determine which of the models was a better descriptor of the actual performance of the WT mice, we compared WT mice performance on individual novel background odors with the performance of the different classifiers as we varied the number of glomeruli used.

We used the imaging data from 32 recording sessions from 6 WT mice, where we used training set and test set mixtures. To determine the response of a classifier to an individual novel background odor on each recording day, we created 100 instantiations of each mixture of the training set and the test set using Eq. (1) using $CV = CV_{uncorr} = 0.25$. For the linear classifiers, we calculated a classifier that separated the go stimuli from the no-go stimuli from the training set. Then, we applied that classifier to the test set and determined the fraction of trials that were correctly classified. We used a similar procedure as the one used for the linear classifiers to calculate the NNC performance of individual novel background odors. To determine the performance of a novel background odor, we averaged the responses to all the imaging sessions that used that novel background odor as part of the test set. We plotted the performance of a classifier against the behavioral performance of the two groups of animals that used the asynchronous odor presentation (4 WT mice) and synchronous odor presentation (4 WT mice). We calculated the linear correlation ($r$) between a classifier performance and the WT behavioral performance. To calculate the distribution of correlation values ($r$) for each classifier, we performed a Montecarlo simulation where we repeated the above procedure 500 times.

When using all the available glomeruli, the nearest neighbor classifier was more correlated ($0.58 \pm 0.01$, mean $\pm$ s.d., 500 simulations, see Supplementary Fig. 11A–C) with the performance of the animal compared to the SVM ($0.35 \pm 0.02$, mean $\pm$ s.d., 500 simulations, $p < 1e-6$,

*t*-test) and logistic classifier (0.39 ± 0.02, mean ± s.d., 500 simulations, *p* < 1e−6, *t*-test). These differences in correlation with animal behavior were statistically significant, although the average performances of these algorithms were very similar (see Fig. 4F).

The performance of the classifiers reached a plateau using 27.1 ± 3.6 glomeruli for the NNC, 24.5 ± 2.2 glomeruli for the SVM classifier, and 36.3 ± 2.9 for the logistic regressor (mean ± s.e.m.). We wondered whether a classifier's correlation with behavior would also reach a plateau performance with fewer glomeruli included. As above, we varied the number of glomeruli used by the NNC by changing the minimal selectivity for the training set of the glomeruli used in the similarity calculation. For the linear classifiers, we reduced the number of glomeruli used by increasing the value of the regularization constant $\lambda$.

For all the classifiers, the correlation between the classifier performance and the animal behavior monotonically decreased as we reduced the number of glomeruli included, although the classifier performance was maintained (see Supplementary Fig. 11D–G). The correlation with behavior for the NNC became significantly reduced with respect to the one calculated using the full set of glomeruli when the number of glomeruli dropped below 88 glomeruli ($p < 1e-6$, *t*-test, $n = 500$ simulations). There was also a monotonic reduction in the correlation between the SVM and the logistic regressor as the number of glomeruli used is reduced. Although the NNC performance reached a plateau after 27.1 ± 3.6 glomeruli for our behavior, correlation with behavior increased with more glomeruli being included suggesting that WT mice employ a large fraction of glomeruli when making decisions in our task.

### WT mice performance was less sensitive to diversity of training data than the NNC

NNC requires a training set that includes an appropriate match to a new mixture; otherwise, the NNC performance might decrease (see Fig. 7A). To confirm this, we reduced the diversity of training examples, from 16 training mixtures (4 contextual background odors x 4 target odors) to 8 mixtures resulting in a reduced training set. The combinations of contextual backgrounds and target odors were different between the reduced training set and the reduced test set (see Fig. 7B). Each contextual background odor was presented with a target go odor and a target no-go odor to avoid creating biases for the contextual background odors. The reduced test set consisted of 88 testing mixtures (down from 176 mixtures) that included the 11 novel background odors. The reduced set task became harder not only because of the presence of a novel background odor but also because the known contextual background and target were a novel combination that was not presented together in the reduced training set. We tested the classifiers using the imaging data of animals exposed to real mixtures (32 recording sessions, 6 WT animals). As expected, the performance of the NNC decreased significantly ($p = 3.5e-04$, *t*-test, see Fig. 7C) from 70.8 ± 2.9% (mean ± s.e.m.) when trained with the full set to 53.3 ± 3.7% (mean ± s.e.m.) when trained with the restricted set.

The performance of the linear classifiers also decreased when trained with the restricted set. The logistic regression classifier performance dropped from 70.1 ± 2.5% (mean ± s.e.m.) when trained with the full set to 56.2 ± 4.0% (mean ± s.e.m., $p = 0.0045$, *t*-test) when trained with the reduced set. The SVM classifier performance dropped from 70.1 ± 2.2% (mean ± s.e.m.) when trained with the full set to 55.8 ± 4.0% (mean ± s.e.m., $p = 0.003$, *t*-test). In fact, the performances of the three classifiers when trained with the reduced set were not significantly different from the chance level of 50% ($p = 0.38$ for NN; $p = 0.16$ for SVM; $p = 0.14$ for logistic, *t*-test). If WT mice employed these algorithms, their performance should drop to chance levels if the diversity of the training data is limited.

In order to test whether WT mice would be able to identify target odors when reducing the diversity of the training set, we trained a new

batch of three WT mice using the restricted training set with the asynchronous presentation of the odors. The WT mice performance on novel backgrounds with the restricted test set was 74.1% (228 trials) and it was significantly different from chance level ($p = 3.00e-14$, binomial test). The performance was not significantly different ($p = 0.48$, Fisher exact test) from the previous group of animals trained with the full set of training mixtures (76.9%, 334 trials, 4 WT mice, asynchronous task). Thus, WT mice could use less diverse training data compared to the NNC or the linear classifiers without having their performance affected.

### A sparse deconvolution algorithm had better performance than NNC and linear classifier for reduced training data

NNC and linear classifiers could not match the behavior of the WT mice when the diversity of training examples was reduced. One possible way to improve the performance respect of the NNC and the linear classifiers would be to use sparse deconvolution algorithms which have been proposed as being implemented by the nervous system[37–40]. These deconvolution algorithms store the glomerular patterns produced by all odors known to an animal in a dictionary. The algorithms decompose an observed signal into contributions selected from this dictionary, while minimizing the number of odors used, permitting generalization to multiple combinations of dictionary odors. These algorithms are more complex than NNC and linear classifiers. However, these sparse deconvolution algorithms offer the advantage over the NNC and linear classifiers that once an animal learns a dictionary, the animal can apply it for multiple combinations of odors, whereas the NNC and linear classifiers performance depends on the diversity of the training examples.

To determine whether deconvolution algorithms could account for the performance of the WT mice in the presence of novel background odors that are not part of the dictionary of odors used, we performed simulations using the Lasso (*least absolute shrinkage and selection operator*)[41], a standard sparse representation algorithm. The Lasso finds the combination of elements of a dictionary that could reconstruct the observed pattern of glomerular activation in the least-square error sense. The reconstruction produced by the Lasso minimizes the sum of the absolute value (or L1 norm) contribution of the dictionary elements weighted by a regularization constant $\lambda$ that is:

$$\text{Cost} = \sum_{i=1}^{n} \left( s_i - \sum_{j=1}^{m} c_j d_{i,j} \right)^2 + \lambda \sum_{j=1}^{m} |c_j| \tag{2}$$

where *n* is the number of glomeruli, *m* is the number of elements in the dictionary, $s_i$ is the observed activation of the *i*th glomerulus, $d_{i,j}$ is the *i*th glomerular activation of the *j*th dictionary element, $c_j$ is the concentration estimated and $\lambda$ is the sparseness constrain.

We tested the Lasso by presenting it with glomerular activations using artificial mixtures (see Fig. 7D, E for an example). We used the imaging data from 5 individual WT mice (775 glomeruli). For each mouse, we tested the Lasso with the reduced test set that included 88 odor mixtures with novel background odors.

When presented with a mixture of target odor, contextual background odor, and novel background odor, the Lasso estimated an odor concentration for each dictionary element. The Lasso estimated a large concentration for the target odor and the contextual background odor that were in the mixture. Because the novel background odors were not part of the dictionary, they were represented by concentrations distributed over multiple dictionary elements. To convert the odor concentration vector produced by the Lasso into a go/no-go readout that could be evaluated as correct or incorrect, we compared the values of the estimated odor concentration vector assigned to the go odors with the values assigned for the no-go odors. If the maximum of the values assigned to the two go odors was larger/smaller than the

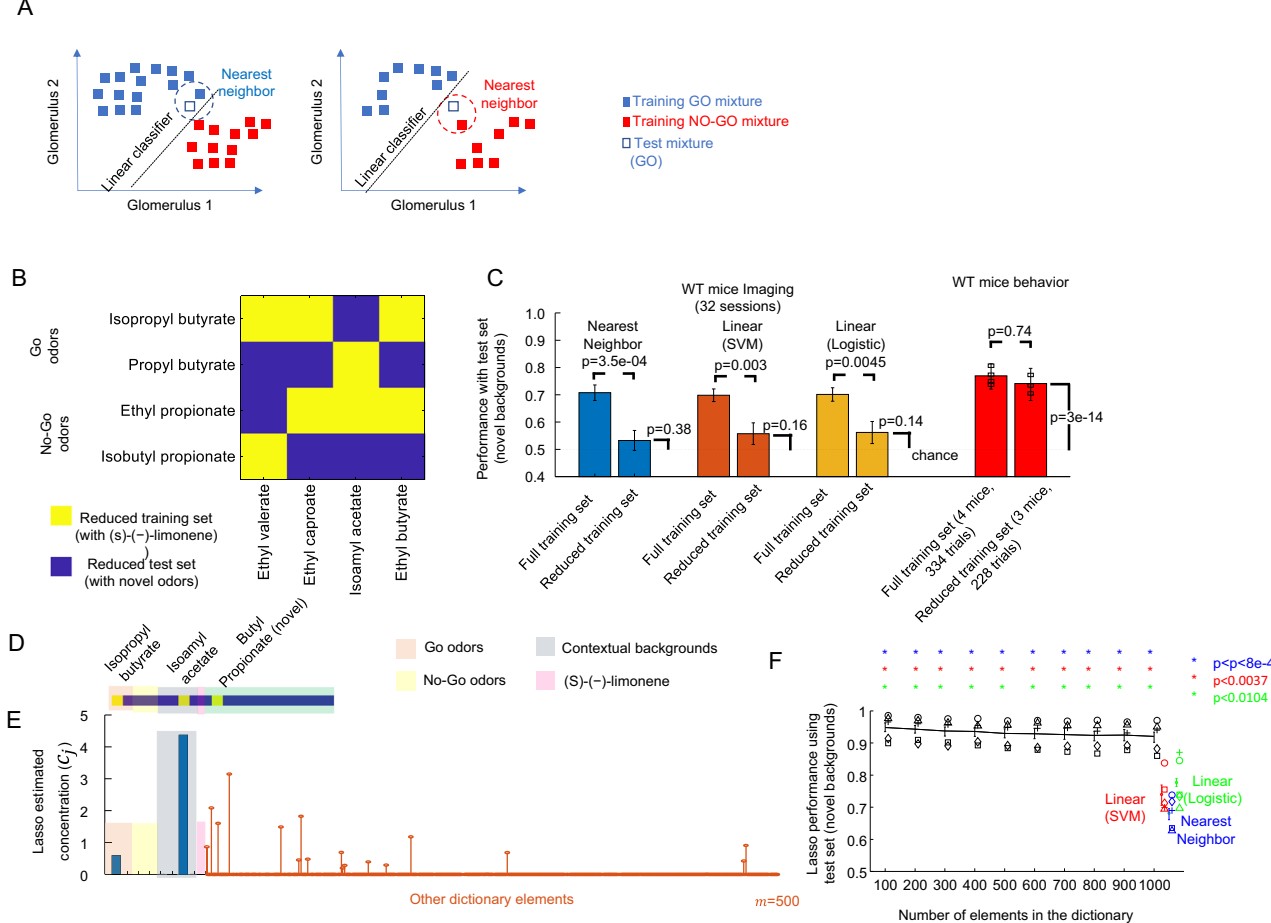

**Fig. 7 | Robust performance of WT mice with reduced training set. A** NNC and a linear classifier can misclassify a test set sample when the number of training mixtures is reduced. **B** Combinations of contextual background odors and target odors for the reduced training and test set. **C** Average performance (±s.e.m.) for linear classifiers and NNC (32 recording sessions 6 WT mice, 161.3 ± 39.8 glom per session). Comparison of algorithm performances using different training sets was done using a two-tailed *t*-test, with *n* = 32 recording sessions. Group performance and 95% confidence intervals for a mouse behavior. The group performance of a new cohort of three WT mice that were also trained with the reduced training set and tested with the reduced test set in the asynchronous task (*n* = 228 trials) was compared to the four WT mice that had trained with the full set and were evaluated with the full set (*n* = 334 trials). The *p*-value was calculated using the Fisher exact test. **D–F** Sparse representation algorithm, Lasso, identifies the odors present in a mixture. **D** Representation of 20 possible odors used to create odor mixtures. The yellow squares mark the odors selected for an example mixture consisting of a go

target odor, a contextual background odor, and a novel background odor. **E** Concentrations assigned by the Lasso. The dictionary included 9 known elements and 491 randomly generated elements. Bars correspond to the estimated concentrations of the 9 known elements. Stems correspond to estimated concentrations for the randomly generated dictionary elements. The maximum of the estimated concentration was larger for the two targets go odors compared to the two target no-go odors. **F** Performance of the Lasso calculated for different sizes of dictionaries was compared to the performances of the NNC, SVM, and logistic regression. Individual symbols represent average performances for the 5 WT mice. Error bars represent the 10-90% percentiles of the 13200 simulations performed per dictionary size for the Lasso and the SVM, logistic regression, and NNC. Significance was calculated using a two-tailed *t*-test and comparing the average performance per animal between the NNC, SVM, and logistic regression against the Lasso performance for different dictionary sizes.

maximum of the values assigned to the two no-go odors, we considered that the Lasso produced a go/no-go readout. If the Lasso readout (go or no-go) matched the label of the target odor present in the input mixture, the Lasso readout was considered correct. The performance of an animal for a given dictionary size was calculated as the average performance over 30 randomly generated dictionaries of each dictionary size.

The average Lasso performance over the 5 WT animals was 93.2 ± 0.9% on the reduced test set (mean ± s.d., 10 dictionary sizes from 100 to 1000 elements, see Fig. 7F). Lasso algorithms performed significantly better than NNC and the linear classifiers when these other algorithms were trained with the reduced training set and tested on the reduced test set. For each imaged WT mice (5 WT mice) and each dictionary size, the Lasso produced significantly better performance than the NNC (68.2 ± 1.4%, mean ± s.e.m., *p* < 8e−4, paired *t*-test, *n* = 5), the linear SVM (73.9 ± 2.5%, *p* < 0.0037) and logistic regression

(77.7 ± 1.1%, *p* < 0.0104). Sparse deconvolution methods outperformed NNC and linear methods and are a potential strategy for WT mice to use for target discrimination in the presence of novel background odors.

## *Cntnap2*$^{-/-}$ mice glomerular responses had higher trial-to-trial variability compared to WT mice

We tested the *Cntnap2*$^{-/-}$ mouse model of autism[42] (JAX Stock No. 017482) because *Cntnap2*$^{-/-}$ mice have either equal[43] or better[44] performance than WT mice in simple olfactory tasks, suggesting a functional olfactory system. Nevertheless, *CNTNAP2* is expressed in olfactory receptor neurons[11] and its absence might reduce neural excitability, affecting glomerular representations. Neurons in the olfactory bulb of *Cntnap2*$^{-/-}$ mice have reduced odor-evoked responses and increased trial-to-trial variability[45]. Therefore, we compared the amplitude and the variability of the odor-evoked responses.

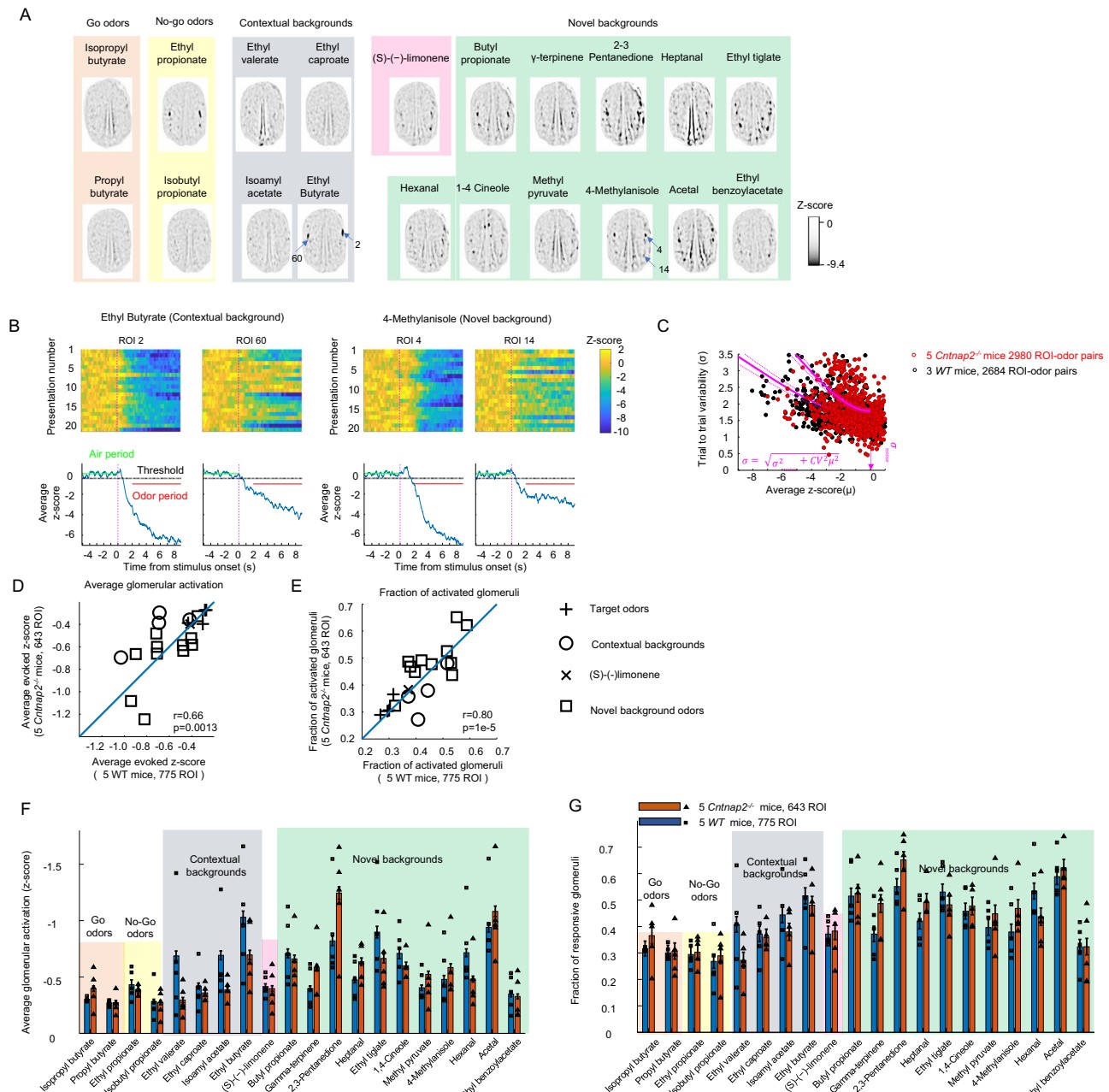

**Fig. 8 | Odor-evoked glomerular responses in *Cntnap2⁻ᐟ⁻* mice were of similar strength as WT mice. A** Average evoked response *z*-score images of a *Cntnap2⁻ᐟ⁻* mice in response to the 20 odors used during the behavior. **B** Example responses as *z*-scores from individual glomeruli of a *Cntnap2⁻ᐟ⁻* mouse on individual odor presentations and as averages. **C** Average response per odor plotted against the standard deviation calculated over trials for 5 *Cntnap2⁻ᐟ⁻* mice (red dots) and 3 WT mice (black dots). The magenta lines are the fitted functions for both genotypes used to estimate the coefficient of variation and imaging noise levels. **D** Average activation of the glomeruli produced by individual odors for the 5 *Cntnap2⁻ᐟ⁻* mice plotted against average activation for 5 WT mice. The value of r corresponds to the Pearson linear correlation and the blue line is the identity line. **E** Fraction of glomeruli activated per odor for the 5 *Cntnap2⁻ᐟ⁻* mice plotted against the fraction of glomeruli activated in 5 WT mice. The value of *r* corresponds to the Pearson linear correlation and the blue line is the identity line. **F** Average activation per odor ± s.e.m. for 5 *Cntnap2⁻ᐟ⁻* mice and 5 WT mice. Each symbol is the average activation of one mouse. **G** Fraction of glomeruli activated per odor with 95% confidence intervals.

We measured intrinsic image responses to the 20 target and background odors at the concentrations used for behavioral measurement in 643 glomeruli recorded from 5 awake *Cntnap2⁻ᐟ⁻* mice with 128.6 ± 19.3 glomeruli (mean ± s.d.) recorded per session (see Fig. 8A for an example). Glomerular activation patterns looked similar to the glomerular activation patterns in WT mice and glomerular responses were observed on individual odor presentations (see Fig. 8B). However, the trial-to-trial variability was higher for *Cntnap2⁻ᐟ⁻* compared to WT mice (see Fig. 8C). The variability in the absence of an odor response ($\sigma_{noise}$) was 1.83 (95% CI: 1.76–1.89) which was slightly higher than $\sigma_{noise}$ in WT (1.59 with 95% CI: 1.51–1.66). Therefore, the threshold for considering a *z*-score response was set at $\sigma_{noise}/\sqrt{n} = 0.46$, that is average ROI-odor responses that were larger than −0.46 were set to zero. *Cntnap2⁻ᐟ⁻* mice also had trial-to-trial variations that were correlated across glomeruli, similar to WT mice. After subtracting that correlated variability, *Cntnap2⁻ᐟ⁻* mice had an uncorrelated variability $CV_{uncorr}$ of 0.44 (95% CI: 0.42–0.47) that was larger than the value of 0.25 (95% CI: 0.23–0.27) of the awake WT mice

(see Supplementary Fig. 12). The higher uncorrelated trial to trial variability in *Cntnap2*$^{-/-}$ mice compared to WT mice could potentially affect *Cntnap2*$^{-/-}$ mice performance in odor identification in novel background odors.

### The amplitude and structure of odor similarities of *Cntnap2*$^{-/-}$ mice were similar to WT mice

We wondered whether the *Cntnap2*$^{-/-}$ mice odor-evoked activity was weaker than WT mice activity. We compared the average activity of the glomerular activation per odor between WT (5 animals, 775 glomeruli) and *Cntnap2*$^{-/-}$ mice (5 animals, 643 glomeruli). *Cntnap2*$^{-/-}$ mice average glomerular responses were not systematically weaker than the WT responses ($p = 0.50$, binomial test, 20 odors, see Fig. 8D, F) nor did the odors activate a smaller fraction of the glomeruli ($p = 0.11$, binomial test, 20 odors, see Fig. 8E, G). In fact, odors that produced strong glomerular activation patterns in the WT mice also produced strong glomerular activation patterns in the *Cntnap2*$^{-/-}$ mice. There was a significant linear correlation ($r = 0.66$, $p = 0.0013$, Pearson linear coefficient) between the average glomerular response for a given odor between WT mice and Cntnap2$^{-/-}$ mice. The fraction of glomeruli that

responded to a given odor and exceeded the detection threshold (z-score = −0.46 for *Cntnap2*$^{-/-}$ mice and z-score = −0.42 for WT mice) was also linearly correlated between WT mice and Cntnap2$^{-/-}$ mice ($r = 0.80$, $p = 1e{-}5$). The average glomerular activity produced by an individual odor in the WT mice was not significantly different from the activity in the *Cntnap2*$^{-/-}$ mice ($p > 0.13$, for all 20 odors, *t*-test). The fraction of glomeruli that responded to a given odor was also not significantly different between the Cntnap2$^{-/-}$ mice and the WT mice ($p > 0.11$, for all 20 odors, *t*-test). Thus, *Cntnap2*$^{-/-}$ mice glomerular responses were not systematically weaker than WT mice glomerular responses.

Odor pairs whose glomerular activation patterns were similar in the WT mice were also similar in the *Cntnap2*$^{-/-}$ mice and odor pairs that had different glomerular activation patterns in the WT mice were also different in the *Cntnap2*$^{-/-}$ mice. We quantified the similarity between pairs of glomerular patterns as the normalized dot product between average z-scores produced by an odor. We used all the glomeruli recorded for each genotype to create a large vector to calculate the normalized dot product (775 glomeruli from 5 WT mice, 643 glomeruli from 5 *Cntnap2*$^{-/-}$ mice, see Fig. 9A). The matrix of odor

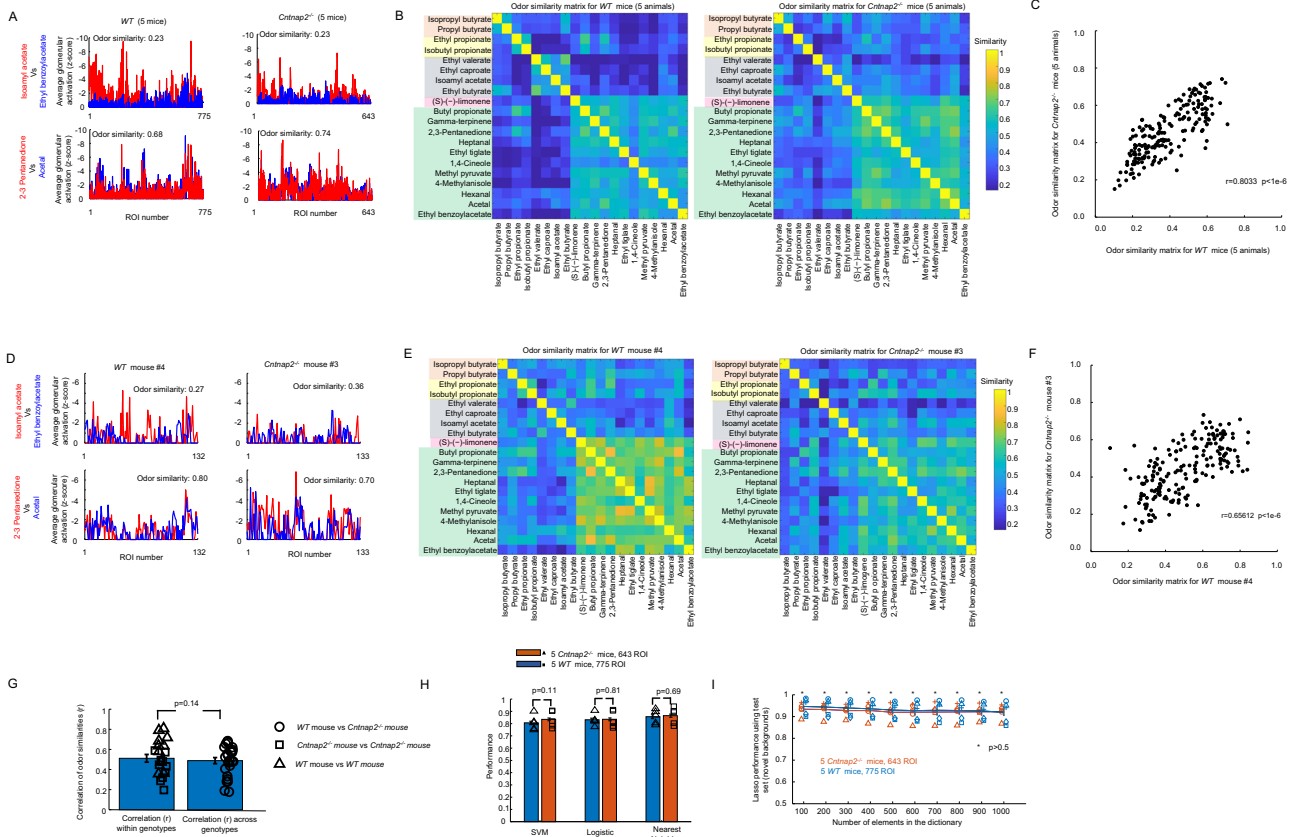

**Fig. 9 | Pairs of odors that evoked similar activity patterns in WT mice also evoked similar patterns in *Cntnap2*$^{-/-}$ mice. A** Patterns of glomerular activation using all available ROI per genotype for 4 example odors. Isoamyl acetate and ethyl benzoylacetate produced different patterns of glomerular activation in WT and *Cntnap2*$^{-/-}$ mice whereas 2–3 pentanedione and acetal produced similar glomerular activation patterns in both WT and *Cntnap2*$^{-/-}$ mice. **B** Odor similarity matrices for *Cntnap2*$^{-/-}$ mice and WT mice were calculated using all available ROI per genotype. **C** Similarity between 190 odor pairs calculated using all the WT mice glomerular responses versus similarity between odors calculated using all the *Cntnap2*$^{-/-}$ mice glomerular responses. **D** Patterns of glomerular activation using ROI from example individual animals of each genotype. **E** Odor similarity matrices for the example *Cntnap2*$^{-/-}$ mouse and WT mouse. **F** Similarity between 190 odor pairs calculated using the example WT mouse data against the similarity from the example

*Cntnap2*$^{-/-}$ mouse. **G** Distribution of linear correlation coefficients of odor similarities ($r$) for pairs of animals of the same genotype (WT vs. WT, 10 animal pairs and *Cntnap2*$^{-/-}$ vs. *Cntnap2*$^{-/-}$ 10 animal pairs) and for pairs of animals of different genotypes (WT vs. *Cntnap2*$^{-/-}$, 25 animal pairs). Error bars represent the mean ± s.e.m. Comparison between correlation coefficients was done using a two-tailed *t*-test. **H** Performance (mean ± s.e.m.) of SVM, logistic, and NNC classifiers calculated using *Cntnap2*$^{-/-}$ and WT mice glomerular activation data for target detection in novel environments trained with the full training set and tested with the full training set. Symbols represent average performance per animal. Performances were compared using a two-tailed *t*-test, with $n = 50$ mice-odor pairs per genotype. **I** Performance (mean ± s.e.m.) of the Lasso using *Cntnap2*$^{-/-}$ and WT mice glomerular data for the reduced test set using different sizes of dictionaries. Significance was calculated using a two-tailed *t*-test, with $n = 5$ mice per genotype.

similarities was comparable across the two phenotypes (see Fig. 9B). We calculated the Pearson linear correlation between odor similarities for the 190 odor pairs in WT mice and *Cntnap2*$^{-/-}$ mice. There was a strong correlation ($r = 0.80$, $p < 1e-6$, see Fig. 9C) between the odor similarities in WT mice and odor similarities in *Cntnap2*$^{-/-}$ mice. The similarities among the standard background odors considered separately (four contextual backgrounds and (s)-(−)-limonene) were also significantly correlated across genotypes ($r = 0.71$, $p = 0.02$, $n = 10$ similarity comparisons) as well as the similarities among the 11 novel background odors ($r = 0.36$, $p = 0.007$, $n = 55$ similarity comparisons).

The linear correlation between odor similarity patterns was also significant when comparing glomerular activation patterns from individual animals across genotypes. We compared the odor similarities pattern in 5 WT mice and 5 *Cntnap2*$^{-/-}$ mice, resulting in 25 comparisons across the two genotypes (see Fig. 9D–F for an example of odor similarity correlation between a WT and a *Cntnap2*$^{-/-}$ mouse). All 25 comparison's across genotypes produced positive significant linear correlations ($p < 0.014$, see Fig. 9G). The average odor similarity correlation across the two genotypes was $0.49 \pm 0.03$ (mean ± s.e.m., $n = 25$ pairs of WT-*Cntnap2*$^{-/-}$ mice) which was not significantly different ($p = 0.14$, $t$-test) to the average odor similarity correlation within the same genotype ($0.51 \pm 0.04$, mean ± s.e.m., 10 pairs of WT−WT comparisons and 10 pairs of *Cntnap2*$^{-/-}$–*Cntnap2*$^{-/-}$ comparisons). Average glomerular activation patterns in *Cntnap2*$^{-/-}$ mice were not significantly suppressed compared to WT mice and odors produced similar average patterns in *Cntnap2*$^{-/-}$ and WT mice.

### Increased trial-to-trial variability in glomerular activity in *Cntnap2*$^{-/-}$ did not significantly reduce performance of algorithms

Although average glomerular responses in the *Cntnap2*$^{-/-}$ and WT mice were similar, algorithms performing odor detection in novel environments could still be affected by the increased trial-to-trial variability in *Cntnap2*$^{-/-}$ mice ($CV_{uncorr} = 0.44$) compared to WT mice ($CV_{uncorr} = 0.25$). We created artificial mixtures using the imaging data from *Cntnap2*$^{-/-}$ mice (5 mice, 643 ROI) and simulated the increased variability using a coefficient of variation of $CV_{uncorr} = 0.44$ and the threshold of glomerular activation of *Cntnap2*$^{-/-}$ mice of −0.46. We trained the SVM, logistic regression, and NNC using the full set of training examples (16 mixtures) and tested them on the 160 mixtures that included 10 novel background odors. We compared the *Cntnap2*$^{-/-}$ mice performance of the classifiers with the performance obtained from the data from WT mice (5 mice, 775 ROI). We performed the analysis on each individual animal and pooled the average performance for each odor resulting in 50 performances (5 animals*10 novel background odors= 50 animal−odor pairs). The performance of the algorithms using *Cntnap2*$^{-/-}$ mice data was not significantly different from the algorithms that used *WT* mice data (see Fig. 9H). Using *Cntnap2*$^{-/-}$ mice glomerular imaging data, the performance of the SVM was $83.5 \pm 1.1\%$ (mean ± s.e.m.) which was not significantly different from WT mice glomerular imaging data ($81.1 \pm 1.2\%$, mean ± s.e.m., $p = 0.11$, $t$-test). Using *Cntnap2*$^{-/-}$ glomerular imaging data, the performance of the logistic regression was $84.2 \pm 1.3\%$ which was not significantly different from the performance calculated using WT mice glomerular imaging data ($83.0 \pm 1.2\%$, mean ± s.e.m., $p = 0.81$, $t$-test). The performance of the NNC using *Cntnap2*$^{-/-}$ glomerular imaging data was $86.2 \pm 1.1\%$ (mean ± s.e.m.) which was not significantly different from the WT glomerular imaging data ($85.3 \pm 2.2\%$, mean ± s.e.m., $p = 0.69$, $t$-test). We also tested the performance of the algorithms using our dataset of real odor mixtures collected from 6 WT mice (32 recording sessions, 5163 ROI), but simulated with the larger trial-to-trial odor-evoked variability ($CV_{uncorr} = 0.44$) measured in the *Cntnap2*$^{-/-}$ glomerular imaging data (see Supplementary Note 5: Increased trial-to-trial variability did not affect NNC and linear

classifiers and Supplementary Fig. 14A−C). We also used the threshold of glomerular activation of −0.46 that we identified for the *Cntnap2*$^{-/-}$ mice. There was also no significant difference ($p > 0.4$, $t$-test, $n = 32$ recording sessions) in the performance of the high-variability regime (*Cntnap2*$^{-/-}$ like) compared to the reduced variability regime (WT-like). The performance of the algorithms on odor identification in novel environments was only significantly affected by larger values of $CV_{uncorr}$ ($CV_{uncorr} > 0.85$ for SVM and logistic and $CV_{uncorr} > 1.05$ for NNC, see Supplementary Fig. 14D−F) compared to the variability in *Cntnap2*$^{-/-}$ mice ($CV_{uncorr} = 0.44$).

WT mice performance was unaffected by reductions in the diversity of training examples which is more consistent with WT mice performing a sparse deconvolution algorithm like the Lasso than the NNC or linear classifiers. We wondered whether the Lasso performance on the reduced test set could be affected when using the glomerular activation patterns of *Cntnap2*$^{-/-}$ mice. We simulated the Lasso as described above using the glomerular data from the 5 *Cntnap2*$^{-/-}$ mice with their increased trial-to-trial variability ($CV_{uncorr} = 0.44$). The performance of the Lasso using *Cntnap2*$^{-/-}$ mice imaging data in the reduced test set was $92.6 \pm 0.5\%$ (mean ± s.d., 10 tested sizes of dictionary between 100 and 1000 elements) and it was very similar to the performance calculated for the WT mice ($93.2 \pm 0.9\%$, see Fig. 9I). For each dictionary size, we compared the performance of the Lasso using the 5 WT mice imaging data with the 5 *Cntnap2*$^{-/-}$ mice. The performances were not significantly different for any of the dictionary sizes tested ($p > 0.5$, $t$-test).

The similarity in magnitude and shape of the odor-evoked glomerular responses between *Cntnap2*$^{-/-}$ and WT mice suggests similar behavioral performance might be observed in odor detection in novel environments. Although odor-evoked responses are more variable in *Cntnap2*$^{-/-}$ mice, the performance of the NNC, linear classifiers, and a sparse deconvolution algorithm were similar to WT mice.

### Odor detection in *Cntnap2*$^{-/-}$ mouse model of autism was selectively affected in novel background odors

We trained four *Cntnap2*$^{-/-}$ mice using the restricted training set with the asynchronous presentation of the odors. The *Cntnap2*$^{-/-}$ mice performance on the standard background odors, just before the presentation of the novel odors started, was 85.5% (220 trials, 4 *Cntnap2*$^{-/-}$ mice) and it was not significantly different ($p = 0.75$, Fisher exact test) from the performance of the WT mice that also trained with the restricted training set (87.1%, 140 trials, 3 WT mice, see Fig. 10A). However, when tested with the restricted test set with novel background odors, the *Cntnap2*$^{-/-}$ mice performance dropped to 61.1% (283 trials, see Fig. 10B), which was significantly lower ($p = 0.002$, Fisher exact test) than the performance of WT mice (74.1%, 228 trials). Although glomerular activation patterns and odor target recognition in known background odors were very similar between *Cntnap2*$^{-/-}$ and WT mice, odor identification performance in novel background odors was lower in *Cntnap2*$^{-/-}$ mice.

### Training with background odor reduced target detection deficits in *Cntnap2*$^{-/-}$ mice

Although previous work in WT mice has shown that the difficulty of target identification in the background was determined by the glomerular representation[7], we hypothesized that the novelty of the background was an additional crucial factor that affected the capability of *Cntnap2*$^{-/-}$ mice to detect the target odors. Therefore, removing the novelty of a background odor should produce a significant improvement in target identification for a background odor that *Cntnap2*$^{-/-}$ mice struggled with when it was a novel background odor. To test this hypothesis, we trained a new group of two *Cntnap2*$^{-/-}$ mice and three WT mice but instead of using the original training set that included (s)-(−)-limonene, the mice trained with 8 odor mixtures from the reduced test set that included butyl propionate, which was

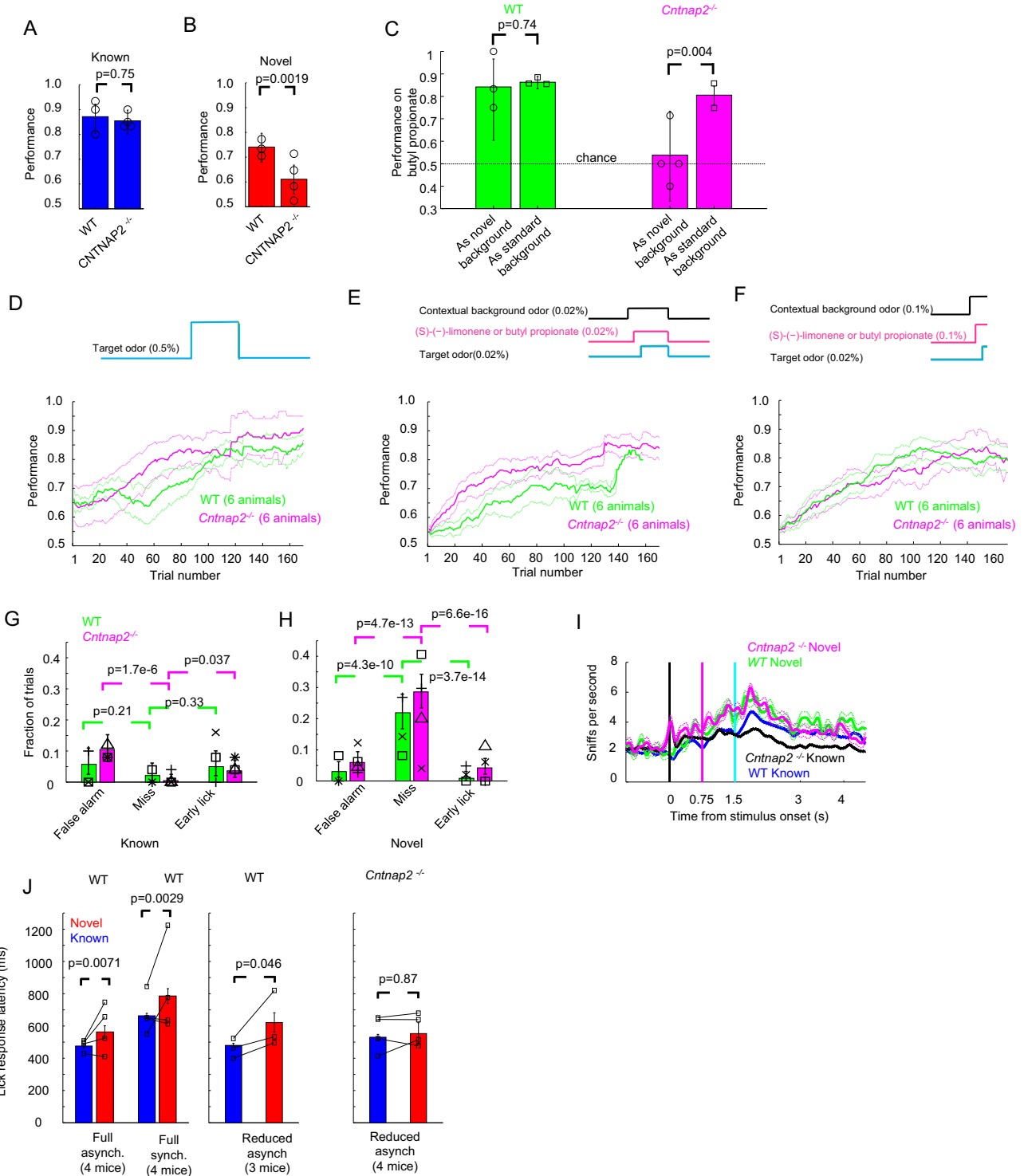

one of the original 11 novel background odors. Glomerular responses for butyl propionate were similar between *Cntnap2*⁻/⁻ mice and WT mice (see Supplementary Fig. 15). The training of the *Cntnap2*⁻/⁻ mice and WT mice proceeded until they performed at least 150 trials at >80% performance at the second to last background concentration of 0.038%, which took a similar amount of training days (*Cntnap2*⁻/⁻ mice, 15 ± 1.4 sessions, mean ± s.d.; WT mice,15.6 ± 1.5 sessions, mean ± s.d.). Performance was evaluated at the final background concentration of 0.1% which is the concentration used when testing novel background odors.

Olfactory training substantially increased the performance on butyl propionate background in *Cntnap2*⁻/⁻ mice. The performance of the *Cntnap2*⁻/⁻ mice when butyl propionate was novel was 53.8% (26 trials, 4 *Cntnap2*⁻/⁻ mice). However, when butyl propionate was used during training, the performance significantly increased to 80.5% (365 trials, 2 *Cntnap2*⁻/⁻ mice, *p* = 0.004, Fisher exact test, see Fig. 10C). On the other hand, the olfactory training using butyl propionate did not produce a significant increase in the performance of the WT mice. The performance of the WT mice when butyl propionate was used as a novel background odor was 84.2% (19 trials, 3 mice). However, when butyl propionate was used during training, the performance only increased to 86.3% (614 trials, 3 mice). This increase was not significant (*p* = 0.74, Fisher exact test). Our results demonstrate that *Cntnap2*⁻/⁻ mice deficits were not determined exclusively by

**Fig. 10 | *Cntnap2*<sup>−/−</sup> mice discrimination in novel backgrounds was selectively affected. A** Performance and 95% confidence interval for four *Cntnap2*<sup>−/−</sup> mice (*n* = 220 trials) and three WT mice (*n* = 140 trials) on known backgrounds for the reduced training set. Symbols indicate individual animal performance. Significance was calculated using a two-tailed Fisher exact test. **B** Performance in the novel background for the reduced test set. *p*-values were calculated with a two-tailed Fisher exact test. **C** Performance and 95% confidence intervals for butyl propionate when it was used as a novel and as a standard background for different mice cohorts. *p*-values were calculated using the Fisher exact test. **D–F** Odor learning curves were not affected in *Cntnap2*<sup>−/−</sup> mice compared to WT mice. Solid lines represent the mean response calculated using a 60-trial sliding window and dotted lines represent the mean±s.e.m calculated for 6 *Cntnap2*<sup>−/−</sup> and 6 WT mice. **D** Learning curves for the second pair of go target and no-go target at a high concentration (0.1%). **E** Learning curves for the first exposure of the standard background odors at 0.025% (1/5 of the final concentration) with the targets at their final concentration (0.025%). **F** Learning curves for the first exposure of the standard background odors at the final concentration of the background odors (0.1%). **G** Fraction of types of errors for WT and *Cntnap2*<sup>−/−</sup> mice for standard background odors for four *Cntnap2*<sup>−/−</sup> mice (*n* = 220 trials) and three WT mice (*n* = 140 trials). Error bars are the 95% confidence intervals. *p*-values were calculated using a two-tailed Fisher exact test. **H** Fraction of trials of types of errors with the 95% confidence interval for WT and *Cntnap2*<sup>−/−</sup> mice for novel background odors. **I** Sniff responses for *Cntnap2*<sup>−/−</sup> and WT mice for novel and standard background odors. Lines represent mean ± s.e.m. **J** WT mice lick response times were slower in the presence of novel background odors compared to the known background odors in both the asynchronous and the synchronous tasks. Error bars represent the s.e.m. *p*-values were calculated using a two-tailed Wilcoxon rank-sum test.

the glomerular responses but were affected by the novelty of the background odor.

### *Cntnap2*<sup>−/−</sup> mice odor learning rate was similar to WT mice

Although the training protocol for the *Cntnap2*<sup>−/−</sup> mice was similar to the WT mice, we wondered whether their olfactory learning rates were slower as shown for spatial memory tasks[46]. We were also expecting slower learning in *Cntnap2*<sup>−/−</sup> mice because, at different stages of the training process, both the target odor and the background odors were novel to the animal. Both WT mice and *Cntnap2*<sup>−/−</sup> mice successfully learned to discriminate between isobutyl propionate (0.34%, no-go stimulus) and isopropyl butyrate (0.34%, go-stimulus) on the first day. This learning includes olfactory discrimination but also non-olfactory related behaviors like learning to lick the water tube as well as the structure of the task. To directly measure the odor learning capability, we analyzed the second session (day 3), when the mice learned to discriminate between ethyl propionate (0.34%, no-go stimulus) and propyl butyrate (0.34%, go-stimulus) after having already learned the motor aspects of the task. Similar performance was reached for both genotypes after 100 trials (87.6% for 6 *Cntnap2*<sup>−/−</sup> mice with 211 trials, 82.8% for 6 WT, 239 trials, *p* = 0.18, Fisher exact test, trials between 100 and 170, see Fig. 10D). We also compared the learning rate between both genotypes using the normalized integration value which has higher values for faster learning[47]. There was no difference in normalized integration value (*Cntnap2*<sup>−/−</sup>: 0.78 ± 0.05, WT: 0.72 ± 0.03, *p* = 0.35, *t*-test). Thus, the *Cntnap2*<sup>−/−</sup> mice learning rates for the target odors were similar to the WT mice.

Interestingly, when target odors were presented for the first time together with low-concentration background odors (0.025% of vapor saturation), *Cntnap2*<sup>−/−</sup> mice quickly learned to identify the no-go target odors (isobutyl propionate and ethyl propionate, 0.025% of saturated vapor) from the go target odors (isopropyl butyrate and propyl butyrate, 0.025% of saturated vapor). In fact, *Cntnap2*<sup>−/−</sup> mice performance plateaued after 100 trials at a significantly higher level than the WT mice (*Cntnap2*<sup>−/−</sup>:83.3%, 301 trials; WT: 74.0%, 204 trials, *p* = 0.0131, Fisher exact test, see Fig. 10E). *Cntnap2*<sup>−/−</sup> mice were not slower than WT mice in learning to identify target odors in the presence of background odors as quantified by the normalized integration value (*Cntnap2*<sup>−/−</sup>: 0.75 ± 0.02, mean ± s.e.m., *n* = 6 animals, WT: 0.66 ± 0.03, *n* = 6 animals, *p* = 0.276, *t*-test).

We also analyzed the learning rates when mice were first exposed to the training backgrounds at the final concentration (0.1% vapor pressure). Performance of both the *Cntnap2*<sup>−/−</sup> mice (6 animals) and WT mice (6 animals) reached a plateau after 100 trials. The performance in the plateau phase (trials between 100 and 170, see Fig. 10F) was similar between both genotypes (87.6% for *Cntnap2*<sup>−/−</sup> mice with 211 trials, 79.4% for WT, 355 trials, *p* > 0.9, Fisher exact test). There was no difference in normalized integration value (*Cntnap2*<sup>−/−</sup>: 0.71 ± 0.03, mean ± s.e.m., WT: 0.73 ± 0.03, *p* = 0.85, *t*-test). *Cntnap2*<sup>−/−</sup> mice olfactory deficits with novel background odor were only evident when

novel background odors were presented as catch trials and not during training.

### *Cntnap2*<sup>−/−</sup> mice errors in novel environments were not caused by excessive licking

The *Cntnap2*<sup>−/−</sup> mice low performance might be caused by their hyperactivity[44]. Hyperactivity could result in uncontrolled licking, increasing the fraction of false alarms and/or early licks. For the known background odors in *Cntnap2*<sup>−/−</sup> mice, performance was high (85.5%, see Fig. 10G). Most of the errors were indeed caused by excessive licking consistent with *Cntnap2*<sup>−/−</sup> being hyperactive. There were significantly more false alarms(10.4% of 220 trials, *p* = 1.7e-6, Fisher exact test) and early licks(3.6%, *p* = 0.037, Fisher exact test) compared to misses, which were quite rare (0.4%). In the WT mice, misses were also the least frequent type of error (2.1% of 140 trials) compared to false alarms (5.7%, *p* = 0.21, Fisher exact test) and early licks (5.0%, *p* = 0.33, Fisher exact test). There was no significant difference in the types of error for standard background odors between the phenotypes (*p* > 0.2, Fisher exact test). *Cntnap2*<sup>−/−</sup> mice performance on known backgrounds may be affected by hyperactivity to a small degree.

In contrast, the main type of error made by *Cntnap2*<sup>−/−</sup> mice in novel background odors was misses (28.6% of 283 trials, see Fig. 10H) and this fraction was significantly larger than false alarms (6.0%, *p* = 4.7e−13, Fisher exact test) and early licks (4.2%, *p* = 6.6e−16); that is, most of the errors in novel backgrounds in *Cntnap2*<sup>−/−</sup> mice were not caused by excessive licking but by not licking to the go stimulus. WT mice also had a significantly larger number of misses (21.9% of 228 trials) compared to false alarms (3.1%, *p* = 4.3e−10) and early licks (0.8%, *p* = 3.7e−14, Fisher exact test). Increased misses by *Cntnap2*<sup>−/−</sup> mice in novel background odors are not consistent with being caused by *Cntnap2*<sup>−/−</sup> mice hyperactivity.

### *Cntnap2*<sup>−/−</sup> mice did not increase their sniff rate for target odors

Children with autism have lower sniff modulation in response to odors[48] and this lack of sniffing modulation might be affecting *Cntnap2*<sup>−/−</sup> mice performance. WT mice that trained with the full training set increased their sniff rate in response to target odors (see Fig. 5E and Fig. 6D). In contrast, *Cntnap2*<sup>−/−</sup> mice had a smaller elevation of their sniff rate in response to the target odors in the presence of known background odors (see Fig. 10I). We quantified the sniff rate in response to the target in the 750 ms interval following target odor onset. *Cntnap2*<sup>−/−</sup> had 3.29 ± 0.06 sniffs per second (mean ± s.e.m.) in response to the target (1046 target presentations, four *Cntnap2*<sup>−/−</sup> mice) and it was lower than the sniff rate of the WT mice that trained with the same reduced training set (4.06 ± 0.06 sniffs per second, 957 target presentations, three WT mice, *p* = 5.11e−16, Wilcoxon ranksum test). The lack of elevation of the sniff rate in response to the target odors in *Cntnap2*<sup>−/−</sup> mice also resulted in fewer sniffs in response to the target before a lick response. We counted the number of sniffs in the time interval between the first inhalation following the target odor

presentation and the first lick response (both hits and false alarms). *Cntnap2*⁻/⁻ mice took 2.3 ± 0.1 sniffs (mean ± s.e.m., *n* = 462 lick responses, 4 mice) whereas WT mice took 2.5 ± 0.1 sniffs (mean ± s.e.m., *n* = 496 lick responses, 3 mice, *p* = 0.008, Wilcoxon rank-sum test).

### *Cntnap2*⁻/⁻ mice increase in sniffing rate for novel background odors was similar to WT mice

Although *Cntnap2*⁻/⁻ mice did not modulate their sniffing responses to target odors, *Cntnap2*⁻/⁻ mice did increase their sniff rate in response to the first presentation of the novel background odors. This increased sniff response was similar to the three WT mice that performed the same reduced test set task and similar to the WT mice that performed the asynchronous (see Fig. 5E) and synchronous tasks (see Fig. 6D) with the full test set. The sniff rate in the 750 ms following the onset of the novel background odor but preceding the onset of the target was 4.35 ± 0.23 sniffs per second (77 responses, 4 *Cntnap2*⁻/⁻ mice) and it was not significantly different from the sniff rate for the WT mice (4.06 ± 0.25 sniffs per second, 66 responses, 3 WT mice, p = 0.40, Wilcoxon ranksum test). The sniff rate after the onset of the target odor was 5.35 ± 0.23 sniffs per second (77 responses, 4 *Cntnap2*⁻/⁻ mice) and it was also not significantly different from the sniff rate for the WT mice (5.21 ± 0.27 sniffs per second, 66 responses, 3 WT mice, *p* = 0.58, Wilcoxon ranksum test). The baseline sniffing before odor onset was similar between *Cntnap2*⁻/⁻ mice and WT mice (*Cntnap2*⁻/⁻ 2.29 ± 0.15 sniffs per second, 77 trials, WT 2.16 ± 0.18 sniffs per second, 66 trials, *p* = 0.26 Wilcoxon ranksum test). *Cntnap2*⁻/⁻ mice did detect and explore the novel background odors. However, an increase in sniffing in response to the novel background odors was not sufficient for *Cntnap2*⁻/⁻ mice to match the performance of the WT mice.

### *Cntnap2*⁻/⁻ mice did not increase lick response latency for novel background odors

Although WT mice increase their target response latencies by 70–100 ms when challenged with difficult odor discrimination tasks[24], WT mice have much shorter target response latency increases of just 16 ms when sampling mixtures of increased difficulty for odor detection in known background odors[7]. We wondered if Cntnap2⁻/⁻ mice might also have only small increases in their target response latency when challenged with the increased difficulty of identifying odors in novel backgrounds. We quantified the target response latency as the time interval between the first inhalation following the target odor presentation and the first lick response (both hits and false alarms). Although *Cntnap2*⁻/⁻ mice increased their sniff rates, they did not increase their target response latency in the presence of novel background odors. *Cntnap2*⁻/⁻ mice target response latency with novel background odors was 553.0 ± 67.1 ms (*n* = 49 lick responses, 4 *Cntnap2*⁻/⁻ mice, see Fig. 10J) and it was not significantly slower than the latency with known background odors (530.3 ± 17.2 ms, *n* = 462 lick responses, *p* = 0.87, Wilcoxon rank-sum test). In contrast, WT mice that performed the same asynchronous reduced training set task, increased their latency when they were challenged with novel background odors. Target response latency increased from 479.5 ± 12.4 ms (*n* = 496 reactions, 3 WT mice) in standard backgrounds to 621.5 ± 60.2 ms in novel backgrounds (*n* = 67 lick responses, 3 WT mice, *p* = 0.046, Wilcoxon rank-sum test). *Cntnap2*⁻/⁻ mice had intermediate target response latencies that were not different from WT mice for known odors and novel background odors (p = 0.40 between known background odors and *p* = 0.25 between novel odors, Wilcoxon rank-sum test). *Cntnap2*⁻/⁻ mice took 2.4 ± 0.2 sniffs before licking in response to the targets on the novel background odors (*n* = 49 lick responses, 4 *Cntnap2*⁻/⁻ mice) and WT mice took a similar number of sniffs (2.6 ± 0.2, *n* = 67 lick responses, 3 WT mice, *p* = 0.65, Wilcoxon rank-sum test).

Increased target response latencies with novel background odors were also present in WT mice that trained with the full training set.

The target response latency for the asynchronous case for novel background odors was 563.3 ± 38.0 ms and it was significantly slower (*n* = 146 lick responses, 4 WT mice, *p* = 0.007, Wilcoxon rank-sum test) than the response for the known background odors (475.8 ± 11.6 ms, 800 lick responses). The latency for novel background odors for the synchronous case was 785.5 ± 45.6 ms and it was also significantly slower (109 lick responses, 4 WT mice, *p* = 0.0029 Wilcoxon rank-sum test) than the latency for known background odors (662.8 ± 15.7 ms, *n* = 577 trials).

*Cntnap2*⁻/⁻ mice produced a sniffing response to the novel odors that was different from their response to the standard background odors and similar to the sniffing response of WT mice to novel background odors. *Cntnap2*⁻/⁻ mice reacted to the presence of novel background odors but they did not delay their responses for novel background odors as WT mice did (see Fig. 10I). Sniffing reaction to the presence of a novel background odor similar to WT mice was not sufficient for the *Cntnap2*⁻/⁻ mice to match WT mice target detection in novel background odors.

## Discussion

We have developed an olfactory CAPTCHA for mice that revealed the odor generalization capabilities of WT mice and the limitations of the *Cntnap2*⁻/⁻ mouse model of autism. WT mice could identify odors in novel environments, even on the first presentation. Further exposure to an initially novel background or having the novel background precede the target onset did not significantly improve WT mice performance. By using intrinsic images of the dorsal glomeruli in WT mice, we determined that responses from individual glomeruli could not generalize to novel background odors. WT mice performance was unaffected by a reduction in the diversity of training mixtures, whereas the NNC and the linear classifiers' performance fell to chance levels. *Cntnap2*⁻/⁻ mice glomerular activation patterns, were similar to WT mice, and *Cntnap2*⁻/⁻ mice learning rate and performance for known backgrounds were indistinguishable from WT. However, *Cntnap2*⁻/⁻ mice performance for novel backgrounds was significantly lower than that of WT mice. *Cntnap2*⁻/⁻ mice performance on a novel background odor could be rescued by pre-training, matching the performance of WT mice.

WT mice increased their sniffing responses as the target odor was presented with known backgrounds. This sniffing response was absent in *Cntnap2*⁻/⁻ mice and might constitute an additional mechanism that contributes to odor detection in the presence of background odors, which have been proposed as suppressing the responses to preceding novel background odor[30]. However, sniff increases only occurred for the first presentations of a given novel background odor. WT mice are able to solve the task on later appearances of the novel background where the sniff rates had returned to the levels seen for known background odors.

Mice are sensitive to fast differences in temporal profile between odor streams which could be used to separate odor sources[49]. We have not explored the role of these fluctuations extensively for target identification in novel environments. The only temporal difference tested was a 750 ms onset difference between the novel background odor and the target in the asynchronous task. This temporal difference did not contribute significantly to target detection in our task because the performance obtained on the synchronous task was almost identical to the performance on the asynchronous task. This stands in contrast with invertebrate experimental data: honeybees required 5 s or longer to segregate novel odors, which is two orders of magnitude longer than the time required for known odors[35]. Although *Cntnap2*⁻/⁻ mice did increase their sniff rate to novel background odors in a pattern similar to WT mice, this elevated sniff rate was not sufficient to produce a performance that matched WT mice in novel background odors.

A limitation of our imaging data is that we can only sample glomeruli from the dorsal surface of the olfactory bulb. We also averaged

glomerular responses over a 7 s window, an order of magnitude slower than the animal reaction time, which ignores the rich temporal dynamics exhibited by the glomeruli that might also contribute to odor identification[50]. Including glomerular activity from non-optical accessible ventral glomeruli might permit generalization to novel background odors using linear classifiers or NNC. However, the performance of these algorithms reached a plateau using 24–36 glomeruli. This is consistent with a glomerular representation of odors being highly redundant as revealed by lesion studies[51]. Our imaging data were from awake naïve animals that were passively exposed to individual odors and odor mixtures.

Performance using single glomeruli activation on known background odors was similar to WT and *Cntnap2*[-/-] mice behavioral performance. However, activation of individual glomeruli is not sufficient to trigger suprathreshold neural responses[52] in target areas and might require plastic changes to enhance the activation of downstream areas. Potential plasticity mechanisms include experimentally observed changes in receptor expression produced by odor experience[53] and presynaptic modulation of the olfactory receptor neurons themselves[54–56]. In addition, mitral/tufted cell activity mostly accesses activity from an individual glomerulus[21,57] and their modulation could also increase the effect of individual glomeruli activation in target areas. Neural responses in mitral/tufted cells do increase towards the rewarded odor[58,59]. There is also increased frequency coupling across bands for odors associated with reward[60] in the olfactory bulb. Mitral/tufted cell responses could also be quickly modulated to increase their response to target odors[61]. However, individual glomeruli did not generalize well to novel background odors with a performance of 53.0%, which was much lower than the performance of WT mice (asynchronous 76.9%, synchronous 77.2%) but close to *Cntnap2*[-/-] mice performance (61.1%). Our target concentrations were in the micromolar range and there was a high overlap between the target and the background. Strategies that rely on individual glomeruli might be more effective at nano or picomolar concentrations where glomerular activation is sparser[62].

Although there has been interest in computational neuroscience on potential deconvolution methods used by the mammalian olfactory system with odor mixtures[37–39], so far experimental results[19] showed that linear classifiers represented an upper limit for target recognition in mixtures[7]. In previous work[19] WT mice had a training set consisting of individual odors and were presented with intermixed catch-trials with mixtures of the same odors. WT mice performance on the mixtures was lower than the performance of a linear classifier and was not consistent with the use of a deconvolution method like the Lasso. In contrast, both our training set and catch trials used mixtures, with catch-trials consisting of mixtures with novel background odors and novel combinations of known background and target odor. By having the mice being previously trained with odor mixtures and not only with individual odors, we were able to unveil WT mice strategies that extend beyond the limits imposed by linear classifiers and the NNC, as well as the limitations of *Cntnap2*[-/-] mice in target detection.

*Cntnap2*[-/-] mice average glomerular activations produced by odors were very similar to the activations seen in WT mice. This is consistent with a previous report of *Cntnap2*[-/-] mice with intact olfactory function, which outperformed WT mice on the buried food task[44]. However, the trial-to-trial variability of glomerular responses was higher in *Cntnap2*[-/-] mice compared to WT mice. This increased variability was not large enough to affect the performance of linear classifiers, the NNC, or Lasso deconvolution for odor identification in novel background odors in our task. However, the higher variability might affect the performance of other types of odor classification where mice need to detect small differences between very similar odor mixtures[23].

Linear classifiers and NNC performance plateaued when using 24–36 selected glomeruli. However, these algorithms required at least 100 glomeruli to predict the performance of WT mice on individual novel background odors. By using a larger number of glomeruli, WT mice might mitigate the effect of noisy glomerular responses. WT mice responses to novel background odors were delayed by 100–120 ms compared to responses to known background odors. This increase in response time for novel background odors is consistent with the increased difficulty of the task[63] which might require longer temporal integration. This delay was not present in the *Cntnap2*[-/-] mice and might have contributed to the lower performance of *Cntnap2*[-/-] mice in novel environments.

The extra delay in WT mice could also reflect the recruitment of circuits that perform a deconvolution algorithm in the presence of novel background odors. The additional mitral cell activity produced by novel odors[6] might recruit the piriform cortex that could implement recurrent computations required for identifying odors in novel backgrounds using its association fiber system[64].

*Cntnap2*[-/-] mice have multiple circuit deficits that might affect their capability to detect odors in the presence of strong novel background odors. Although we found that increased variability in the *Cntnap2*[-/-] mice glomerular responses was not large enough to affect classifiers' performances, increased variability in other downstream neural structures could affect the performance, resulting in the lower performance of *Cntnap2*[-/-] mice[65]. The lack of *CNTNAP2* causes a broadening of the transmitted action potential resulting in enlarged neurotransmitter release[66] which might cause excessive activation by strong background odors, masking the target odors. However, *Cntnap2*[-/-] mice differences in performance with respect to WT mice on novel background odors became only apparent when novel background odors were presented as catch trials among trials with other well-rehearsed background odors and not when *Cntnap2*[-/-] mice trained over days with the same background odor. The increased release of neurotransmitters would affect both situations when a background odor was novel and when it was a standard background odor, whereas the deficit in *Cntnap2*[-/-] mice was selective for novel background odors.

*Cntnap2*[-/-] mice could be using individual glomeruli or linear classifiers to identify target odors in known background odors, as these algorithms reached performances that matched *Cntnap2*[-/-] mice behavior for known background odors but did not produce good behavior for novel background odors, similar to *Cntnap2*[-/-] mice. These algorithms do not require inhibitory activity and their performance would not be affected in *Cntnap2*[-/-] mice by their reduced expression of GABA[44]. In contrast, deconvolution algorithms that approximate the Lasso behavior require computations[37–40] including normalization[67] which might make them more likely to be affected by reduced inhibition. Changes in cortical spine development[68] and reduced local spine density in cortex[69] in *Cntnap2*[-/-] mice might also affect the capability to integrate information over multiple glomeruli which is necessary to identify odors in novel background odors.

Our results fit in the broader context of previous research with mouse models of autism as well as open new avenues for future research. First, behavioral deficits observed in our task have direct implications for understanding social behavior deficits in *Cntnap2*[-/-] mice[44] and lack of interest in conspecific urine[43], especially as only some of the volatile odor components in a complex mixture like urine signal social information[70]. Thus, our paradigm will allow for future studies using ethologically relevant odors such as those found during social encounters with conspecifics. Second, we demonstrate equal mean average responses but increased variability at the glomerulus level in *Cntnap2*[-/-] mice using intrinsic signal imaging. Thus, future studies can compare our findings with the response to novelty present in neurons along the olfactory pathway to elucidate the circuit mechanism that affects the computations involved in autism sensory issues. Third, our paradigm can be used to examine other mouse models of autism in future studies to test the hypothesis that impaired

response to novelty is a core feature of the autistic brain. Finally, having demonstrated that *Cntnap2*[−/−] mice are impaired at our task, future studies can test pharmacological and behavioral interventions with the goal of finding new approaches to treating olfactory deficits in children with autism[48,71] which could lead to food neophobia[72] and impact the quality of life.

## Methods

All procedures were approved by the institutional animal care and use committee of the New York Institute of Technology, College of Osteopathic Medicine (protocol 2017-GOA-0142).

### Odor stimulus

Odor mixtures were composed of three odors; a target odor, and two background odors. In each trial, the target odor was selected from four possible odors and was presented at a relatively low concentration (0.025% of saturated vapor). We used low odor concentrations so that the background odors could have higher concentrations than the target odors, without having a high absolute concentration that might be aversive to the animals. The odors were selected for their non-toxicity as well as their capability to activate dorsal glomeruli[73,74], and absence of reported innate responses[75].

Both the algorithmic and behavioral performance was based on the discrimination of the target odors. Two of the target odors were labeled go-odors and were associated with water rewards in the behavioral experiments and the other two target odors were the no-go odors that were not rewarded with water. The second odor in the mixture was a contextual background odor, which was selected to be one out of four odors and was presented at a higher concentration (0.1% of saturation vapor). This contextual background odor made the task harder and prepared the animals to identify the target odors in variable backgrounds. The third odor in the mixture was selected from 12 possible background odors and was also presented at a higher concentration (0.1% of vapor pressure). The combination of four targets, with the four contextual background odors and the 12 variable backgrounds resulted in 192 possible odor mixtures.

The training set consisted of 16 mixtures of a target and contextual background, and (s)-(−)-limonene and was used for algorithm training and the animal's behavioral training.

The test set consisted of mixtures of a target and contextual background, but instead of (s)-(−)-limonene, they contained one out of the other 11 novel background odors, which will result in 176 possible test mixtures.

### Odor delivery

We used a serial air-dilution machine[76] that delivered mixtures of target odors embedded in background odors at higher concentrations (see Supplementary Fig. 1). We used a triple serial air-dilution to achieve reliable low odor concentrations for both targets and backgrounds. Each odorant was contained in a 60 ml vial filled halfway with 3 mm glass beads and 3 ml of odorant, which were refilled at the start of each session as necessary. Air was pushed into the odor vials to collect saturated vapor and was serially diluted in air three times, resulting in a diluted odor stream that reached the animal. The odor machine had three groups of odors divided into three-manifolds. The manifold M5 contained the four target odors. The manifold M1 contained the four contextual background odors. The manifold M3 contained (s)-(−)-limonene and the 11 novel background odors. The odor concentration delivered to the animals could be adjusted independently for each manifold. For instance, on day 1, the dilution of 0.5/3 0.5/1.5 0.5/5.2 means that 0.5 liters per minute of odor were pushed into the odor vial. This 0.5 lpm flow was diluted into a 3 lpm air stream (first dilution). Of the resulting 3.5 lpm diluted odor, 0.5 lpm was further diluted into a 1.5 lpm air flow (second dilution). 0.5 lpm from the second dilution was diluted into 4.7 lpm air stream (third dilution)

resulting in a total dilution of 0.5/(3 + 0.5)*0.5/(1.5 + 0.5)*0.5/5.2 = 0.34% from saturated vapor.

Each of the three groups of odors (targets, contextual backgrounds, and novel background plus (s)-(−)-limonene) had a single air valve to direct the odorized air to the animal or into the exhaust. The same valve opened for the go-target odors as well as for the no-go target odors so it was not possible to solve the task by using the valve sound. Similarly, the valve delivering (s)-(−)-limonene was the same valve that delivered the 11 novel odors, so the valve sound was the same for training set trials as well as for the test set trials. All tubing used was Teflon coated and had an internal diameter of 1/8 of an inch. For each stream of odorized air corresponding to each manifold, there was a non-odorized air stream with a matched flow that was redirected in a complementary manner, such as to maintain the total airflow into the animal constant when switching on and off each of the odorized air streams.

### Odor machine latencies

The odor machine has a long delay and a short delay. The long delay is the time required for the odorized air to go from the odor vial to the valve next to the animal snout. The short delay is the time for the odorized air to go from the valve next to the animal snout to the animal snout. We calculated these delays based on the airflows, length of tubing, and tubing cross-sections,

The target odors required 3.4 s to travel from the odor vial to the last valve close to the animal snout and the background odors required 2.6 s. Therefore, we used a 7 s delay between the odor vial valve opening and the opening of the valves close to the animal snout to guarantee the presence of odorized air at the valve, ready to be delivered. We used also a relatively long intertrial interval (30.4 ± 3.3 s, mean ± s.d.) to allow for enough time to wash away residual odor in the tubing.

The three odorized airstreams were collected into a single tube and merged with an air carrier of 4.6 lpm, resulting in a 5.2 lpm flow which was kept constant during training and testing. In order to reduce the air velocity at the animal snout, there was a t-connection just before the airflow reached the animal snout, that directed part of the odor mixture flow into the exhaust, resulting in a reduction of the flow into the animal from 5.2 to 0.7 lpm. The air velocity at the animal's nose was 1.47 m/s. The total delay between the opening of the valve next to the animal nose to the odor reaching the animal, including air transport delay and valve opening mechanical delay (15 ms) was calculated to be 108 ms based on the airflows, length of tubing, and tubing cross-sections. We have used a metal oxide sensor to confirm the latency calculations, as well as to determine the reliability of the odor delivery (see Supplementary Fig. 2).

### Metal oxide measurements of odor machine latencies

We used the Figaro TGS 2620 Organic Solvent Vapor Sensor powered using a 5 V power supply as described before[77]. We replaced one of our contextual background odors with a similar vial containing ethanol at 95% concentration. We connected a 2 kΩ resistor to the output of the sensor resulting in signal responses to ethanol at 0.1% of vapor pressure that was appreciable over the noise level. We sampled the voltage using a NI USB-6003 at 1000 Hz. We removed the metal case of the TGS 2620 sensor and located the sensor close (<4 mm) to where the animal snout would be. We adjusted the airflows to deliver a concentration of 0.1% of vapor pressure. We applied 60 ethanol pulses of 11.5 s separated by 30 s, which mimics the intertrial interval we used during the behavior and imaging. The response time was calculated by performing a single-tailed *t*-test to detect where the first time point in the voltage trace following valve opening was larger than the baseline value at the $p < 0.05$ level, with the baseline calculated as the sensor reading value preceding the opening of the valve next to the animal.

## Behavioral training

Adult female C57BL/6J mice (WT) and *Cntnap2*$^{-/-}$ mice (2–4 months old, original weight 20–24 g) were implanted with titanium head bars[78] and water deprived for at least 7 days before training started (see Supplementary Table 1: Animals used for behavior). Animals were weighted every day. Every day after training, water was supplemented such as an animal's weight reached 80–85% of the weight just before water deprivation started. The water-deprived mice were trained to detect odors in 2 stages before Testing: Target training and Target in known background training (see attached Supplementary Data 1 for a detailed training schedule).

**Target training.** On the first training stage, mice were exposed only to the target odors (no background odors). After the animal's heads were fixed, animals were allowed to rest for at least 5 min before starting the training session. Immediately after head fixation, animals' inhalation patterns were symmetrical with long inhalation and long exhalations. As animals relaxed, their sniff pattern changed to deeper shorter inhalations and shallower exhalations, indicating that animals were ready for the start of the training. Animals were initially exposed to one of the go target odors at high concentration (isopropyl butyrate, 0.34% of saturated vapor) and odor delivery was coupled with direct delivery of a small amount of water (4 µl) after 1 s of odor exposure to elicit licking. Once the animals reliably licked for 10–20 trials, the direct delivery of water was stopped and animals were required to lick the water tube in response to the odor for water to be delivered. The target odor was presented for 3 s. Animals continued to perform using only the go odor. Once they performed >100 trials without early licks (reaction time >300 ms, suggesting they were reacting to odor and not to valve clicking sounds), a no-go target odor (isobutyl propionate, 0.34% saturation vapor) trial was initially introduced with a probability of 10%, so as not to discourage the animal from the continued performance of the task. The minimal reaction time constrain of 300 ms played a role only on the first day of training because animals naturally converged to a response time of ~700 ms from valve opening onset. Once the animals showed robust continuous responses even in the presence of the no-go stimuli, the probability of the no-go stimuli was increased to 50%. Most animals learned to discriminate the first odor pair (one go odor versus one no-go odor) on the first day of training. On day 2, a similar procedure was used for the second odor pair (go odor: propyl butyrate, no-go odor: ethyl propionate 0.34% saturation vapor pressure). On days 3 and 4 both odor pairs were introduced simultaneously and the concentration was reduced over days to reach the target concentration of 0.025%. As described in other go/no-go tasks[28], animals had a tendency to make false alarm errors by licking for the no-go stimuli. In order to discourage that behavior, the probability of no-go stimuli was set to 75% at the beginning of some training sessions. Once animals stopped responding to the no-go stimuli, the go-stimuli probability was returned to 50%. Care was taken to return the go-stimuli probability to 50% before the reaction time of the animals increased to over 1 s, as this indicated that the animals were losing interest and were about to stop the behavior altogether. All behavioral measurements reported here were done at a go-stimuli probability of 50%. Early licks or false alarms produced increased inter-trial intervals (see Supplementary Fig. 7 for a diagram of the time-out schedule).

**Target in known background training.** Mixtures that include background odors were introduced in this phase. The background odors (one of the four contextual background odors and (s)-(−)-limonene) were introduced initially at a concentration of 0.025%, the same concentration as the target odors. Odor delivery started with the delivery of the contextual background odor, followed 0.75 s later by (s)-(−)-limonene. The target odor was presented 1.5 s after the onset of the contextual background odor. The background odors were continuously presented during the 3 s target odor presentation. For some

mice continuous white noise of ~70 db SPL was delivered through a speaker to mask the sound of the background odor valve opening to avoid having animals reacting to the sound of the background valve opening. Animals quickly learned to recognize the targets embedded in the background odors. Over several days the concentration of the background odors was increased in 3 steps (see Supplementary Data 1 for a detailed training schedule). Animals performed at the final concentrations (target 0.025%, background 0.1%) for at least 2 sessions (>400 trials), before the introduction of the novel background odors. All animals, except the animals performing the reduced training set task, were trained using all 16 odor mixtures. For the animals doing the reduced training set, they were exposed only to the 8 odor mixtures specified in Fig. 7B for their entire training.

**Testing.** The 11 novel background odors were presented over 3 consecutive days. Each day, animals started a pre-training session on the known background odors (training set) until their performance reached over 80% for more than 50 trials. Once the reliable performance was obtained, animals were switched to the test trial session that included a small number of novel background odors mixture trials. Each novel background odor was presented at most 4 times per training session and each novel background odor presentation was separated from presentations of the same novel background odor by at least 25 min. On day 1 of testing, only five different novel background odors were presented. On day 2, the other six novel backgrounds were presented. On day 3, all 11 novel background odors were presented. The performance on the known background odors interleaved with the novel odors should be above 80%, or the session was discarded. Only 4 sessions from 63 sessions across 21 animals had to be discarded because of low performance on the known background odors.

**Synchronous trial training and testing.** For the four animals that performed the synchronous task, initial training was identical to animals doing the asynchronous task. Animals did two extra sessions with the training set using the final concentrations of the target and background odor, but where the backgrounds preceded the onset of the target by only 50 ms. The 50 ms delay was introduced to reduce the possibility that the target odor might reach the animal's olfactory epithelium before the background odors arrived.

For the animals trained with the reduced training set, the 11 novel background odors were presented over 2 days. All 11 novel background odors were presented each day, with each novel background odor presented at most 4 times per day.

## Odor pre-exposure

Before starting on water deprivation, a group of 5 WT mice was exposed to the odor group E. Briefly, we put a filter paper soaked in 0.1 ml of each group E odorant in an empty cage. Animals were individually placed in the cage for 5 min at a time. Each day, this procedure was repeated for all five odors of group E. The same procedure was repeated for at least 5 days preceding the onset of the behavioral training. Odor pre-exposure was not done in the 24-h period preceding the beginning of the testing phase.

## Surgical procedure

Female mice >60 days old, 20–25 g were anesthetized using ketamine/xylazine (KX, initial dose 70/7 mg/kg), further supplemented to keep the pedal withdrawal reflex diminished. Respiration and lack of pain reflexes were monitored throughout the experiment. Ophthalmic ointment was applied to the eyes. Aseptic technique was used, first clipping hair and prepping with betadine on the skin. Lidocaine and iodine were applied topically to the skin (as an analgesic and antiseptic, respectively). After the animals were deeply anesthetized, they were mounted in a stereotaxic frame with ear bars. A small incision (2–3 cm)

was made into the skin above the surgical site. A titanium head bar was cemented on the skull near the lambda suture using light-cured Vitrobond (3M). For animals used for imaging, we also implanted 3 mm windows over the olfactory bulb, as described before[78]. Animals were allowed to recover for 1 week before starting water deprivation. During the imaging sessions, the animal's head was held firmly in place by mounting the titanium head bar onto a custom-built holder. The animals were awake during the imaging session.

## Imaging
Intrinsic optical imaging of the olfactory bulb[18] was done using a pair of back-to-back SLR lenses with a 50 mm $f$/1.4 lens used as objective and a second lens Tamron AF 90 mm f/2.8 Di SP AF/MF 1:1 Macro Lens coupled to an sCMOS camera (CS2100M, Thorlabs). The camera was fitted with a long pass filter with a cut-on wavelength of 500 nm (FELH0500, Thorlabs). This setup resulted in a resolution of 3.3 μm per pixel. White light from a flashlight was used to find the surface of the olfactory bulb. The imaging plane was set between 200 and 250 μm below the vasculature on the surface of the bulb. Single odors and odor mixtures were presented for 9 s randomly interleaved.

**Intrinsic imaging.** Images were acquired at a rate of 40 Hz. The surface of the bulb was illuminated using an infrared ($\lambda = 780$ nm) fiber-coupled LED (M780F2, Thorlabs) connected to a 1000 μm fiber of 0.50 NA (M59L01−Ø1000 μm, 0.50 NA, Thorlabs). The fiber output was collimated using a fiber collimation package $f = 8.00$ mm, NA = 0.50 (F240SMA-780-780, Thorlabs). Illumination was set up to reach 80% of the pixel saturation value. To avoid saturation of the camera, the aperture of the 50 mm lens was closed to $f$/16, which resulted in a NA for the lens of 0.0313. This configuration resulted in a diffraction-limited resolution of 12.5 μm for a wavelength of $\lambda = 780$ nm.

**GCaMP6f imaging.** Images were acquired at a rate of 4 Hz. We used a 470 nm LED (M470L4, Thorlabs) mounted with a GFP excitation filter (MF469-35, Thorlabs) and a diffuser (ACL2520U-DG6-A, Thorlabs) to produce a uniform and pattern-free illumination. The illumination was setup to the minimum value that produced noticeable increases in fluorescent responses to avoid photobleaching. The aperture of the objective lens was opened to the maximum value of $f$/1.4 to maximize light collection.

Image acquisition started 7 s before odor onset. An inter-trial interval of 40 s was used between odor presentations in order to avoid adaptation effects. Each stimulus (either single odors or mixtures) was repeated between 16 and 26 times.

## Image analysis
Images in response to the same stimulus were averaged over the repeats ($n > 16$). A normalized signal d$f/f_0$ was calculated for the average image using as f0 the average response of the 5 s period preceding odor onset. In order to remove the broad hemodynamic signal in response to odors in intrinsic images, the images were convolved with a Gaussian of radius $\sigma = 40$ μm and this low-pass-filtered signal was subtracted from the original signal. In order to remove high spatial frequency spatial noise, the resulting images were further convolved with a Gaussian of radius $\sigma = 12$ μm. In order to normalize this signal, a z-score was calculated for each pixel using the values of d$f/f_0$ of the 5 s before the onset of the odor period to calculate the mean and the standard deviation. Average z-score odor responses were calculated using the period between 2- and 9-s following odor onset. ROIs were drawn manually using ImageJ[79] over activated glomeruli across all odors presented. Activated glomeruli were drawn using the minimal projection over all the odors presented. We quantified glomerular activation as the mean value of the z-score across all selected pixels in an ROI.

## Measurement of trial-to-trial variability in WT mice
We quantified the trial-to-trial variability of the glomerular response using the coefficient of variation (CV). To determine the coefficient of variation (CV) of WT mice, we calculated the z-score in response to individual odor presentation in three animals: two WT mice that were exposed to all 20 odors and one WT mouse that was exposed to the training set and test set odors where cineole was the novel background odor. Odors were presented as 9 s pulses and repeated between 20 and 27 times. To have a good estimate of the variability we only considered responses from ROIs that showed a strong response with an average z-score < −2 for at least one of the odors tested. This resulted in 141 ROIs selected and 2684 ROI−odor pairs. As described above, the glomerular response to an individual trial was the average z-score of the response between 2 and 9 s from odor onset. We calculated the mean($\mu$) and the total variance($\sigma^2$) of the glomerular response over the trials. The total variance increased with the mean value of the evoked response consistent with a previous report[19]. In addition, there was also a pedestal of variability for ROI−odor pairs responses that were independent of the average activation and even appeared for ROI−odor pairs with zero or positive average responses which are related to imaging-related noise. The CV relates the average odor response to the observed standard deviation: $\sigma^2(\mu) = CV^2\mu^2$. Therefore, the total observed trial-to-trial variability $\sigma(\mu)^2$ is given by

$$\sigma^2(\mu) = \sigma^2_{noise} + CV^2\mu^2 \qquad (3)$$

We fitted this function using the Matlab function *fit*. $\sigma_{noise}$ correspond to the variability in the absence of an odor response and was 1.59 with the 95% CI [1.52, 1.66]. The coefficient of variation was 0.34 with the 95% CI [0.30, 0.37] which is within the range of previous measurements[19] of 0.37 ± 0.07 (mean ± SD) using calcium imaging in anesthetized mice.

## Measurement of uncorrelated component of trial-to-trial variability in WT mice
A large fraction of the trial-to-trial variability for a given odor was shared across glomeruli which had been partially ascribed to global anesthesia effects[19]. Our intrinsic imaging data was measured in awake WT mice, so we expected that the correlated noise component would be smaller compared to the anesthetized condition. However, there were still correlated responses between glomeruli (see Supplementary Fig. 3), that is, a trial where a glomerular response produced a larger-than-average odor response would also produce larger-than-average odor responses in other glomeruli.

To determine the coefficient of variation of the uncorrelated variability, we used a previously develop method[19]. Briefly, for each odor presentation, we plotted the average response of an ROI−odor pair across trials against the response for a particular odor presentation. For each odor presentation, we fitted a line for all the simultaneously recorded ROIs responses. The population correlated fluctuations fall on a line because the fluctuations are proportional to the average response of each glomerulus. The deviations from a fitted line correspond to the contribution of the uncorrelated noise for that ROI for that odor presentation. For each ROI−odor pair we calculated the variance ($\sigma_{uncorr}$) from the distribution of uncorrelated noise across trials. The coefficient of variation of the uncorrelated response $CV_{uncorr}$ relates the average response to the mean response $\mu$ as $\sigma^2_{uncorr}(\mu) = CV^2_{uncorr}\mu^2$. However, in the intrinsic imaging, there was still uncorrelated variance in the absence of an average odor response due to imaging noise which we called $\sigma^2_{uncorrnoise}$. Therefore, the total observed uncorrelated variance was

$$\sigma^2_{uncorrnoise}(\mu) = \sigma^2_{uncorrnoise} + CV^2_{uncorr}\mu^2 \qquad (4)$$

We fitted the standard deviation of the uncorrelated response $\sigma_{uncorr}(\mu)$ using the average response of the mean $\mu$. We estimated a value of $\sigma_{uncorrnoise}$ of 1.47 with 95%CI [1.46, 1.49]. This value is very similar to our noise estimate $\sigma_{noise} = 1.59$ which indicates that the imaging noise was mostly uncorrelated across glomeruli and probably related to imaging noise. The coefficient of variation for the uncorrelated variability $CV_{uncorr}$ was 0.25 with 95% CI[0.23, 0.27]. This value is larger than the value measured by Mathis et al $0.099 \pm 0.019$ in anesthetized WT mice.

### Relationship between intrinsic imaging and calcium imaging of olfactory bulb output

We implanted a C57BL/6J-Tg(Thy1-GCaMP6f) GP5.11Dkim/J[20] mouse with a window over the olfactory bulb as described above. This mouse line expresses GCaMP6f in mitral and tufted cells and has been used to measure the output of the olfactory bulb[21]. The mouse was presented with 20 odors using 9 s odor pulses at the concentrations used for the behavior. We measured the intrinsic odor response by using a 780 nm light and collecting the reflected light as described above. Afterward, we measured the bulbar output response by using a 470 nm blue light and collecting the green fluorescent responses. ROIs were drawn based on the intrinsic signal. We analyzed the responses of 131 ROIs resulting in 2620 ROI–odor pairs.

To determine the level of intrinsic glomerular activation that resulted in significant activation of postsynaptic cells, we estimated the fluorescent signal that corresponded to ROI–odor pairs that lacked an odor-evoked response. ROI–odor responses that have a positive value on their intrinsic signal were considered non-responding as the intrinsic signal was characterized by negative deflections. The fluorescence signal corresponding to those non-responsive ROI–odor pairs determined our noise distribution for the fluorescence signal. The noise distribution for the fluorescent signal corresponded to a $z$-score of $-0.36 \pm 0.10$ (mean ± s.e.m., 1083 non-responsive ROI–odor pairs). We compared the fluorescence signal of the data binned data according to the $z$-score on the intrinsic signal with the noise distribution of the fluorescence signal. Negative intrinsic $z$-scores were associated with significant positive fluorescent $z$-scores (see Supplementary Fig. 4). Even fluorescence signals associated with relatively small intrinsic imaging $z$-scores (intrinsic $z$-score between 0 and $-0.3$) produced a fluorescence $z$-score ($0.20 \pm 0.15$, mean ± s.e.m., $n = 543$ ROI–odor pairs) that was significantly different from the fluorescent noise distribution ($p = 0.03$, $t$-test).

### Measurement of trial-to-trial variability in *Cntnap2*[−/−] mice

We measured the trial-to-trial reliability of the intrinsic responses of the *Cntnap2*[−/−] mice using the same procedure as the one used for the WT mice. We measured the individual odor presentations in 5 *Cntnap2*[−/−] mice that were presented with 20 odors. We selected ROIs that showed a strong response with a $z$-score $< -2$ for at least one of the odors tested. This resulted in 149 ROIs selected and 2980 ROI–odor pairs. The estimated coefficient of variability reflected the larger variability of the *Cntnap2*[−/−] mice. The coefficient of variation was 0.64 with 95% CI [0.60, 0.68] which is almost twice as large as the coefficient of variation measured in WT mice (0.34). The variability in the absence of an odor response $\sigma_{noise}$ was 1.83 with 95% CI [1.76, 1.89] which is similar to the value of 1.59 with 95% CI [1.52, 1.66] measured in the WT mice.

### Measurement of uncorrelated component of trial-to-trial variability in *Cntnap2*[−/−] mice

*Cntnap2*[−/−] glomerular responses were also correlated on a trial-by-trial basis in *Cntnap2*[−/−] mice (see Supplementary Fig. 12). We followed a similar procedure as the one used for the WT mouse to calculate $CV_{uncorr}$. We estimated a value of $\sigma_{uncorr}$ of 1.65 with 95% CI [1.64, 1.67]. This value is also very similar to our noise estimate $\sigma_{noise} = 1.83$ which

indicates that the imaging noise was also mostly uncorrelated for *Cntnap2*[−/−] mice, similar to WT mice. The coefficient of variation for the uncorrelated variability $CV_{uncorr}$ was 0.44 with 95% CI [0.42, 0.47]. This value is larger than the value of 0.25 with 95% CI [0.23, 0.27] of the awake WT mice reflecting the larger variability of the *Cntnap2*[−/−] mice odor-evoked responses.

### Recording responses from odor mixtures

For each novel background odor, we acquired images in response to mixtures of the target, contextual background, and (s)-(−)-limonene (16 training mixtures). We also acquired images of the same mixtures of targets and contextual backgrounds, where (s)-(−)-limonene was replaced by one novel odor (16 testing mixtures per novel background odor). Each mixture was repeated between 16 and 25 times. Because of the long acquisition time required, we split some recording sessions in two for individual novel background odors. In those cases, on each day we presented either one go-odor and one no-go odor with all 4 contextual background odors (8 training mixtures and 8 testing mixtures), or we presented the two go-odors and the two no-go odors with two of the contextual background odors (8 training mixtures and 8 testing mixtures). We trained the linear classifier and used as templates for the nearest neighbor classifier the mixtures that contained (s)-(−)-limonene. For the split session, we report the performance of the classifiers calculated in the two individual recording sessions. We recorded data from 10 novel odors because one of the novel background odors, 2–3 pentanedione, stuck to the odor tube, and the large number of trials required (over 150) clogged the odor vial valve.

### Linear classifiers

Linear classifiers were created using the Matlab (R2017b, ver 9.3.0.713579) function *fitclinear*. The function was trained using as input the glomerular activation patterns (as $z$-scores) of the 16 odor mixtures (or 8) that included (s)-(−)-limonene that conformed to the training set. The function calculated the weights $w_i$ and the bias $w_0$, that is,

$$c_j = \sum_{i=1}^{n} w_i s_{i,j} + w_0 \tag{5}$$

where $s_{i,j}$ is the average glomerular activation of the $i$th glomerulus of the $j$th training mixture and $n$ is the number of ROIs. The training variable $c_j$ had the value of $+1$ for the 8 mixtures (or 4 mixtures), from the training set that included a go odor, and a value of $-1$ for the 8 mixtures (or 4 mixtures) that included a no-go odor.

When using real odor mixtures, the test set was based on the average glomerular responses of the 16 (or 8) mixtures that included a novel background odor. For the testing with virtual odor mixtures, the test set was based on the 176 virtual mixtures (or 88) that included the novel background odors. The complete test set consisted of simulated 100 instantiations of each average glomerular response that the test set was based on using Eq. (1).

The output of the linear classifier (either a go or no go) was based on the sign of the output of the filter when applied to the test set mixtures, with positive values considered a go stimulus.

### Nearest neighbor classifier

For the nearest neighbor classifier, we used as templates the average glomerular activation patterns (as $z$-scores) of the 16 odor mixtures (or 8) that included (s)-(−)-limonene that conformed to the training set. The complete test sets were the same ones as the ones used for the linear classifiers. A dot product was calculated between the templates and the test set mixture. The nearest neighbor was determined as the template with the largest dot product. The output of the algorithm (either a go or no go) was based on the target odor content of the nearest neighbor.

In order to create all the classifiers used, we preprocessed the $z$-score of the recorded glomerular activation by centering, a common preprocessing technique in data science that has been applied to neural decoding[80]. We centered the responses of each glomeruli by subtracting the average response of that individual glomerulus from the training set. Centering resulted in increased performance of the linear classifiers as well as the NNC (see Supplementary Fig. 5).

## Measurement of saturation non-linearity
In order to measure the saturating nonlinearity of the intrinsic imaging signals, we presented 2 odors in separate trials as well as mixed together and compared the sum of the intrinsic signals (as $z$-score, see above) of the individual odors with the intrinsic signal produced by the mixture. We used ethyl butyrate (0.1%) and ethyl tiglate (0.1%) as well as ethyl valerate(0.1%) and ethyl tiglate(0.1%). We recorded 143 glomeruli on two WT mice and found 68 glomeruli that responded to either the odors used or to the mixture with a value that exceeded the $z$-score threshold (see Supplementary Fig. 13).

We calculated the deviation from the identity and found that there was a $39\% \pm 0.05$ ($n = 68$) deviation from the identity line for responses of individual glomeruli. As this measure includes both noises as well as saturation, we fitted the data using a saturating non-linearity (shown in green) similar to[19], that is $\sigma(R\_o) = 2A/(1 + e^{\wedge}(R\_o *s)) -A$, where $R\_o$ is the sum of the $z$-score produced by the two odors measures separately and $\sigma(R\_o)$ is the $z$-score measured for the mixture of both odors. $A$ is the asymptote value and $s$ is the saturation constant. The estimated saturating values of the fitted model were $A = -8.0934$, $s = 0.2091$.

## Creation of virtual odor mixtures
Virtual odor mixtures from the training and test set were created by adding the glomerular activation patterns (as a $z$-score produced by individual glomeruli) of individual odors from the WT mice (5 mice, 775 ROI) and the *Cntnap2⁻/⁻* mice (5 mice, 643 ROI). The 16 mixtures of the training set were generated by adding the contribution of three odors: a target odor (0.025% saturated vapor), one contextual background odor (one out of 4 odors, concentration 0.1 % saturated vapor) and (s)-(−)-limonene (0.1% saturated vapor). $z$-score were curtailed such as activations that had $z$-score larger than the threshold were set to zero. The threshold for WT mice was $z$-score $= -0.42$ and for *Cntnap2⁻/⁻* mice was $z$-score $= -0.46$. The sum of activations was processed through the saturating nonlinearity (see Supplementary Fig. 13B) resulting in an average response to the mixture μ. To create individual instantiations of an odor presentation, uncorrelated gaussian noise with zero mean and standard deviation proportional to the measured activation μ was added to each glomerular activation data μ using Eq. (1). The constant of proportionality used was $CV_{uncorr} = 0.25$ for WT mice, and $CV_{uncorr} = 0.44$ for *Cntnap2⁻/⁻* mice.

## Lasso deconvolution comparison with NNC, linear SVM, and logistic regression
In order to test if the Lasso could identify target odors in novel backgrounds we created large dictionaries of glomerular activation patterns based on our glomerular imaging from 5 WT mice in response to the individual odors used with $155 \pm 38.1$(mean ± s.d.) ROIs recorded per animal. The created dictionaries included the glomerular activation patterns of all nine individual odors used in animal training (4 targets + 4 contextual background +(s)-(−)-limonene) but did not include the 11 novel background odors as the animals had no exposure to them during training. We assume that the dictionaries should also contain glomerular patterns for other odors besides the odors that animals were exposed to during training. To include novel dictionary elements while preserving the average activation per glomerulus and the interglomerular correlation, we calculated the mean and

covariance matrix from all the 20 odors used in the experiments. We generated the additional dictionary elements using a gaussian process with this mean and covariance matrix. Although we imaged the odor responses at the concentrations used during the behavior, each dictionary element was normalized to unit variance. We tested dictionary sizes between 100 and 1000 elements. The Lasso algorithm was implemented using the *lasso* function from Matlab 2017B (Mathworks). The regularization constant was $\lambda = 0.0001$. For each dictionary size, we simulated 30 different dictionaries and each dictionary performance was evaluated using an instantiation of the 88 mixtures of the reduced test set using the virtual odor mixture, resulting in 13,200 test mixtures per dictionary size. The NNC, the linear SVM, and logistic regression were trained using the average virtual mixtures of the reduced training set (88 mixtures) and their performance was evaluated with the same instantiations of the reduced test set used to test the lasso.

## Sniff detection
Sniff was detected using an airflow sensor (1000 SCCM AWM300V, Honeywell) circuit connected opposite to the animal nose[81]. A soft viton o-ring (ID = 3/16 inches, Macmaster Carr part number 1284N108) was glued to the odor delivery port to create a tight seal without discomforting the animal. The airflow signal was acquired after passing it through an antialiasing filter to remove high-frequency turbulence-associated noise. We used as antialiasing filter a low-pass RC filter with $R = 9.4\,k\Omega$ and $C = 0.3\,\mu F$, given a cutoff frequency of $56.5\,Hz$. The signal was acquired at 1000 Hz using a NI USB-6003 USB board.

## Sniff rate quantification
Sniff rates were calculated by counting the number of inhalations in a given period divided by the period length. For the asynchronous odor delivery, we evaluated the sniff rate in four 750 ms periods: the baseline air period, the contextual odor period, the (s)-(−)-limonene period, and the target period. The baseline air period was defined as the 750 ms preceding the onset of the contextual background odors. The contextual air period was the 750 ms period starting at the onset of the contextual background odor and finishing at the onset of (s)-(−)-limonene The limonene period started at the onset of (s)-(−)-limonene deliver and stopped at the onset of the target odor. The target period was defined as the 750 ms window starting at the onset of the target delivery. For the synchronous odor delivery, we evaluated the sniff rate in two periods: a baseline of 750 ms preceding the onset of odors and a 750 ms period starting with the onset of the background odors. All the odor onsets included a correction for the 121 ms of odor delivery delay (see Supplementary Fig. 2).

## Changes in sniff rate produced by novel background odors
In order to determine the change in sniff rate produced by the novel odors, we calculated the number of inhalations in a 1s window between 300 and 1300 ms following the onset of the novel odor or the (s)-(−)-limonene odor. In order to have a comparison that considers the non-stationarity of the sniff rate across trials, we compared the presentation of mixtures with novel background odors to the preceding trial with known background odors. To determine the reaction time in response to a target odor, we assumed that the odor reached the olfactory epithelium at the onset of the first inhalation following the target odor valve opening plus the 121 ms of odor delivery delay calculated based on air flows and cross-section of the tubing and confirmed using the odor sensor. We measured the lick latency from this time point.

## Reporting summary
Further information on research design is available in the Nature Portfolio Reporting Summary linked to this article.

## Data availability

Small dataset files to reproduce the figure panels are available at https://github.com/gotazu/nat_comm_2022. Larger datasets files are available from gotazual@nyit.edu upon request. Source data are provided with this paper.

## Code availability

All the Matlab code necessary to reproduce the figure panels are available at https://github.com/gotazu/nat_comm_2022. An excel sheet (M_files_figures) indicates the m file that will reproduce a given figure panel.

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

## Acknowledgements

We thank the members of the Animal Facility Team for their animal care support and the Academic Technologies Group for their informatics support. We acknowledge the New York Institute of Technology College of Osteopathic Medicine's Center for Biomedical Innovation for the computing equipment used in the course of this research. This work was

supported by startup funds (G.H.O.) from the New York Institute of Technology College of Osteopathic Medicine, departmental funds from the Department of Biomedical Sciences (G.H.O.), and an In-House Grant Program (G.H.O.).

## Author contributions

Y.L., M.S., T.S., A.R., N. Sharma, R.S., M.C., D.M., J.W., and G.H.O. performed the behavioral experiments. Y.L. and G.H.O. performed the imaging experiments. Y.L. and G.H.O. performed the surgeries. Y.L., M.S., T.S., A.R., N. Sharma, R.S., D.M., J.W., A.A., N. Shah, and G.H.O. built the odor deliver systems, and the behavioral and imaging setups. Y.L., M.C., J.S.M., J.C., and G.H.O. analyzed imaging data. R.L.R. and G.H.O. analyzed and discussed the results. G.H.O. designed the experiments, designed the analysis pipeline, wrote the analysis code, wrote the manuscript, and supervised the project.

## Competing interests

The authors declare no competing interests.
