## [Peer Review File · Nature Communications]

Robust odor identification in novel olfactory environments in miceREVIEWER COMMENTS

Reviewer #1 (Remarks to the Author):

In this manuscript Otazo and colleagues develop a new olfactory behavioral paradigm (background / foreground separation with new distractor backgrounds) and show that while WT mice readily learn and generalize, one mouse model of autism struggles.

I find the paradigm relevant and well studied, developed and described and while the autism model is only a very small part of the paper I can see the relevance of the paper for the autism research community. Overall my main issues with the paper are regarding presentation in the text, description in the methods, references, discussion as well as some related data analysis issues.

- They show that one linear classifier methods (SVM) doesn't perform as well as the animals. However, given the number of parameters, priors etc, it is a bold claim to conclude that linear classifiers in general can't perform. The authors need to broaden the number of linear classifier methods they test to comfortably make this claim.
- The effect of feedback inhibition on odour identification in a novel background is mentioned many times throughout the paper. E.g. the authors speculate that reduced inhibition causes impaired odour identification in *Cntnap2*^{-/-} mice but the paper clearly falls short in providing evidence for this. Considering that the senior authors has injected Muscimol into PCx in previous work it should be feasible to directly experimentally test this. If this is too much work at this point in time, the authors need to substantially reduce these mechanistic speculations.
- Considering that 90% of the paper consists of data from WT animals, the emphasis on the mouse model of autism in both title and abstract is too strong for my taste. My suggestion would be to either rephrase to a more descriptive version that better represents the data from WT mice or provide more data in *Cntnap2*^{-/-} mice to justify the emphasis.
- L 113 Why do novel backgrounds result in lower glomerular activation? Or why were they chosen as such? This seems to make target detection easier?
- Some of their regression fits, e.g. in 5D ($\rho=1.1$) look odd; how much does the fit depend on a few outliers (that might put the underlying assumption of normality to determine significance into question).
- Why does the linear classifier performance drop so much when they use the real data (5G) vs. their pseudo-data (5B)?
- L 479 "The first one was to determine if the Lasso assigned a larger contribution to the correct target (go or no-go) than to the incorrect target. If the Lasso estimate did so, this was considered a correct discrimination." I'm not sure what they're doing here. First, what is the incorrect target? Second the Lasso is not a discrimination method, it's a sparse recovery method: you stick in glomerular activations, and it spits out an odour concentration vector. It's agnostic to what you want to do with the odour vector (whether it's correct or not). This is another example of their computational methods being very unclear.
- L 293: Sparse deconvolution algorithms "use the training data more efficiently than the NNC"; this is incorrect, in that at least the two algorithms he cites which I am aware of (Koulakov 2011 and Grabska Barwinska 2017), do not have a training step, and are provided at the start with the correct dictionary.
- The methods, particularly the computational methods, weren't clear. For example, I had to infer from the discussion that they used 7-second averages of the imaging signal as inputs to their classifiers, but I couldn't find this in the actual methods. Also, how many glomeruli did they use? They say (in the discussion I think) that it's ~200, but they should state this in the methods.
- The argument that autism mouse models struggle with novel odours on a computational level only based on the observation that they avoid environments with novel background odours seems far-fetched.
- I find it a very interesting observation that reaction time / number of sniffs very quickly fell back to trials with known background. In the light of odour novelty and sniffing they really should cite Esquivelzeta Rabell...Haesler, 2017, Current Biology.

- The authors should discuss why the sniff rate increase is less pronounced in synchronous presentation condition (Figure 4E) compared to the asynchronous condition (Figure 3B).
- L 224: The authors should elaborate on the statement “The very simple circuit implementation of a linear classifier is less likely to be affected in mouse models of autism”.
- I find it confusing that the authors refer to potential computational strategies in autism models using broad statements throughout the results section but only show the Cntnap2 -/- data at the very end. The impaired odour identification in novel background seems like a solid observation but, unfortunately, remains purely descriptive.
- L 28-30 : “Critically, the glomerular representations of mixtures with different weak targets but the same strong background odor are dominated by the background odor and are very similar.” – this is unclear – what is this based on, how is this conclusion arrived at – the authors need to expand (I agree with the conclusion but the logic needs to be clear)
- L 33 “Since novel odors produce larger mitral cell activation compared to familiar odors (Kato et al. 2012)” – Isn't this simply due to changes in sniffing?
- L 52-56 – the phrase “temporal gap” is a confusing phrase – it is rather lack of correlation in Hopfield 1991
- Please plot the performance of individual mice in e.g. Fig 2a-d
- Segmented and false color would make the intrinsic imaging figures much easier to grasp
- There are multiple duplications in references – fukunaga 12a,b; igarashi 12
- L 243 “Could be predicted using a non-linear classifier” – that is a bit too general, almost everything can be predicted using a non-linear classifier. They need to specify which type, which conditions etc.
- L 311 “glomerular patterns were very similar” – this needs to be quantified and compared to within WT / Cntnap2 variability
- Please plot Individual animals in 7F,H
- L 358 – “longer time gaps”: Szyszka gaps are in the millisecond range
- L 365 – citing Meister 2015 for low dimensionality of odor space is somewhat misleading - the paper only makes the claim that dimensionality is unknown as the arguments of the Keller et al paper are discussed / partially debunked
- Figure 1E: correct to “intertrial time”
- Line 363: change “glomerular” to glomeruli
- Line 479 "The first one was to determine if the Lasso assigned a larger contribution to the correct target (go or no-go) than to the incorrect target. If the Lasso estimate did so, this was considered a correct discrimination." This is confusing. First, what is the incorrect target? Second the Lasso is not a discrimination method, it's a sparse recovery method: you stick in glomerular activations, and it spits out an odour concentration vector. It's agnostic to what you want to do with the odour vector (whether it's correct or not). This is another example of their computational methods being very unclear.
- L 293: Sparse deconvolution algorithms "use the training data more efficiently than the NNC"; this is incorrect, in that at least the two algorithms he cites which I'm aware of (Koulakov 2011 and Grabska Barwinska 2017), don't have a training step, and are provided at the start with the correct dictionary.
- There are lots of typos throughout and a thorough proofread is critical.

Reviewer #2 (Remarks to the Author):

The manuscript by Li and co-workers performs an innovative study of decoding the targeted stimulus from glomerular activity in a go-no go task similar to the visual CAPTCHA used to distinguish humans from computers. In the olfactory CAPTCHA go-no go task small concentrations of target odorants (go vs. no-go) are presented in the presence of higher concentrations of known and novel background odorants. The task is well-designed and provides potentially important data indicating that olfactory processing can differentiate odorants masked within novel background stimuli by an algorithm related to the nearest neighbor classifier. In addition, the authors show that mice of the Cntnap2-/- model of autism are not able to cope with the presence of novel backgrounds when trained with a small number of target odorants. Unfortunately my enthusiasm is significantly reduced due to poor presentation of glomerular responses to odorants that makes it impossible to judge the validity of the

data, evaluation of results with questionable statistical methods and lack of a thorough consideration of published data.

Major comments

1. The quality of the images of glomerular activation patterns shown throughout the manuscript are poor (e.g. Fig. 1A). Furthermore, it is not stated in the legends what the gray scale intensity represents (z score referenced to no odor?) and no gray scale calibration bars are shown. The most active regions appear to be in the periphery with few responses within raising the question whether the data are valid. Furthermore, there is no information on how individual glomeruli respond to different odorants. A supplementary figure should be provided making clear how individual glomeruli respond to target and background odors as well as mixtures, demonstrating the trial-to-trial reliability of differential glomerular activity and showing how the average glomerular activation shown in Fig. 1C was calculated. The description of the methods for intrinsic optical image acquisition and analysis is poor. What is the criterion for activation of a glomerulus?
2. Throughout the manuscript there is lack of detail on experimental details and statistics. For example, for Fig. 1C it is only stated that :”C. Average glomerular activation”. It is not stated what the error bars are, how many animals were used and how many measurements per mouse were performed. There is no indication whether the different bars are statistically significantly different. No multivariate statistical analysis or correction for multiple comparison is presented when necessary. For every statistical test please give the p value, the number of samples and the number of animals. For statistical tests was there correction for multiple comparisons?
3. The authors did not discuss how the abundant literature on signal processing in the olfactory bulb during the go-no go odor discrimination task relates to their findings (see for example Doucette and Restrepo, PLOS Biology 6:e258, 2008, Chu et al. Neuron 92:174, 2016, Wang et al., J.Neurosci, 39:10002, 2019, Losacco et al., eLife 9:e52583, 2020). These publications show that responses to odorants recorded with different techniques (electrode recordings, multiphoton or fiber fluorescence imaging, or recording of local field potential) become divergent between the rewarded and unrewarded stimuli as the animal learns to differentiate the odorants. How does this relate to the validity of the different signal processing algorithms discussed in this manuscript? Would an increase in divergence in responses to the rewarded and unrewarded odorants favor a winner takes all algorithm?
4. Concerning point 3 do the glomerular odorant responses change as the animal learns to discriminate the target odorants? Were the responses recorded in animals responding to odorants in the go-no go task? Changes in odorant processing during learning may alter glomerular responses through pre-synaptic modulation.
5. When mice were trained with a restricted set there was a reduced performance for both the linear and nearest neighbor classifiers, but not for mouse behavior (Fig. 6D). The authors propose Lasso deconvolution as a potential algorithm used by WT mice. Unfortunately, this is not a satisfying solution because the Lasso algorithm cannot be tested with the data available in this study. Furthermore, a major limitation of this study is that the responses are only measured in the dorsal olfactory bulb. Would the performance of the linear or nearest neighbor algorithms for the restricted odorant set increase if the ventral glomeruli were added to the data set? AAs a control, what happens if decoding input restricted to half of the dorsal glomeruli?
6. The use of glomerular activation data generated by virtual odorant mixtures for the decoding analysis is not granted given the well-known nonlinearities of the olfactory system. It does not appear that there is a correlation between virtual and real odorant mixture decoding. Unless the use of virtual odorant mixtures is validated by data showing that there is a correlation for decoding performance with real odor mixtures I would advise to remove all the virtual odorant data from Fig. 5, and leave only the real odorant mixture results throughout the manuscript.

7. The paragraph starting in line 307 makes the argument that the similarity of odor pair responses correlates between WT and *Cntnap2*^{-/-}. However, the data in Figs. 7A and B do not corroborate this conclusion. First, the examples in Fig. 7A do not illustrate the similarity in response of an odor pair. Responses are only shown for one odorant and they appear fairly different raising the question whether there are significant differences in glomerular input. This would not be surprising given that *Cntnap2* is expressed in the olfactory epithelium. Second, there is no panel showing how the similarity of response patterns is computed. Third, the correlation in Fig. 7B is significant, but there are clear differences between WT and *Cntnap2*^{-/-}. Is the correlation shown in Fig. 7B similar for regular and novel background odorants? The manuscript should show a thorough comparison of odorant response patterns and odorant pair similarities between genotypes.

8. Is the fraction of trials for Miss in Fig. 7F significantly different between genotypes? Since the comparison between genotypes is the main point of Fig. 7F please plot the bar graph with the bars for the two genotypes next to each other and asterisks denoting statistical differences. Comparing the time of response for the asynchronous case between WT (line 267) and *Cntnap2*^{-/-} (line 337) it appears that the response for *Cntnap2*^{-/-} was slower. Is this correct? Was there a difference between genotypes for the number of sniffs required for decision making?

Minor comments

1. Please revise the English throughout.
2. For the paragraph starting in line 97 please reference a diagram that illustrates the experimental design.
3. Line 108. Please reference Slotnick since his publications were those that placed the go-no go task on the map for studying olfaction in rodents.
4. The manuscript refers to “false rejections”. This term is not often used in the literature. Please use the standard terms: Hit, Miss, Correct Rejection and False Alarm.
5. Line 175: change to “quantified the animal’s familiarity”
6. Line 179: change “Consistently” to “consistent”
7. Lines 199-200. Please provide the statistical basis for the statement that “the mice had an increase in false rejections”
8. Fig. 4E. Is this mean+SEM?
9. Line 215. Please show the correlation of brief inhalation and higher performance.
10. Line 221 change “mice’s” to “mouse’s” here and in several other instances.
11. Line 224. What is the basis for the statement that the linear classifier is less likely to be affected in mouse models of autism? Please clarify or remove the statement.
12. Line 416. Change “dids” to “did”
13. Line 428. Was the 0.8% increase statistically significant?
14. Line 451. Please provide statistics.
15. There is a problem with the legend of the data in Fig. 7. Are the data in Figs. 7D and E for the full

or the restricted training set? Please state the results of the statistical tests and give the number of animals and trials included in the data. What are the error bars?

16. For Fig. 4C please plot the novel background bars next to the known background bars and show whether the differences are statistically significant. Was there correction for multiple comparisons?

17. Please provide a diagram for the air-dilution machine and provide PID measurements of odor delivery time course.

18. What is shown in pseudocolor in Sup. Fig. 1?

19. What does the gray scale in Fig. 1A show? Are these negative z scores? Please show a gray scale with the z score range.

Reviewer #3 (Remarks to the Author):

Detecting a relevant odor (figure) in the background of less relevant odors (ground) is a critical task that animals need to solve in a natural environment. Here Li et al. investigated if mice can solve this figure-ground separation problem using a go/no-go task. The major finding is that the mice can correctly perform the task even when the identity of a background odor was switched between a training and a test phase. They presented two quantitative models that can solve the same problem based on neural responses to odors acquired through intrinsic signal imaging; however, both fell short of explaining the actual behavior of mice in a challenging condition. Finally, the authors report that the *Cntnap2*^{-/-} mouse model of autism failed to perform the task when the background odor was switched in a test phase.

Although the authors have revealed an aspect of olfactory behavior, this represents a rather modest conceptual advancement as Rokni et al., (Nat Neurosci, 2014) have already shown in a similar scheme that mice are capable of separating figure from ground odors; the main difference is limited to the use of a novel background odor during the test. The lack of satisfactory behavioral model and the mere description of a performance of *Cntnap2*^{-/-} mice further dampen the enthusiasm for this manuscript.

Major comments:

1. The authors initially frame the work (in the title, abstract, and the introduction) as an investigation into the function of *Cntnap2*^{-/-} mouse model of autism. However, this work is fundamentally about the olfactory behavior and physiology of wild type mice. This is misleading as examination of the *Cntnap2*^{-/-} mouse is very limited and remains very descriptive. The authors repeatedly note that they decided to focus on *Cntnap2*^{-/-} mice because they have reduced neural activity as well as inhibition, and these properties might compromise olfactory discrimination. However, the link remains weak; neither the effect of change in neural activity/inhibition on the behavior nor the state of neural activity/inhibition in *Cntnap2*^{-/-} mice in the target olfactory areas are tested. I would suggest to reframe the abstract and introduction.
2. The performance of a linear classifier is not correlated with that of the animal behavior, and the explanatory power of the other nearest neighbor classifier (NNC) is limited to a particular context (16 mixture testing and not 8 mixture testing). Negating models certainly narrows down the actual algorithms implemented in the brain, but one would obviously prefer to see a model that explains the experimental findings in a work.
3. Linear classifier and NNC are trained using both virtual and actual odor responses. If actual odor responses are available, how come the main analyses in Figures 5 and 6 were conducted with virtual responses (There have been various reports on the linear and non-linear additivity of component responses in the bulb)? How did virtual and actual odor responses compare using intrinsic imaging?
4. There is a text saying that "ROI were drawn manually using ImageJ over activated glomeruli across all odors presented (line 706)", but there is no description on the definition of activated glomeruli, where trial-to-trial variability, area of ROIs, criteria for separating neighboring ROIs, etc are likely to be

important. This needs to be reported. Furthermore, the effect of neglecting the glomeruli that did not pass the criteria (in other words, the effect of noise) on the classification should be investigated.

5. The text should be proof read. There were so many typos, grammatical errors etc.

Minor comments

6. How did animal and model performance differ depending on the identity of target, non-target, and novel background odors? Did neural responses to these odors explain the behavior? (All the figures lump the results for all the odors)

7. Line 183. "Surprisingly, there was no increase in performance even when WT mice recognized the novel background odor,...". Please present the data showing the relationship between the sniff and the animal's performance.

8. Line 273, 303 etc. How come only recurrent computation is discussed? For example, temporal integration can equally be possible.

9. Line 355. "..., showing that adaptation was not necessary in novel background odors." The statement should be softened as adaptation was not confirmed using imaging.

10. Line 360. "Cntnap2^{-/-} mice might rely more on sniffing induced suppression as a mechanism to identify odors in novel backgrounds". What is the rationale behind this?

11. Line 367. "WT mice might achieve faster reaction times using glomeruli with higher affinity for the presented odors that might be located in the ventral surface of the olfactory bulb". Again, what is the rationale behind this statement? In general, there are too much speculative descriptions in the Discussion.

12. Line 378-384. If the olfactory tubercle was performing linear computations in case of discriminating known odor mixtures, what is the possible mechanism that allows the switching of processing between the tubercle and the piriform cortex? Of the authors were to touch upon the involvement of the tubercle, it might be useful to describe some hypotheses.

REVIEWER**COMMENTS**

Reviewer #1 (Remarks to the Author):

In this manuscript Otazu and colleagues develop a new olfactory behavioral paradigm (background / foreground separation with new distractor backgrounds) and show that while WT mice readily learn and generalize, one mouse model of autism struggles. I find the paradigm relevant and well studied, developed and described and while the autism model is only a very small part of the paper I can see the relevance of the paper for the autism research community. Overall my main issues with the paper are regarding presentation in the text, description in the methods, references, discussion as well as some related data analysis issues. We thank the reviewer for the encouraging comments and for the multiple very insightful suggestions. We have addressed them with new analysis and experiments, as well as by clarifying the data analysis using new figures. Briefly, we have substantially expanded the number of imaged animals (both WT and *Cntnap2*^{-/-} mice) and performed a more rigorous analysis of the intrinsic imaging signal. We have also performed new behavioral experiments in *Cntnap2*^{-/-} mice and WT mice showing that the novelty of the background odor is the crucial element that causes olfactory deficits in this mouse model of autism.

They show that one linear classifier methods (SVM) doesn't perform as well as the animals. However, given the number of parameters, priors etc, it is a bold claim to conclude that linear classifiers in general can't perform. The authors need to broaden the number of linear classifier methods they test to comfortably make this claim.

We thank the reviewer for this comment for it has helped clarify our claim. In the revised version of the manuscript we have added more analysis and new figures We have separated the imaging analysis in 3 parts.

In the first part of the image analysis, we have tested whether linear classifiers and the nearest neighbor classifier could generalize based on data from the training set and perform well on the test set. We have substantially expanded the number of linear classifiers tested. We now include not only an SVM regressor but we have added a different linear classifier, a logistic regressor (see section **Linear classifiers and nearest neighbor classifier could identify target odors in novel background odors** on page 7 and see revised **Figure 3**). The classifiers produced very similar performances.

Figure 3 legend. Linear classifiers and nearest neighbor classifiers could identify odors in novel environments. **A.** Example of the average glomerular responses of a WT mouse to the 16 mixtures of the training set and responses to 16 mixtures of the test set with 4-methylanisole as the novel background. **B.** Estimated weights of SVM with linear kernel and logistic linear classifiers trained using the training set for the example. **C.** Output produced by multiplying the weights of the logistic linear classifier with glomeruli activation from the odor mixtures of the training set (blue squares) and the test set (red squares) for the example. **D.** Performance of the logistic regressor calculated on data from 6 WT mice, 32 recording sessions on the test set, with the performance of each recording day using 100 instantiations of each of the responses of the test set using equation 1. Error bars are s.e.m. and p-values were calculated using a t-test. **E.** Same as **C** but using the SVM weights. **F.** Performance of the SVM for the data from 6 WT mice, 32 recording sessions. **G.** Example of a matrix of dot products between the training set and the test set used for calculating the nearest neighbor classifier (NNC). The red squares indicate the location of the most similar mixture from the training set for each mixture in the test set. If the valence (go or no-go) of the target odor in the test mixture matched the valence of the most similar mixture (Nearest Neighbor), the trial was considered correct. **H.** The performance of the NNC for novel background odors for 6 WT mice, 32 recording sessions. Error bars are s.e.m.

We have also added a sparseness constraint to explore how the linear classifiers performance changed as we reduced the number of glomeruli that were included (see Fig 4). We also varied the number of glomeruli used by the nearest neighbor classifier based on the stimulus selectivity of the training set. We found that the performance of the classifiers, both linear and NNC reached a plateau at around 24-36 glomeruli (see section **Plateau performance is achieved with a small fraction of the glomeruli** on page 7 and see **Fig. 4F**).

Figure 4. Linear classifiers and the NNC did not require the use of all available glomeruli to reach plateau performance. **A.** Example of regressor weights for the linear SVM as the sparseness constrain is changed. **B.** The SVM regressor weights were applied to test set mixtures where methyl pyruvate was the novel background odor. **C-D.** Logistic regressor weights were also calculated from the training set and were applied to the test set. **E.** Number of glomeruli used in the NNC were changed by thresholding based on the auROC of the training set of individual glomeruli. Similarity matrices changed as the threshold was changed. The red squares indicate the best match to the training set. **F.** Performances of the linear classifiers and the NNC as a function of the number of ROI included calculated for 32 recording sessions, 6 WT mice. Vertical error bars are the s.e.m of the performance of the classifiers and horizontal error bar are the s.em for the number of glomeruli used for the classifiers.

In the second part of the image analysis, we have tested whether a number of linear classifiers could be used to determine the WT mice performance on individual novel background odors. Although the linear classifiers with as few as 24 glomeruli were sufficient to reach a plateau in performance, they were not enough to have a significant correlation with WT mice performance on individual novel background odors. The nearest neighbor classifier (NNC) was a better predictor of the animal behavior than the linear classifiers (see section **Classifiers that included more glomeruli had higher correlation with animal behavior** on page 12 and see **Figure 8G**).

Figure 8. G. Classifier's correlation with behavior as a function of the number of ROI included. Error bars represent the standard deviation of the distribution of linear correlation coefficients (vertical) and number of glomeruli (horizontal) included calculated using a Montecarlo simulation with 500 repeats. The asterisks indicate the results of t-test comparing the distribution of correlations using all available glomeruli against the distribution of correlations using the reduced number of glomeruli.

In the third part of the image analysis, we have tested the performance of the linear classifiers and the NNC when we reduced the diversity of the training examples using odor mixtures. WT mice solved this task at levels that matched the performance of WT mice that were trained with the full set of training examples. In contrast, all the classifiers set tested fell to chance levels. WT mice can perform a more robust algorithm than the linear classifiers that better generalizes to background odors and odor combinations that WT mice had not been previously exposed (see section **WT mice performance was less sensitive to diversity of training data than the NNC** on page 13 and **Figure 9C**).

Figure 9C. Performance using imaging data of linear classifiers and NNC, when trained with the reduced training set and tested with the reduced test set were compared to the performance when trained with the full training set and tested with the full test set. Imaging data was obtained from 32 recording sessions using real mixtures (6 WT mice, 161.3 ± 39.8 glom per session). Classifiers were created using the average training set (full and reduced) and the performance was calculated for each recording session by generating 100 instantiations of the test set (full and reduced). Comparison between algorithm performances trained using the full and reduced, as well as comparisons of the reduced set performance against chance level were done using a *t*-test. Performance of a new cohort of three WT mice that were also trained with the reduced training set and tested with the reduced test set in the asynchronous task was compared to four WT mice that had trained with the full set and were evaluated with the full set. The *p*-value for the behavior comparison was calculated using the Fisher exact test.

2) The effect of feedback inhibition on odour identification in a novel background is mentioned many times throughout the paper. E.g. the authors speculate that reduced inhibition causes impaired odour identification in *Cntnap2*^{-/-} mice but the paper clearly falls short in providing evidence for this. Considering that the senior authors has injected Muscimol into PCx in previous work it should be feasible to directly experimentally test this. If this is too much work at this point in time, the authors need to substantially reduce these mechanistic speculations.

In the revised version of the manuscript the role of feedback inhibition is briefly mentioned as one of multiple plausible hypothesis that might underlie the odor recognition deficit in *Cntnap2*^{-/-} mice. We have removed mentions of cortical feedback and the role of inhibition from the introduction and results. We present this hypothesis among others in the discussion as follows:

Page 23, lines 1006-1028: "*Cntnap2*^{-/-} mice have multiple circuit deficits that might affect their capability to detect odors in the presence of strong novel background odors. The lack of *Cntnap2* causes broadening of the transmitted action potential resulting in enlarged neurotransmitter release (Scott et al. 2019) which might cause excessive activation by strong background odors,

masking the target odors. However, $Cntnap2^{-/-}$ mice differences in performance with respect to WT mice on novel background odors only became apparent when novel background odors were presented as catch trials among trials with other well-rehearsed background odors and not when $Cntnap2^{-/-}$ mice trained over days with the same background odor. The increased release of neurotransmitters would affect both situations when a background odor was novel and when it was a standard background odor, whereas the deficit in $Cntnap2^{-/-}$ mice was selective for novel background odors.

$Cntnap2^{-/-}$ mice could be using individual glomeruli or linear classifiers to identify target odors in known background odors, as these algorithms reached performances that matched $Cntnap2^{-/-}$ mice behavior for known background odors but did not produce good behavior for novel background odors, similar to $Cntnap2^{-/-}$ mice. These algorithms do not require inhibitory activity and its performance would not be affected in $Cntnap2^{-/-}$ mice because of their reduced expression of GABA (Peñagarikano et al. 2011). In contrast, deconvolution algorithms that approximate the Lasso behavior require computations (Koulakov and Rinberg 2011; Grabska-Barwińska et al. 2017; Li and Hertz 2000; Otazu and Leibold 2011) including normalization (Carandini and Heeger 2012) which might make them more likely to be affected by reduced inhibition. Changes in cortical spine development (Anderson et al. 2012) and reduced local spine density in cortex (Lazaro et al. 2019) in $Cntnap2^{-/-}$ mice might also affect the capability to integrate information over multiple glomeruli which is necessary to identify odors in novel background odors. ”

3) Considering that 90% of the paper consists of data from WT animals, the emphasis on the mouse model of autism in both title and abstract is too strong for my taste. My suggestion would be to either rephrase to a more descriptive version that better represents the data from WT mice or provide more data in $Cntnap2^{-/-}$ mice to justify the emphasis.

We thank the reviewer for this comment. It is indeed a major push for this manuscript to describe and characterize a novel olfactory detection task, where WT mice would break the limits imposed by linear classifiers and have performance that match deconvolution methods and test it on mouse models of autism. WT mice have been previously shown to detect target odors in the presence of background odors (Rokni et al. 2014) and there has been interesting theoretical work developing models for odor recognition that could be applied for complex olfactory scenes (Koulakov and Rinberg 2011; Grabska-Barwińska et al. 2017; Li and Hertz 2000; Otazu and Leibold 2011). However, experimental work showed that WT mice performance could be explained by linear classifiers and did not require deconvolution methods (Mathis et al. 2016). By applying our novel behavior to $Cntnap2^{-/-}$ mice, we were able to detect an olfactory deficit in $Cntnap2^{-/-}$ mice that was not apparent from differences in glomerular responses (see **Figure 10 and 11**), nor from learning rates (see **Figure 12D-F**). The deficit could be overcome by training with a background odor (see **Figure 12C**). $Cntnap2^{-/-}$ mice were considered so far to have olfactory behavior that matched or exceeded WT mice in finding buried treats (Peñagarikano et al. 2011).

To further clarify the role of novelty in the olfactory deficits in the $Cntnap2^{-/-}$ mouse model of autism we have substantially expanded the experiment and analysis of the $Cntnap2^{-/-}$ mice, as follows:

- a) We have increased the number of WT and $Cntnap2^{-/-}$ mice used for imaging, to perform a more thorough comparison of the glomerular responses of the $Cntnap2^{-/-}$ mice with WT mice. We show that the magnitude of the glomerular responses (both in terms of

average responses and fraction of activated glomeruli per odor) in WT mice are not significantly weaker in the *Cntnap2*^{-/-} mice (see page 15, lines 657-673 and see Figure 10 D-G).

Figure 10. **D.** Average activation of the glomeruli produced by individual odors for the 5 *Cntnap2*^{-/-} mice plotted against average activation for 5 WT mice. **E.** Fraction of glomeruli activated per odor for the 5 *Cntnap2*^{-/-} mice plotted against fraction of glomeruli activated in 5 WT mice. Blue line represents the identity line. **F.** Average activation of glomeruli per odor for 5 *Cntnap2*^{-/-} mice and 5 WT mice. Each symbol is the average activation of one mouse. **G.** Fraction of glomeruli activated per odor.

b) We have analyzed the trial to trial variability of the WT mice and the *Cntnap2*^{-/-} mice (see page 3, line 103:131, **Methods: Measurement of trial to trial variability in WT mice** on page 32, and **Methods: Measurement of trial to trial variability in *Cntnap2*^{-/-} mice** on page 34 to see how the variability was estimated). Consistent with a previous report (M. Geramita and Urban 2017), there was increased trial-to-trial variability in the *Cntnap2*^{-/-} mice compared to WT mice (see Figure 1G-H and Figure 10B-C). The variability was also higher in the *Cntnap2*^{-/-} mice after subtracting the common variability across glomeruli (see **Methods: Measurement of uncorrelated component of trial to trial variability in WT mice** on page 33, **Methods: Measurement of uncorrelated component of trial to trial variability in *Cntnap2*^{-/-} mice** on page 34, and see Supplementary Figure 11).

Figure 1. G. Single trial responses for individual odors. **H.** Average z-score indicating the periods that were used to quantify the odor response. The air baseline period is also indicated as well as the z-score threshold (-0.46) used to detect glomerular responses. **Figure 10 B.** Examples responses as z-scores from individual glomeruli of a *Cntnap2^{-/-}* mouse on individual odor presentations and as averages. **Figure 10 C.** Average response per odor plotted against the standard deviation calculated over trials for 5 *Cntnap2^{-/-}* mice (red dots) and 5 WT mice (black dots). The magenta lines are the fitted functions for both genotypes used to estimate the coefficient of variation and imaging noise levels.

Supplementary Figure 11

Intrinsic glomerular responses in awake Cntnap2^{-/-} mice had correlated fluctuations. A. Example of an image of average z-score responses to 22 presentations of ethyl butyrate (0.1%) in an awake *Cntnap2^{-/-}* mouse. The odor was presented as a 9 second pulse. Odor responses are the average z-score calculated using the period between 2 and 9 seconds following odor onset and using the 5 seconds preceding the onset of the odor as the baseline. **B.** Example of trial by trial correlated odor evoked responses of two glomeruli located on different spots. **C.** Glomerular responses for 132 glomeruli for 2 different odor mixture presentations plotted against the average response over 22 odor presentations. Solid lines are the least square linear fit. **D.** We calculated the deviations from the best fitted line for 2980 ROI-odor pairs recorded from 5 *Cntnap2^{-/-}* mice and plotted in red the standard deviations of their deviation from the best fitted line (σ_{uncorr}) against the average response for that ROI-odor pair (μ). The black dots correspond to data from the WT mice. *Cntnap2^{-/-}* mice had higher uncorrelated trial-to-trial variability compared to WT mice.

c) We have expanded the number of WT mice (5 mice) and *Cntnap2*^{-/-} mice (5 mice) used for imaging. The larger number of animals of both genotypes has permitted us to make comparisons between correlation coefficients of odor similarities between individual animals. Briefly, odor pairs that were similar for WT mice were also similar for *Cntnap2*^{-/-} mice. As suggested by the reviewer, we have compared correlation coefficients within genotypes and across genotypes. Individual mice odor similarity coefficient of correlation were equally correlated when compared within genotype as when compared across genotypes (see page 16, lines 674-699 and see Figure 11 D-G).

Figure 11. Pairs of odors that evoked similar activity patterns in WT mice also evoked similar patterns in *Cntnap2*^{-/-} mice. **D.** Patterns of glomerular activation using ROI from example individual animals of each genotype. **E.** Odor similarity matrices for the example *Cntnap2*^{-/-} mouse and WT mouse. **F.** Similarity between 190 odor pairs calculated using the example WT mouse data against the similarity from the example *Cntnap2*^{-/-} mouse. **G.** Distribution of linear correlation coefficients of odor similarities (*r*) for pairs of animals of the same genotype (WT vs WT, 10 animal pairs and *Cntnap2*^{-/-} vs *Cntnap2*^{-/-} 10 animal pairs) and for pairs of animals of different genotypes (WT vs *Cntnap2*^{-/-}, 25 animal pairs). Error bars represent the mean ± s.e.m. Comparison between correlation coefficients were done using a *t*-test.

d) We have compared the performances of the linear classifiers and the NNC using the glomerular data of the *Cntnap2*^{-/-} and the WT mice. The increased trial-to-trial variability of the *Cntnap2*^{-/-} mice glomerular responses was not large enough to affect the performance of the linear classifiers, NNC, nor the Lasso deconvolution (see **Increased trial-to-trial variability in glomerular activity in *Cntnap2*^{-/-} did not significantly reduce performance of algorithms** on page 16 and see Figure 11 H-I).

Figure 11. H. Performance (mean±s.e.m) of SVM, logistic, and NNC classifiers calculated using *Cntnap2*^{-/-} and WT mice glomerular activation data for target detection in novel environments trained with the full training set and tested with the full training set. Symbols represent average performance per animal. **I.** Performance (mean±s.e.m) of the Lasso using *Cntnap2*^{-/-} and WT mice glomerular data for the reduced test set using different sizes of dictionaries.

e) Although our imaging results indicate that odor identification in novel backgrounds should have been very similar between *Cntnap2*^{-/-} mice and WT mice, there was a significant behavioral deficit in the *Cntnap2*^{-/-} mice. We have directly tested that novelty of the background was the key element that affected performance in *Cntnap2*^{-/-} mice. We performed new experiments in a new cohort of *Cntnap2*^{-/-} mice and WT mice. We show that when a novel background odor is introduced gradually, performance is similar between the *Cntnap2*^{-/-} mice and the WT animals. However, the same background odor produced a much lower performance in the *Cntnap2*^{-/-} mice compared to WT mice when the background odor was introduced suddenly at a higher concentration as a novel background odor. This new experiment shows that it was not only the amount of glomerular representation overlap between a background odor and the target that determined the difficulty of a task for *Cntnap2*^{-/-} mice (see **Fig. 12 C** and **page 18, line 759-786**).

Figure 12. C. Performance of butyl propionate when it was used as a novel and as a standard background for different cohorts of *Cntnap2*^{-/-} mice and WT mice. The performance of four *Cntnap2*^{-/-} mice performance was at chance level when butyl propionate was a novel background odor, but significantly increased for the two

Cntnap2^{-/-} mice where butyl propionate was used instead of (s)-(-)-limonene as a standard background during training.

- f) We have also further analyzed the learning rates during training of the *Cntnap2*^{-/-} mice and compared them to WT mice. We compared the learning rates when *Cntnap2*^{-/-} and WT mice learned to discriminate the target odors without backgrounds (see Figure 12D). We also compared the learning rates when target odors were presented for the first time together with low concentration background odors (see Figure 12E), and when they learned to discriminate the target with the backgrounds at high concentration (see Figure 12F). Learning rates were not different between *Cntnap2*^{-/-} and WT mice (see section **Cntnap2**^{-/-} mice odor learning rate is similar to WT mice on page 18).

Figure 12. D. Learning curves for the second pair of go target and no-go target at a high concentration (0.1%) after animals had already learned to discriminate the first pair of odors at that concentration. **E.** Learning curves for the first exposure of the standard background odors at 0.025% (1/5 of the final concentration) with the targets at their final concentration (0.025%). **F.** Learning curves for the first exposure of the standard background odors at the final concentration of the background odors (0.1%).

- 4) L 113 Why do novel backgrounds result in lower glomerular activation? Or why were they chosen as such? This seems to make target detection easier?

The reviewer is right that the novel background odors had slightly lower glomerular activation than the standard background odors. We chose the novel background odors to be not too strong, such as their difficulty as backgrounds would be dominated by their novelty and not by the strength of their glomerular representations (see page 5, lines 150-160).

- 5) Some of their regression fits, e.g. in 5D ($\rho=1.1$) look odd; how much does the fit depend on a few outliers (that might put the underlying assumption of normality to determine significance into question).

The correlation of the performance of the algorithms using imaging with the performance of the individual odors using artificial mixtures resulted in lower correlations with the behavior. Therefore, we have removed correlations between behavior and performance of the algorithms with artificial mixtures from the manuscript, as real mixtures better reflect the challenges that animals have to solve during the behavior (see **Fig. 8A-F** below).

•6) Why does the linear classifier performance drop so much when they use the real data (5G) vs. their pseudo-data (5B)?

Figure 5B from the original submission overestimates the performance of the linear classifier. First, there was no noise ($\rho=0$). Second, artificial mixtures include only saturation, whereas real mixtures include all the nonlinearities produced by the interactions between the 3 odors used. In the revised version of the manuscript, we have increased the number of animals and recording sessions for obtaining glomerular responses from real mixtures (32 recording sessions from 6 WT mice (161.3 ± 39.8 glom per session, mean \pm s.d, 5163 glomeruli). The correlation with animal behavior is higher for the real mixtures, which are the ones that animals have to solve during the behavior. Therefore, we have removed the correlation with the behavior using artificial mixtures from the manuscript (see **Classifiers that included more glomeruli had higher correlation with animal behavior** on page 12 and see **Fig. 8A-F**).

- Synchronous (4 WT mice)
- Asynchronous (4 WT mice)

Figure 8. Classifiers that included more glomeruli had higher correlation with animal behavior. **A.** Performance of the NNC for 10 novel background odors was calculated using all the available glomeruli per imaging session (161 ± 40 ROI, mean \pm s.d, 32 imaging sessions, 6 WT mice). Black circles y coordinates represent the average behavioral performance on a novel background odor from four WT mice that performed the asynchronous task and red circles y coordinate represent the average performance of the four WT mice that performed the synchronous task. The blue line is the linear correlation between imaging data and behavior. **B-C** Similar plot for logistic regression and SVM. **D-F.** Reducing the number of glomeruli reduced correlation with behavior for all three classifiers.

7) L 479 "The first one was to determine if the Lasso assigned a larger contribution to the correct target (go or no-go) than to the incorrect target. If the Lasso estimate did so, this was considered a correct discrimination." I'm not sure what they're doing here. First, what is the incorrect target? Second the Lasso is not a discrimination method, it's a sparse recovery method: you stick in glomerular activations, and it spits out an odour concentration vector. It's agnostic to what you want to do with the odour vector (whether it's correct or not). This is another example of their computational methods being *very* unclear.

We apologize to the reviewer for the lack of clear explanation. The reviewer is right that contrary to the linear classifier, and the nearest neighbor classifier, the reconstruction algorithms determine an estimate of the concentration of odors from a pool of odors that is known to the animal. To convert the odor concentration vector (see page 14, lines 607-619 and Methods: **Lasso deconvolution comparison with NNC, linear SVM, and logistic regression** on page 32) into a go/no-go readout that could be evaluated as correct or incorrect similar to the behavioral evaluation, we compared the values of the estimated odor concentration vector assigned to the go odors with the values assigned for the no-go odors. If the maximum of the two concentrations assigned to the two go odors was larger/smaller than the maximum of the concentrations assigned to the two no-go odors, we considered that the Lasso produced a go/no-go readout. If the Lasso readout (go or no-go) matched the label of the target odor present in the input mixture, the Lasso readout was considered correct. We have added a figure (see **Figure 9E**) and more explanation in the figure caption.

Figure 9E. Concentrations assigned by the Lasso for a dictionary size of 500 elements. Bars correspond to the estimated concentrations of the four contextual background odors, the four target odors, and (s)-(-)-limonene. Stems correspond to estimated concentrations for the randomly generated dictionary elements. The maximum of the estimated concentration was

larger for the two target go odors compared to the two target no-go odors, so this would be considered a correct identification.

8) L 293: Sparse deconvolution algorithms "use the training data more efficiently than the NNC"; this is incorrect, in that at least the two algorithms he cites which I am aware of (Koulakov 2011 and Grabska Barwinska 2017), do not have a training step, and are provided at the start with the correct dictionary.

We have corrected our statement as follows (page 14, lines 594-598):

"These algorithms are more complex than NNC and linear classifiers. However, these sparse deconvolution algorithms offer the advantage over the NNC and linear classifiers that once an animal learns a dictionary, an animal can apply it for multiple combinations of odors, whereas the NNC and linear classifier performance depends on the diversity of the training examples."

•9) The methods, particularly the computational methods, weren't clear. For example, I had to infer from the discussion that they used 7-second averages of the imaging signal as inputs to their classifiers, but I couldn't find this in the actual methods.

We apologize to the reviewer. In the revised version of the manuscript we have expanded the explanation of the methods. We provide more details, including additional individual examples of calculations for the linear classifiers as well as for the nearest neighbor classifier (see **Figures 3 and 4** on the answer to point 1). We have also added examples of single trial responses and included a diagram that indicates the period of time when the baseline air response was measured as well as when the odor evoked response was measured (see **Fig. 1 G-H**). We quantified the odor evoked responses as z-scores using the 5 second air interval preceding the odor presentation as baseline. The average odor response was calculated as the mean value, averaged over repeats, of the z-score during a 7 second window that started 2 s after odor onset.

Figure 1G. Single trial responses for individual odors. **H.** Average z-score indicating the periods that were used to quantify the odor response. The air baseline period is also indicated as well as the z-score threshold (-0.46) used to detect glomerular responses.

We have also added figures explaining how the nearest neighbor classifier was calculated (see Fig. 3G).

Figure 3 G. Example of a matrix of dot products between the 16 training set mixtures and the 16 test set mixtures used for calculating the nearest neighbor classifier (NNC). Each target odor appeared mixed with each of the four contextual background odors. For each mixture in the test set, the red squares indicate the location of the most similar mixture from the training set. If the valence (go or no-go) of the target odor in the test mixture matched the valence of the most similar mixture (Nearest Neighbor), the trial was considered correct.

10) Also, how many glomeruli did they use? They say (in the discussion I think) that it's ~200, but they should state this in the methods

Most of the imaging analysis is based on data from 32 recording sessions from 6 WT mice, with 161.3 ± 39.8 glomeruli per session (mean \pm s.d.), 5163 glomeruli total, where we recorded from mixtures that included the training set and the test set with novel background odors. We also imaged responses from individual odors from 5 WT mice with 155 ± 38.1 (mean \pm s.d.) glomeruli recorded per animal (775 ROI total). We recorded from 5 awake *Cntnap2*^{-/-} mice with 128.6 ± 19.3 glomeruli (mean \pm s.d.) recorded per session with 643 glomeruli total.

11) The argument that autism mouse models struggle with novel odours on a computational level only based on the observation that they avoid environments with novel background odours seems far-fetched.

We have moved the topic to the discussion on how not being able to detect targets in novel background odors might affect mouse models of autism behavior in more ethological relevant situations (**page 23, line 1028-1042**).

12) I find it a very interesting observation that reaction time / number of sniffs very quickly fell back to trials with known background. In the light of odour novelty and sniffing they really should cite Esquivelzeta Rabell...Haesler, 2017, Current Biology.

We have added the citation to Esquivelzeta et al 2017 (page 9, lines 388-408) and we mention that behavioral reaction to novel odors adapts quickly and includes nostril movements besides increased sniffing.

13) The authors should discuss why the sniff rate increase is less pronounced in synchronous presentation condition (Figure 4E) compared to the asynchronous condition (Figure 3B).

We thank the reviewer for this observation. In the revised version of the manuscript (**page 11, lines 462-465**) we have performed statistical tests to determine whether the increase in sniff rate is less pronounced for the synchronous case compared to the asynchronous case. The increase in sniff with respect to baseline during presentation of the target for WT mice performing the synchronous behavior (0.59 ± 0.050 extra sniffs per second, 1191 trials) was significantly smaller than the increase seen in WT mice performing the asynchronous behavior (2.06 ± 0.05 extra sniffs per second, 1563 trials, 4 animals, $p = 3.31 \times 10^{-85}$, t-test). We think that the increase in sniff rate is smaller for the synchronous case because the mice doing the synchronous task were already in an enhanced attentional state, reflected in the increased baseline sniff rate (**page 11, lines 444-459**).

In the synchronous odor presentation, the target started at the stimulus onset, making it easier to be missed by the animals compared to the asynchronous case where the target started 1.5 seconds after stimulus onset. Indeed, compared to the asynchronous case, there was a small but significant increase in the number of misses for standard backgrounds in the synchronous case compared to the fraction of misses in the asynchronous case (from only 0.7% (12/1563) to 3.9% (46/1191), $p = 2.1 \times 10^{-8}$, Fisher exact test, see **Fig. 7C**). WT mice increased their baseline sniffing during synchronous odor presentation compared to the asynchronous case suggesting an enhanced level of alertness in order not to miss the target. The base rate of sniffing before odor onset was higher for WT mice that were doing the synchronous task (2.62 ± 0.04 sniffs per second, $n = 1191$ trials, $n = 4$ animals) compared to WT mice doing the asynchronous task (2.18 ± 0.03 , $n = 1563$ trials, $p = 6.9 \times 10^{-19}$, t-test, $n = 4$ WT mice). These differences were not due to differences in animal batches. The same WT mice that did the synchronous task had also significant lower baseline sniff rates (2.07 ± 0.04 sniffs per second, $n = 1209$, $n = 4$ WT mice) when they were performing the asynchronous task with known background odors during their training process, compared to when they performed the synchronous task ($p = 5.99 \times 10^{-23}$, t-test). The synchronous task seems to require increased baseline attention level given the unpredictable appearance of the target.

14) L 224: The authors should elaborate on the statement “The very simple circuit implementation of a linear classifier is less likely to be affected in mouse models of autism”.

We have clarified our statement. We meant that the linear classifier implementation only requires a population of neurons that receives projections from multiple glomeruli from the olfactory bulb. Several structures in the olfactory pathway have this characteristic.

page 7 ,lines 262-265: *“The mouse olfactory system could combine information from multiple glomeruli and implement simple supervised algorithms using the feedforward connections between the olfactory bulb and its targets, including the olfactory tubercle, anterior olfactory nucleus, and piriform cortex (Haberly and Price 1977).”*

15) I find it confusing that the authors refer to potential computational strategies in autism models using broad statements throughout the results section but only show the *Cntnap2*^{-/-} data at the very end. The impaired odour identification in novel background seems like a solid observation but, unfortunately, remains purely descriptive.

There are multiple circuit variations in the *Cntnap2*^{-/-} mice that might contribute to impairment of odor identification in novel environments. In the revised version of the manuscript, we have identified computational requirements for odor identification in novel environments based on our experimental data. This include showing that a carefully selected single glomerulus can identify a target in well-rehearsed backgrounds but cannot generalize to novel backgrounds (see section **Single glomeruli cannot reliably identify targets in novel backgrounds** on page 6), the requirement to integrate glomerular responses over >24 glomeruli (see section **Plateau performance is achieved with a small fraction of the glomeruli** on page 7) as well as showing that linear classifiers are not able to match the performance of the animals when the number of training examples is reduced (see section **WT mice performance was less sensitive to diversity of training data than the NNC** on page 13).

We have added the following note in the discussion regarding circuit deficits in *Cntnap2*^{-/-} mice that might affect odor detection in novel environments (page 23, lines 1005-1027):

“Cntnap2^{-/-} mice have multiple circuit deficits that might affect their capability to detect odors in the presence of strong novel background odors. The lack of Cntnap2 causes broadening of the transmitted action potential resulting in enlarged neurotransmitter release (Scott et al. 2019) which might cause excessive activation by strong background odors, masking the target odors. However, Cntnap2^{-/-} mice differences in performance with respect to WT mice on novel background odors became only apparent when novel background odors were presented as catch trials among trials with other well-rehearsed background odors and not when Cntnap2^{-/-} mice trained over days with the same background odor. The increased release of neurotransmitters would affect both situations when a background odor was novel and when it was a standard background odor. whereas the deficit in Cntnap2^{-/-} mice was selective for novel background odors.

Cntnap2^{-/-} mice could be using individual glomeruli or linear classifiers to identify target odors in known background odors, as these algorithms reached performances that matched Cntnap2^{-/-} mice behavior for known background odors but did not produce good behavior for novel background odors, similar to Cntnap2^{-/-} mice. These algorithms do not require inhibitory activity and its performance would not be affected in Cntnap2^{-/-} mice because of their reduced expression of GABA (Peñagarikano et al. 2011). In contrast, deconvolution algorithms that approximate the Lasso behavior require computations(Koulakov and Rinberg 2011; Grabska-Barwińska et al. 2017; Li and Hertz 2000; Otazu and Leibold 2011) including normalization(Carandini and Heeger 2012) which might make them more likely to be affected by reduced inhibition. Changes in cortical

spine development(Anderson et al. 2012) and reduced local spine density in cortex(Lazaro et al. 2019) in *Cntnap2*^{-/-} mice might also affect the capability to integrate information over multiple glomeruli which is necessary to identify odors in novel background odors.”

16) L 28-30: “Critically, the glomerular representations of mixtures with different weak targets but the same strong background odor are dominated by the background odor and are very similar.” – this is unclear – what is this based on, how is this conclusion arrived at – the authors need to expand (I agree with the conclusion but the logic needs to be clear)

We have calculated the similarities between the target go odors and the target no-go odors, as well as between the mixtures that include the target go odors and mixtures that include the target no go odors. The similarity was calculated as the normalized dot product of the glomerular representations. The go and no-go target odors had low similarity of 0.37 ± 0.08 (mean \pm s.d). In contrast the similarity between go and no-go mixtures (different valence) with standard backgrounds was high, 0.69 ± 0.13 and 0.73 ± 0.12 for mixtures with novel background odors (see Fig 1 L-Q and section **Background odors increased the similarity between odor mixtures** on page 5). These data indicate that the presence of the background odors make the stimuli that the mice need to discriminate more similar to each other.

Figure 1 L. Example of the average glomerular response of one WT mouse to the two go target odors and the two no-go target odors. **M.** Similarity matrix between the target odors. **N-P.** Examples of the odor responses for a WT mouse to the 16 mixtures used in the training set (**N**) and to the 16 test set mixtures where the novel background odor was hexanal (**P**). **O-Q.** Average similarity matrix from 6 WT mice. Mixtures with different valences became more similar compared to the target odors alone.

17) L 33 “Since novel odors produce larger mitral cell activation compared to familiar odors (Kato et al. 2012)” – Isn’t this simply due to changes in sniffing?

Response to novel odors does include increased sniffing (Jordan et al., 2018; Verhagen et al., 2007). However, there is also reduced responses in mitral cell activity as mice familiarized with an odor without change in respiration as reported by Kato et al 2012. We have added a clarification on **page 1, line 38**.

18) L 52-56 – the phrase “temporal gap” is a confusing phrase – it is rather lack of correlation in Hopfield 1991

We have corrected the citation of Hopfield 1991 in page 10, line 424-425: “*Unsupervised algorithms(Hopfield 1991) can use differences in the temporal profile between the background odor and target odor to segment them.*”

19) Please plot the performance of individual mice in e.g.Fig 2a-d

We have added the performance of individual mice on all the figures.

20) Segmented and false color would make the intrinsic imaging figures much easier to grasp

We thank the reviewer for the comment. We are showing the ROI drawn. However, we feel that it is important to report how the images would look like without an arbitrary choice of a color map. We have also added multiple examples of matrices showing the responses of the individual glomeruli to the training set and the test set (for an example see Figure 1 **N-P**).

Figure 1 N-P. Examples of the odor responses for a WT mouse to the 16 mixtures used in the training set (**N**) and 16 test set mixtures using where the novel background odor was hexanal (**P**).

21) There are multiple duplications in references – fukunaga 12a,b; igarashi 12

We have corrected the references.

22) L 243 “Could be predicted using a non-linear classifier” – that is a bit too general, almost

everything can be predicted using a non-linear classifier. They need to specify which type, which conditions etc.

The reviewer is right. In the revised version of the manuscript we introduce the nearest neighbor classifier as being a classifier that has been proposed to be implemented by the olfactory system. Then we clarify that the nearest neighbor is a nonlinear classifier.

23) L 311 “glomerular patterns were very similar” – this needs to be quantified and compared to within WT / Cntnap2 variability

We thank the reviewer for this suggestion. The within genotype variability is the right scale to compare the magnitude of the difference across genotypes. We have increased the number of imaged animals to 10 (5 WT and 5 Cntnap2 animals), as shown above on the response to point 3, part c.

•24) Please plot Individual animals in 7F,H

We have added individual animal's performances to all figures.

25) L 358 – “longer time gaps”: Szyszka gaps are in the millisecond range

The reviewer is right that for most of the honeybee literature a few milliseconds of odor asynchrony are sufficient to separate odor mixtures when the odors are familiar to the bee. Interestingly, in Sehdev and Szyszka (2019) onset asynchrony necessary to segregate unknown odorants is 5 seconds, with one second odor not being enough. Although the experimental paradigms are different, this is a novel and interesting result which suggests that mammalian olfactory system does not require the long adaptation periods for novel background odors compared to invertebrate ones. This Szyszka result is relatively new so we describe it in more detail in the discussion of the revised version of the manuscript on **page 35, line 935-942**:

“Suppression of the preceding novel background did not contribute significantly for target detection because the performance obtained on the synchronous task was almost identical to the performance on the asynchronous task. This stands in contrast with invertebrate experimental data: honeybees required 5 seconds or longer to segregate novel odors, which is two orders of magnitude longer than the time required for known odors (Sehdev and Szyszka 2019).”

26) L 365 – citing Meister 2015 for low dimensionality of odor space is somewhat misleading - the paper only makes the claim that dimensionality is unknown as the arguments of the Keller et al paper are discussed / partially debunked

We have removed the citation of Meister 2015. We are now citing the work of Slotnik and Bisulco that demonstrated the high degree of redundancy in the olfactory system by showing that extensive lesions of the olfactory bulb failed to affect odor discriminations. This paper provides more direct evidence of the redundancy of the olfactory system.

page 22, lines 950-952: *“This is consistent with glomerular representation of odors being highly redundant as revealed by lesion studies (Slotnick and Bisulco 2003).”*

•27) Figure 1E: correct to. “intertrial time”

We have corrected the figure.

28) Line 363: change “glomerular” to glomeruli

We have corrected it.

•28)There are lots of typos throughout and a thorough proofread is critical.

We apologize for those and we have fixed them.

Reviewer #2 (Remarks to the Author):

The manuscript by Li and co-workers performs an innovative study of decoding the targeted stimulus from glomerular activity in a go-no go task similar to the visual CAPTCHA used to distinguish humans from computers. In the olfactory CAPTCHA go-no go task small concentrations of target odorants (go vs. no-go) are presented in the presence of higher concentrations of known and novel background odorants. The task is well-designed and provides potentially important data indicating that olfactory processing can differentiate odorants masked within novel background stimuli by an algorithm related to the nearest neighbor classifier. In addition, the authors show that mice of the *Cntnap2*^{-/-} model of autism are not able to cope with the presence of novel backgrounds when trained with a small number of target odorants. Unfortunately my enthusiasm is significantly reduced due to poor presentation of glomerular responses to odorants that makes it impossible to judge the validity of the data, evaluation of results with questionable statistical methods and lack of a thorough consideration of published data.

We thank the reviewer for recognizing the innovative aspects of our work and for the many insightful suggestions. We have addressed these concerns regarding the intrinsic imaging data by performing new imaging experiments, and by refining the imaging analysis and statistics to better characterize glomerular responses using intrinsic imaging in *Cntnap2*^{-/-} and WT mice. We have also rewritten the paper by setting it better up in the framework of previous work.

Major comments

1. The quality of the images of glomerular activation patterns shown throughout the manuscript are poor (e.g. Fig. 1A). Furthermore, it is not stated in the legends what the gray scale intensity represents (z score referenced to no odor?) and no gray scale calibration bars are shown.

The most active regions appear to be in the periphery with few responses within raising the question whether the data are valid.

Furthermore, there is no information on how individual glomeruli respond to different odorants. A supplementary figure should be provided making clear how individual glomeruli respond to target and background odors as well as mixtures, demonstrating the trial-to-trial reliability of differential glomerular activity and showing how the average glomerular activation shown in Fig. 1C was calculated.

The reviewer is correct in that most of the strongly active glomeruli seem to be located on the periphery of the bulb. Although the more centrally located glomeruli can be strongly activated using high concentrations of odors (Vincis et al. 2012), we have avoided those high concentrations as they might be aversive for the animals during the behavior.

We have added in this figure and all the figures a legend on the scale bar indicating what is being shown, z-score referenced to no odor.

To address the concern regarding the signal to noise ratio given the low odor concentrations used, we have performed a quantitative analysis of the sources of variability in the intrinsic signal which we used to determine the threshold of glomerular activations that we could reliably detect using intrinsic imaging as well as the trial-to-trial variability in glomerular responses that might affect animal behavioral performance.

- a) Page 3, lines 103-110: “We could detect odor evoked responses on individual odor presentations using intrinsic imaging (see **Fig. 1G-H**), as previously shown (Meister and Bonhoeffer 2001). The imaged z-score response of an individual glomerulus changed from trial to trial. These variations in the imaged z-score reflect both real variations in the glomerular responses as well as imaging noise in the intrinsic signal. Variable glomerular responses could potentially affect odor recognition in mice whereas the imaging noise mostly affects our experimental capability to detect weakly activated glomeruli by an odor. Imaging noise can be reduced by averaging over multiple odor repeats whereas the mice need to make decisions based on single odor presentations.”

Figure 1 G. Single trial responses for individual odors. **H.** Average z-score indicating the periods that were used to quantify the odor response. The air baseline period is also indicated as well as the z-score threshold (-0.46) used to detect glomerular responses.

- b) Page 3, lines 111-123: “To quantify these two variability sources, we plotted the standard deviation of the trial by trial glomerular responses against the average glomerular response (see **Fig. 1I**). The standard deviation increased with larger average responses, consistent with a previous report using glomerular calcium imaging (Mathis et al. 2016) that showed that the standard deviation σ was proportional to μ , the average glomerular response, with the proportionality constant given by the coefficient of variation (CV). This model would predict zero variation for a glomerulus that was not activated by an odor. However, there was a measurable variance σ^2_{noise} in the intrinsic glomerular response even in the absence of an average odor evoked response which originated from the imaging noise. Using the data from 2684 ROI-odor pairs from 3 WT mice, we estimated the coefficient of variation CV and the σ^2_{noise} by fitting the function $\sigma^2(\mu) = \sigma^2_{noise} + CV^2\mu^2$. The estimated coefficient of variation (CV) was 0.34 (95% CI: 0.30 - 0.37) which is similar to the value of 0.37 ± 0.07 (mean \pm SD) estimated using calcium imaging in anesthetized mice (Mathis et al. 2016).”

Figure 1 I. Average odor response versus trial-to-trial variability. Trial-to-trial variability was the combination of a component that scaled with the average odor response plus a constant. Purple line indicates mean fitted function and dotted lines are the 95% confidence intervals of the fit.

c) Page 3, lines 123-129: “This coefficient of variation includes both uncorrelated fluctuations of individual glomerular responses as well as fluctuations that are correlated across the whole glomerular population. WT mice performance might be mostly limited by the uncorrelated variability because, by having access to all set of glomeruli, mice could compensate for the correlated fluctuation of the whole population(Mathis et al. 2016). Therefore, we calculated a coefficient of variation CV_{uncorr} for the uncorrelated fluctuations (see **Supp. Fig. 3**) after subtracting the population response fluctuations. CV_{uncorr} was 0.25 (95% CI: 0.23 - 0.27).

WT

Supplementary Figure 3. Intrinsic glomerular response in awake WT mice had correlated fluctuations. **A.** Example of an image of average z-score responses to 27 presentations of a mixture of ethyl caproate (0.1%), (s)-(-)-limonene (0.1%), and isobutyl propionate (0.025%) in an awake WT mouse. The odor was presented as a 9 second pulse. Odor responses are the average z-score calculated using the period between 2 and 9 seconds following odor onset and using the 5 seconds preceding the onset of the odor as the baseline. **B.** Example of trial by trial correlated odor evoked responses of two glomeruli. Linear correlation was significant ($R=0.66$, $p=0.0008$, linear correlation coefficient). **C.** Glomerular responses for 156 glomeruli for 2 different odor mixture presentations plotted against the average response over 27 odor

presentations. Solid lines are the least square linear fits. Glomerular odor response variations had population correlated fluctuation and an uncorrelated fluctuation. The population correlated fluctuation is given by the least squares linear fits, whereas deviation from the least squares linear fits correspond to the uncorrelated fluctuation. **D.** We calculated the deviations from the least squares fitted line for 2684 ROI-odor pairs recorded from 3 WT mice and plotted the standard deviations of their deviation from the best fitted line (σ_{uncorr}) against the average response for that ROI-odor pair (μ). The total variance σ_{uncorr}^2 is given by the sum of an odor independent variance $\sigma_{uncorr\ noise}^2$ and a response dependent variance. The purple line is the fitted line and the dotted lines are the 95% confidence interval of the fit.

- d) In addition to the characterization of the noise in our intrinsic responses, we have also compared the responses obtained using intrinsic imaging in a Thy1-GCaMP6f mice, where GCaMP6f is expressed in mitral and tufted cells. The locations of the responses of the glomeruli measured with intrinsic imaging matched the fluorescent response. This shows that signals from weakly activated glomeruli, determined from intrinsic imaging, were able to propagate to the output stage of the olfactory bulb (**Supplementary Figure 4**).

Supplementary Figure 4

Glomerular activation measured with intrinsic signal correlate with increases in fluorescence in Thy1-GCaMP6f mouse. We implanted a C57BL/6J-Tg(Thy1-GCaMP6f) GP5.11Dkim/J (Dana et al. 2014) mouse with a window over the olfactory bulb as described above. **A**. We presented 20 odors using 9 second odor pulses at the concentrations used for the behavior in an awake mouse and recorded intrinsic signal using 780 nm infrared light. **B**. We also used a 470 nm blue light and recorded the green fluorescent responses that correspond to mitral and tufted cells, the output neurons of the olfactory bulb. **C,D**. Examples of average z-score of the intrinsic imaging averaged

over 2 to 9 seconds from odor onset. The negative deflections in odor evoked glomerular responses in the intrinsic signal colocalized with odor evoked positive deflections in the fluorescence signal. The fluorescent signal had larger magnitude compared to the intrinsic signal. **E.** Examples of time course responses from individual glomeruli to 9 second odor pulses that start at 0 s. Even intrinsic signals with relatively small z-scores (see ROI 46 in response to ethyl propionate) resulted in increases in the fluorescence signal. **F.** Relationship between the ROI-odor intrinsic signal, defined in the intrinsic signal, and the fluorescence signal. Black dots represent individual ROI-odor pairs and circles are the mean of data binned according to the z-score of the intrinsic imaging. The error bars are the standard deviation of the fluorescence signal of the ROI-odor pair in a bin. We compared the fluorescence signal of the binned data according to the z-score with the distribution of fluorescence in the absence of odor responses (noise distribution). The responses were significantly different from the noise distribution for responses with a mean z-score response of intrinsic imaging of -0.3 ($p < 0.02$) or larger.

We have show the average glomerular responses to the different targets and backgrounds (see **page 4, lines 150-165** and see **Figure 1K**). We also show the fraction of glomeruli that responded to the targets and backgrounds (see **Figure 1J**). We have added symbols for each individual WT mice (5 mice total).

We quantified the odor evoked responses as z-scores using the 5 second air interval preceding the odor presentation as baseline (see **page 3, lines 94-97**). The average odor response was calculated as the mean value, averaged over repeats, of the z-score during a 7 second window that started 2 s after odor onset. **Figure 1G-H** (see above) shows the different periods that were used to calculate the z-score responses.

Figure 1 A. J. Average fraction of glomeruli activated by odors. The error bars are 95% confidence interval. Symbols correspond to individual WT mice. **K.** Average z-score response to an odor. Error bars are s.e.m.

We have also analyzed the degree of overlap between the target and the background because target detection in backgrounds depends on the overlap between the background and the targets (Rokni et al. 2014). Page 4, line 167-179 “We assessed the overlap between the targets and the backgrounds by determining the fraction of the significantly activated glomeruli for a target odor that were also significantly activated by a background odor. A large fraction of the glomeruli that responded to a target also responded to individual contextual backgrounds ($47.8 \pm 8.3\%$, mean \pm s.d., 16 target-background combinations), with $85.6 \pm 3.4\%$ (mean \pm s.d., 4 target odors) of the target responding glomeruli also responding to at least one of the contextual background odors. A large fraction of the target responding glomeruli also responded to individual novel background odors ($51.4 \pm 10.6\%$, 44 target-background combinations), with $96.9 \pm 2.1\%$

(mean±s.d., four target odors) of the target activated glomeruli responding to at least one of the novel background odors. Almost all the glomeruli that responded to the targets (99.4±6%, mean±s.d., 4 targets) would also respond to at least one of the contextual or novel background odor. The large overlap between the target representation and the background representation suggests we created a relatively difficult task caused by the background odors.”

The description of the methods for intrinsic optical image acquisition and analysis is poor. What is the criterion for activation of a glomerulus?

The revised version of the manuscript includes a more detailed description in the methods of the optical imaging acquisition and analysis (see **Methods: Imaging on page 31, Image analysis on page 32, Measurement of trial to trial variability in WT mice on page 32, and Measurement of uncorrelated component of trial to trial variability in WT mice on page 33**). We determined the threshold for detection of glomerular activation based on the imaging noise and the number of odor exposures. The average activation produced by the odors was larger than the calculated z-score threshold.

Page 4, lines 132-138: “The estimated trial to trial variability associated to the imaging noise σ_{noise} was 1.59 z-score (95% CI: 1.51 - 1.66). We averaged responses over n trials, so this noise would appear as a standard error of the mean around zero of σ_{noise}/\sqrt{n} . This s.e.m determines our threshold to detect mean glomeruli activation that was reliably different from zero. We used at least $n=16$ trials so the standard error of the mean was <0.42 . Glomeruli-odor responses that had average z score of -0.42 or larger were considered non-responsive. This threshold was smaller than the responsive ROI-odor responses (-1.37 ± 1.17 z-score, mean±s.d., $n=6477$ ROI-odor pairs, 5 WT mice). “

The separation between the threshold and the activation of the responsive glomeruli was large enough that perturbations of the z-score threshold used did not affect the performance of the linear classifiers and the NNC (see **Supplementary Figure 6**).

Supplementary Figure 6

Performance of linear classifiers and NNC for odor identification in novel background odors were robust to changes in the threshold of glomerular responses using intrinsic imaging. **A.** Average number of glomeruli-odor responses included per recording session as a function of the threshold of z-score responses. Error bars are the s.e.m for 32 recording sessions (6 WT mice, 5163 ROI). The number of ROI-odor responses fluctuated between -16% and +17% as the threshold for z-

score responses was changed between -0.18 (most permissive) to -0.66 (most restrictive). The reference z-score threshold was -0.42 which resulted in 72885 odor-ROI responses. **B.** Performance on individual recording sessions of the linear classifiers and NNC as a function of the z-score threshold. Error bars are s.e.m. The performance of the NNC, and the linear SVM and logistic regression were not affected by changes in the threshold of detection of the z-score.

2. Throughout the manuscript there is lack of detail on experimental details and statistics. For example, for Fig. 1C it is only stated that : "C. Average glomerular activation". It is not stated what the error bars are, how many animals were used and how many measurements per mouse were performed.

In the revised version of figure 1, we state in the figure caption that the error bars for average glomerular activation are the standard error of the mean. We have also added individual symbols for each of the 5 WT mice used to generate this figure. The minimal number of odor repetitions was 16. For all the figures, a description of the error bars used is stated in the figure captions. Most of the imaging analysis is based on data from 32 recording sessions from 6 WT mice, with 161.3 ± 39.8 glomeruli per session (mean \pm s.d), 5163 glomeruli total, where we recorded from mixtures that included the training set and the test set with novel background odors. We also imaged responses from individual odors from 5 WT mice with 155 ± 38.1 (mean \pm s.d.) glomeruli recorded per animal (775 ROI total). We recorded from 5 awake *Cntnap2*^{-/-} mice with 128.6 ± 19.3 glomeruli (mean \pm s.d.) recorded per session with 643 glomeruli total.

There is no indication whether the different bars are statistically significantly different. No multivariate statistical analysis or correction for multiple comparison is presented when necessary. For every statistical test please give the p value, the number of samples and the number of animals. For statistical tests was there correction for multiple comparisons?

We apologize to the reviewer. In the revised version of the manuscript all the figures have symbols indicating whether bars are statistically significant when a statistical test was performed. Figures caption indicate the type of test used, and the number of animals used. We have added a supplementary table with the numbers of animals that performed each behavior.

We have used correction for multiple comparisons when we performed statistical tests on multiple samples simultaneously and we were not testing an individual hypothesis. We have used the Bonferroni correction for multiple comparisons to determine which type of error were different between novel backgrounds versus known backgrounds (Figure 5B for the asynchronous case and Figure 7C for the synchronous case). We found that novel background caused a significant increase in the number of misses and early lick events.

We have also used Bonferroni correction for multiple comparisons to determine for which of the 8 presentations of the novel background odors produced an increase in sniffs rate (Figure 6C for the asynchronous case and Figure 7F for the synchronous case). We found that the number of sniffs rate in response to a novel background odor was significantly elevated on the first presentation and was not significantly elevated on subsequent presentations.

Fig 5B

Fig 6C

Fig 7C

Fig 7F

Figure 5B (Asynchronous case). Types of error for known and novel background odors. P-values were calculated using the Fisher exact test adjusted using the Bonferroni correction. **Figure 6C (Asynchronous case).** Mean±s.e.m of the increase in sniff rate for novel odor compared to preceding trial with (s)-(-)-limonene as a function of the novel background odor presentation number (8 presentation total over two days). P-values were calculated using the Bonferroni corrected t-test. **Figure 7C (Synchronous case).** Types of errors made for known and novel background odors. **Figure 7F (Synchronous case).** F. Increase in sniff rate for novel background odor compared to preceding trial with (s)-(-)-limonene as a function of the novel background odor presentation number.

3. The authors did not discuss how the abundant literature on signal processing in the olfactory bulb during the go-no go odor discrimination task relates to their findings (see for example Doucette and Restrepo, PLOS Biology 6:e258, 2008, Chu et al. Neuron 92:174, 2016, Wang et al., J.Neurosci, 39:10002, 2019, Losacco et al., eLife 9:e52583, 2020). These publications show that responses to odorants recorded with different techniques (electrode recordings, multiphoton or fiber fluorescence imaging, or recording of local field potential) become divergent between the rewarded and unrewarded stimuli as the animal learns to differentiate the odorants. How does this relate to the validity of the different signal processing algorithms discussed in this manuscript? Would an increase in divergence in responses to the rewarded and unrewarded odorants favor a winner takes all algorithm?

The reviewer raises an interesting point regarding experimental results on olfactory bulb's neural plasticity and how these results might relate to the algorithms that mice might employ for odor identification in novel environments. Responses postsynaptic to the glomerular responses could include influences from multiple olfactory receptors. These divergent responses postsynaptic from the glomeruli might contribute to the implementations of several types of classifiers including linear classifiers or the NNC.

Our imaging analysis is based on individual glomerular responses. Amplification of responses originating from individual glomeruli could be necessary for strategies that use few glomeruli. It is a plausible hypothesis that during training mice might identify the best glomeruli that could detect the target odors and apply this single glomerulus criteria when challenged with novel background odors. With this in mind we have reanalyzed our imaging data to determine whether mice would have been able to solve the behavior based on individual glomeruli (see **Single glomeruli cannot reliably identify targets in novel backgrounds** on **page 6**). Individual glomeruli could solve the task for known background odors at $87.9 \pm 1.4\%$ which is similar to the mice performance (both WT and *Cntnap2*^{-/-}) for known background odors. As activation of individual glomeruli is not sufficient to produce suprathreshold responses in downstream neurons (Davison and Ehlers 2011), amplification of glomerular responses might be required to implement single glomeruli detectors. We have added a note referring to previous work on plasticity in the bulb in the discussion (page 22, lines 953-966).

However, the best individual glomeruli (see **Figure 2**) that could separate the go mixtures from the no-go mixtures in the training set, resulted in poor performance when challenged with the novel background odors, with performance of 53.0%, which was much lower than the behavioral performance of WT mice (asynchronous 76.9%, synchronous 77.2%) but close to *Cntnap2*^{-/-} mice performance (61.1%).

Figure 2 Odor responses from individual glomeruli could not identify target odors in novel environments. **A.** Example of the average glomerular responses of a WT mouse to the training set (16 mixtures) and responses to the test set with γ -terpinene as the novel background. The vertical line marks the ROI that best discriminated between the go mixtures from the no-go mixtures from the training set calculated using the auROC. **B.** AuROC for the training set and the test set of the example. **C.** AuROC for the training set plotted against the auROC for the test set calculated from 32 recording sessions from 6 animals using real mixtures (5163 glomeruli). **D.** ROC for the best glomerulus calculated from the training set for the above example. The circle indicates the optimal threshold to differentiate between the go mixtures and the no-go mixtures from the training set. **E.** Example of the responses of the best ROI, determined by the training set, to the training set and the test set. The blue line represents the optimal threshold calculated from

the training set. F. Performance of the best glomerulus for the training set and the test set for 6 WT mice, 32 recording sessions. Error bars represent standard error of the mean.

4. Concerning point 3 do the glomerular odorant responses change as the animal learns to discriminate the target odorants? Were the responses recorded in animals responding to odorants in the go-no go task? Changes in odorant processing during learning may alter glomerular responses through pre-synaptic modulation.

The revised version of the manuscript indicates that our intrinsic imaging measurements have been done in awake naive animals that were passively exposed to the odor and odor mixtures. Therefore, our decoder analysis does not include neither plasticity in olfactory receptor gene expression produced by olfactory experience (Ibarra-Soria et al. 2017) nor learned presynaptic modulations of neurotransmitter release (Kass et al. 2016; 2013; Abraham et al. 2014). Our imaging data represents the initial state of olfactory input that is used as a basis for the development of strategies based on modulation and plasticity that animals employ for difficult odor discrimination (Abraham et al. 2014).

Whether the plasticity that might have occurred at different levels, glomerular, mitral/tufted, olfactory cortex, during training might help for the odor recognition in novel environments is an interesting topic. However, it is difficult to measure the effect of the novel environment and relate it to our behavioral data. In order to get reliable images, we require multiple presentations of the mixtures that include the novel background odor, which might alter the responses, compared to the very brief exposure that mice get during the behavior (8 exposures total, with 14 seconds total per novel background odor).

5. When mice were trained with a restricted set there was a reduced performance for both the linear and nearest neighbor classifiers, but not for mouse behavior (Fig. 6D). The authors propose Lasso deconvolution as a potential algorithm used by WT mice. Unfortunately, this is not a satisfying solution because the Lasso algorithm cannot be tested with the data available in this study.

The reviewer is right in that it is not possible to determine the exact deconvolution algorithm used by the mice because it would require knowing the whole set of odors stored by the animal, which is not accessible with current technology. In the revised version of the manuscript we show that a broad set of implementations of the Lasso, with different dictionaries and different dictionary sizes, resulted in improved performance compared to linear classifiers and the Nearest Neighbor classifier (see section **A sparse deconvolution algorithm has better performance than NNC and linear classifier for reduced training data** on page 14 and **Figure 9 D-F**) making deconvolution a plausible solution.

Figure 9. D-F. Sparse representation algorithm, Lasso, identifies the odors present in a mixture. Mixture of a go target odor (isopropyl butyrate), contextual background odor (isoamyl acetate), and a novel background odor that was not part of the dictionary used by the Lasso. **E.** Concentrations assigned by the Lasso for a dictionary size of 500 elements. Bars correspond to the estimated concentrations of the four contextual background odors, the four target odors, and (s)-(-)-limonene. Stems correspond to estimated concentrations for the randomly generated dictionary elements. The maximum of the estimated concentration was larger for the two target go odors compared to the two target no-go odors, so this would be considered a correct identification. **F.** Performance of the Lasso calculated for different size of dictionaries was compared to the performances of the NNC, SVM, and logistic regression. Individual symbols represent average performances for the 5 WT mice. Error bars represent the 10-90% percentiles of the 13200 simulations performed per dictionary size and the SVM, logistic regression and NNC. Significance was calculated using a t-test and comparing the average performance per animal between the NNC, SVM and logistic regression against the Lasso performance for different dictionary sizes.

Furthermore, a major limitation of this study is that the responses are only measured in the dorsal olfactory bulb. Would the performance of the linear or nearest neighbor algorithms for the restricted odorant set increase if the ventral glomeruli were added to the data set? As a control, what happens if decoding input restricted to half of the dorsal glomeruli?

We thank the reviewer for this idea. To answer it, we have systematically explored how the responses of the tested algorithms change as we reduced the number of glomeruli used in the calculation. We used a sparseness constrain to limit the number of glomeruli included for the linear classifiers. We also restricted the number of glomeruli used for the nearest neighbor classifier by selecting glomeruli that exceeded certain selectivity (calculated based on auROC for the training set).

Although single glomeruli were not sufficient to generalize to novel background odors, plateau performance was reached using between 24 and 36 glomeruli (see **Plateau performance is achieved with a small fraction of the glomeruli** on page 7 and Figure 4).

Figure 4. Linear classifiers and the NNC did not require the use of all available glomeruli to reach plateau performance. **A.** Example of regressor weights for the linear SVM as the sparseness constraint is changed. **B.** The SVM regressor weights were applied to test set mixtures where methyl pyruvate was the novel background odor. **C-D.** Logistic regressor weights were also calculated from the training set and were applied to the test set. **E.** Number of glomeruli used in the NNC were changed by thresholding based on the auROC of the training set of individual glomeruli. Similarity matrices changed as the threshold was changed. The red squares indicate the best match to the training set. **F.** Performances of the linear classifiers and the NNC as a function of the number of ROI included calculated for 32 recording sessions, 6 WT mice. Vertical error bars are the s.e.m of the performance of the classifiers and horizontal error bar are the s.e.m for the number of glomeruli used for the classifiers.

However, when we tried to predict to performance of the WT mice, the number of glomeruli increased to at least 100 glomeruli (see **Classifiers that included more glomeruli had higher correlation with animal behavior** on **page 12** and see **Figure 8**). So animals seem to be using more glomeruli than are actually necessary to reach a given performance. We have added to the discussion the following section (page 23, line 993-996):

”Linear classifiers and NNC performance plateaued when using 24-36 selected glomeruli. However, these algorithms required at least 100 glomeruli to predict the performance of WT mice on individual novel background odors. By using larger number of glomeruli, WT mice might mitigate the effect of noisy glomerular responses.”

○ Synchronous (4 WT mice)
 ○ Asynchronous (4 WT mice)

Figure 8. Classifiers that included more glomeruli had higher correlation with animal behavior. **A.** Performance of the NNC for 10 novel background odors was calculated using all the available glomeruli per imaging session (161 ± 40 ROI, mean \pm s.d, 32 imaging sessions, 6 WT mice). Black circles y coordinates represent the average behavioral performance on a novel background odor from four WT mice that performed the asynchronous task and red circle's y coordinate represent the average performance of the four WT mice that performed the synchronous task. The blue line is the linear correlation between imaging data and behavior. **B-C** Similar plot for logistic regression and SVM. **D-F.** Reducing the number of glomeruli reduced correlation with behavior for all three classifiers. **G.** Classifier's correlation with behavior as a function of the number of ROI included. Error bars represent the standard deviation of the distribution of linear correlation coefficients (vertical) and number of glomeruli (horizontal) included calculated using a Montecarlo simulation with 500 repeats. The asterisks indicate the results of t-test comparing the distribution of correlations using all available glomeruli against the distribution of correlations using the reduced number of glomeruli.

6. The use of glomerular activation data generated by virtual odorant mixtures for the decoding analysis is not granted given the well-known nonlinearities of the olfactory system. It does not appear that there is a correlation between virtual and real odorant mixture decoding. Unless the use of virtual odorant mixtures is validated by data showing that there is a correlation for decoding performance with real odor mixtures I would advise to remove all the virtual odorant data from Fig. 5, and leave only the real odorant mixture results throughout the manuscript.

We agree with the reviewer that the glomerular responses to actual mixtures reflect better the data that is available for the animals to make their decisions. To do this, we have expanded the number of recordings of real odor mixtures with 32 recording sessions from 6 WT mice (161.3 ± 39.8 glomeruli per session, mean \pm s.d, 5163 glomeruli). We have used the real mixtures data when we compare animal behavior to classifier behavior.

7. The paragraph starting in line 307 makes the argument that the similarity of odor pair responses correlates between WT and *Cntnap2*^{-/-}. However, the data in Figs. 7A and B do not corroborate this conclusion. First, the examples in Fig. 7A do not illustrate the similarity in response of an odor pair. Responses are only shown for one odorant and they appear fairly different raising the question whether there are significant differences in glomerular input.

This would not be surprising given that *Cntnap2* is expressed in the olfactory epithelium. Second, there is no panel showing how the similarity of response patterns is computed. Third, the correlation in Fig. 7B is significant, but there are clear differences between WT and *Cntnap2*^{-/-}. Is the correlation shown in Fig. 7B similar for regular and novel background odors? The manuscript should show a thorough comparison of odorant response patterns and odorant pair similarities between genotypes.

The reviewer raises an important point regarding the comparison between glomerular responses between *Cntnap2* knockout mice and WT mice. Our objective was to determine how much of the *Cntnap2*^{-/-} mice behavior could be accounted by differences in the olfactory epithelium that would be reflected in the intrinsic optical signal. We have expanded significantly the dataset by recording from 10 animals (5 WT and 5 *Cntnap2*^{-/-} mice). This expanded data set has permitted us to do a more thorough analysis for comparing glomerular responses across genotypes. Although

Cntnap2^{-/-} mice have no reported olfactory deficits (Peñagarikano et al. 2011), CNTNAP2 is expressed in the olfactory epithelium and *Cntnap2*^{-/-} mice have been shown to have reduced neural responses to odors in the olfactory bulb as well as more trial-to-trial variability compared with WT mice (M. Geramita and Urban 2017). In the revised version of the manuscript we have directly compared average glomerular responses, similarity of odor response profiles, and glomerular response variability between *Cntnap2*^{-/-} mice and WT mice. The correlation shown in the previous version of the manuscript on Fig. 7B is similar for regular and novel background odors.

a) We have analyzed the trial to trial variability of the WT mice and the *Cntnap2*^{-/-} mice (see **page 3, line 103:131, Methods: Measurement of trial to trial variability in WT mice** on page 32, and **Methods: Measurement of trial to trial variability in *Cntnap2*^{-/-} mice** on page 34 to see how the variability was estimated). Consistent with a previous report (M. Geramita and Urban 2017), there was increased trial-to-trial variability in the *Cntnap2*^{-/-} mice compared to WT mice (see **Figure 1G-H** and **Figure 10B-C**). The variability was also higher in the *Cntnap2*^{-/-} mice after subtracting the common variability across glomeruli (see **Methods: Measurement of uncorrelated component of trial to trial variability in WT mice** on page 33, **Methods: Measurement of uncorrelated component of trial to trial variability in *Cntnap2*^{-/-} mice** on page 34, and see **Supplementary Figure 11**).

Figure 1. G. Single trial responses for individual odors. **H.** Average z-score indicating the periods that were used to quantify the odor response. The air baseline period is also indicated as well as the z-score threshold (-0.46) used to detect glomerular responses.

Figure 10 B. Examples responses as z-scores from individual glomeruli of a *Cntnap2*^{-/-} mouse on individual odor presentations and as averages. **Figure 10 C.** Average response per odor plotted against the standard deviation calculated over trials for 5 *Cntnap2*^{-/-} mice (red dots) and 5 WT mice (black dots). The magenta lines are the fitted functions for both genotypes used to estimate the coefficient of variation and imaging noise levels.

Supplementary Figure 11

Intrinsic glomerular response in awake *Cntnap2^{-/-}* mice had correlated fluctuations. **A.** Example of an image of average z-score responses to 22 presentations of ethyl butyrate (0.1%) in an awake *Cntnap2^{-/-}* mouse. The odor was presented as a 9 second pulse. Odor responses are the average z-score calculated using the period between 2 and 9 seconds following odor onset and using the 5 seconds preceding the onset of the odor as the baseline. **B.** Example of trial by trial correlated odor evoked responses of two glomeruli located on different spots. **C.** Glomerular responses for 132 glomeruli for 2 different odor mixture presentations plotted against the average response over 22 odor presentations. Solid lines are the least square linear fit. **D.** We calculated the deviations from the best fitted line for 2980 ROI-odor pairs recorded from 5 *Cntnap2^{-/-}* mice and plotted in red the standard deviations of their deviation from the best fitted line (σ_{uncorr}) against the average response for that ROI-odor pair (μ). The black dots correspond to data from the WT mice. *Cntnap2^{-/-}* mice had higher uncorrelated trial-to-trial variability compared to WT mice.

- b) We have analyzed the fraction of glomeruli activated by each odor as well as the average activation produced by each odor. These two quantities were correlated between the *Cntnap2^{-/-}* mice (5 mice) and the WT mice (5 mice). We did not find a generalized suppression of average glomerular activity in *Cntnap2^{-/-}* mice for the odors that we used at the concentrations we used for evaluating the behavior.

Page 15, line 657-673: “We wondered whether the *Cntnap2^{-/-}* odor evoked activity was weaker than WT mice. We compared the average activity of the glomerular activation per odor between WT (5 animals, 775 glomeruli) and *Cntnap2^{-/-}* mice (5 animals, 643 glomeruli). *Cntnap2^{-/-}* mice average glomerular responses were not systematically weaker than the WT responses ($p=0.50$, binomial test, 20 odors, see **Fig. 10D, F**) nor the odors activated a smaller fraction of the glomeruli ($p=0.11$, binomial test, 20 odors, see **Fig. 10E, G**). In fact, odors that produced strong glomerular activation patterns in the WT mice also produced strong glomerular activation patterns in the *Cntnap2^{-/-}* mice. There was a significant linear correlation ($r=0.66$, $p=0.0013$, Pearson linear coefficient) between the average glomerular response for a given odor between WT mice and *Cntnap2^{-/-}* mice. The fraction of glomeruli that responded to a given odor and exceeded the detection threshold (z-score=-0.46 for *Cntnap2^{-/-}* mice and z-score=-0.42 for WT mice) was also linearly correlated between WT mice and *Cntnap2^{-/-}* mice ($r=0.80$, $p=1e-5$). The average

glomerular activity produced by an individual odor in the WT mice was not significantly different to the activity in the *Cntnap2*^{-/-} mice ($p > 0.13$, for all 20 odors, t-test). The fraction of glomeruli that responded to a given odor was also not significantly different between the *Cntnap2*^{-/-} mice and the WT mice ($p > 0.11$, for all 20 odors, t-test). *Cntnap2*^{-/-} mice glomerular responses were not systematically weaker compared to WT mice.”

Figure 10. **D.** Average activation of the glomeruli produced by individual odors for the 5 *Cntnap2*^{-/-} mice plotted against average activation for 5 WT mice. **E.** Fraction of glomeruli activated per odor for the 5 *Cntnap2*^{-/-} mice plotted against fraction of glomeruli activated in 5 WT mice. Blue line represents the identity line. **F.** Average activation of glomeruli per odor for 5 *Cntnap2*^{-/-} mice and 5 WT mice. Each symbol is the average activation of one mouse. **G.** Fraction of glomeruli activated per odor.

c) We have compared odor similarities between odor pairs using the normalized dot product (page 16, lines 674-687). We created a large vector for all the glomeruli recorded for each phenotype (775 glomeruli from 5 WT mice, 643 glomeruli from 5 *Cntnap2*^{-/-} mice) and calculated the similarity as the normalized dot product between glomerular representations of pairs of odors. We have plotted the similarity matrix mice using all glomeruli available for both the WT and the *Cntnap2* mice. We have added a pair of examples showing that odors that were similar for the WT mice were also similar for the *Cntnap2*^{-/-} mice, whereas pairs of odors that were different for WT mice were also different for *Cntnap2*^{-/-} mice (see Figure 11 A-C).

The similarities among the standard background odors considered separately (four contextual backgrounds and (s)-(-)-limonene) were also significantly correlated across genotypes ($r = 0.71$, $p = 0.02$, $n = 10$ similarity comparisons) as well as the similarities among the 11 novel background odors ($r = 0.36$, $p = 0.007$, $n = 55$ similarity comparisons).

Figure 11. Pairs of odors that evoked similar activity patterns in WT mice also evoked similar patterns in *Cntnap2*^{-/-} mice. **A.** Patterns of glomerular activation using all available ROI per genotype for 4 example odors. Isoamyl acetate and ethyl benzoylacetate produced different patterns of glomerular activation in WT and *Cntnap2*^{-/-} mice whereas 2-3 pentanedione and acetal produced similar glomerular activation patterns in both WT and *Cntnap2*^{-/-} mice. **B.** Odor similarity matrices for *Cntnap2*^{-/-} mice and WT mice calculated using all available ROI per genotype. **C.** Similarity between 190 odor pairs calculated using all the WT mice glomerular responses versus similarity between odors calculated using all the *Cntnap2*^{-/-} mice glomerular responses.

d) To determine whether odor pair similarity across genotypes was greater than odor pair similarity within genotypes, we also compared odor similarities between individual animals from each genotype (page 16, lines 688-699). We compared the odor similarities pattern in 5 WT mice and 5 *Cntnap2*^{-/-} mice, resulting in 25 comparisons across the two genotypes (see Fig. 11D-F for an example of odor similarity correlation between a WT and a *Cntnap2*^{-/-} mouse). All the 25 comparison's across genotypes produced positive significant linear correlations ($p < 0.014$, see Fig. 11G). The average odor similarity correlation across the two genotypes was 0.49 ± 0.03 (mean \pm s.e.m, $n=25$ pairs of WT- *Cntnap2*^{-/-} mice) which was not significantly different ($p=0.14$, t-test) to the average odor similarity correlation within the same genotype (0.51 ± 0.04 , mean \pm s.e.m, 10 pairs of WT-WT comparisons and 10 pairs of *Cntnap2*^{-/-}-*Cntnap2*^{-/-} comparisons).

Figure 11. Pairs of odors that evoked similar activity patterns in WT mice also evoked similar patterns in *Cntnap2*^{-/-} mice. **D.** Patterns of glomerular activation using ROI from example individual animals of each genotype. **E.** Odor similarity matrices for the example *Cntnap2*^{-/-} mouse and WT mouse. **F.** Similarity between 190 odor pairs calculated using the example WT mouse data against the similarity from the example *Cntnap2*^{-/-} mouse. **G.** Distribution of linear correlation coefficients of odor similarities (r) for pairs of animals of the same genotype (WT vs WT, 10 animal pairs and *Cntnap2*^{-/-} vs *Cntnap2*^{-/-} 10 animal pairs) and for pairs of animals of different genotypes (WT vs *Cntnap2*^{-/-}, 25 animal pairs). Error bars represent the mean±s.e.m. Comparison between correlation coefficients were done using a t-test.

- e) Although average glomerular responses in the *Cntnap2*^{-/-} and WT mice were similar, algorithms performing odor detection in novel environments could still be affected by the increased trial-to-trial variability in *Cntnap2*^{-/-} mice ($CV_{uncorr} = 0.44$) compared to WT mice ($CV_{uncorr} = 0.25$). To test this (see **Increased trial-to-trial variability in glomerular activity in *Cntnap2*^{-/-} did not significantly reduce performance of algorithms on page 16**), we created artificial mixtures using the imaging data from *Cntnap2*^{-/-} mice (5 mice, 643 ROI) and simulated the increased variability using a coefficient of variation of $CV_{uncorr} = 0.44$ and *Cntnap2*^{-/-} mice threshold of glomerular activation of -0.46. We trained the SVM, logistic regression, and the NNC using the full set of training examples (16 mixtures) and tested them on the 160 mixtures that included 10 novel background odors. We compared the performance of the classifiers trained with *Cntnap2*^{-/-} mice imaging data with the performance of the classifiers trained with WT mice imaging data (5 mice, 775 ROI). We performed the analysis on each individual animal and pooled the average performance for each odor resulting in 50 performances (5 animals * 10 novel background odors= 50 animal-odor pairs). The performance of the algorithms using *Cntnap2*^{-/-} mice data was not significantly different from the algorithms that used the WT mice data (see **Fig. 11H**). Using *Cntnap2*^{-/-} mice glomerular imaging data, the performance of the SVM was 83.5±1.1% (mean ± s.e.m.) which was not significantly different from WT mice glomerular imaging data (81.1±1.2%, mean ± s.e.m, p=0.11, t-test). Using *Cntnap2*^{-/-} glomerular imaging data, the performance of the logistic regression was 84.2± 1.3% which was not significantly different from the performance calculated using WT mice glomerular imaging data (83.0±1.2%, mean ± s.e.m, p= 0.81, t-test). The performance of the NNC using *Cntnap2*^{-/-} glomerular imaging data was 86.2±1.1% (mean±s.e.m,) which was not significantly different from the WT glomerular imaging data (85.3±2.2%, mean ± s.e.m, p=0.69, t-test). We also tested the performance of the algorithms using our dataset of real odor mixtures collected from 6 WT mice (32 recording sessions, 5163 ROI), but simulated with the larger trial-to-trial odor evoked variability ($CV_{uncorr} = 0.44$) measured in the *Cntnap2*^{-/-} glomerular imaging data (**Supplementary material 5: Increased trial-to-trial variability did not affect NNC and linear classifiers** and **Supp. Fig. 13**). We also used the threshold of glomerular activation of -0.46 that we identified for the *Cntnap2*^{-/-} mice. There was also no significant difference (p>0.4, t-test, n=32 recording sessions) in the performance of the high-variability regime (*Cntnap2*^{-/-} like) compared to the reduced variability regime (WT like).

Figure 11. H. Performance (mean±s.e.m) of SVM, logistic, and NNC classifiers calculated using *Cntnap2*^{-/-} and WT mice glomerular activation data for target detection in novel environments trained with the full training set and tested with the full training set. Symbols represent average performance per animal. **I.** Performance (mean±s.e.m) of the Lasso using *Cntnap2*^{-/-} and WT mice glomerular data for the reduced test set using different sizes of dictionaries. **Supplementary Figure 13. Performances of the linear SVM, logistic, and NNC on novel background odors were not affected by the higher trial to trial variability seen in *Cntnap2*^{-/-} mice.** **A.** Performance of linear SVM classifier in the low variability (WT like) and high variability regime (*Cntnap2*^{-/-} like). The performance for each recording session was calculated by creating 100 instantiations of the test set (16 mixtures that included a novel background odor) as shown in Figure 3. To simulate the *Cntnap2*^{-/-} mice data, we used the higher trial to trial variability ($CV_{uncorr} = 0.44$) and the lower z-score threshold (z-score<-0.46) whereas to simulate the WT mice we used the lower variability ($CV_{uncorr} = 0.25$) and the higher z-score threshold (z_score<-0.42). The p-values were calculated using a t-test comparing the high variability and the low variability results (n=32 recording sessions).

f) We also used the *Cntnap2*^{-/-} mice imaging data to test the Lasso with the reduced training set (see **page 17, lines 730-741**). Performance of the Lasso was not affected by the higher glomerular activity variability of the *Cntnap2*^{-/-} mice. We simulated the Lasso as described above using the glomerular data from the 5 *Cntnap2*^{-/-} mice with their increased trial to trial variability ($CV_{uncorr} = 0.44$). The performance of the Lasso using *Cntnap2*^{-/-} mice imaging data in the reduced test set was 92.6±0.5% (mean±s.d., 10 tested sizes of dictionary between 100 and 1000 elements) and it was very similar to the performance calculated for the WT mice (93.2±0.9%, see **Fig. 11I**). For each dictionary size, we compared the performance of the Lasso using the 5 WT mice imaging data with the 5 *Cntnap2*^{-/-} mice. The performances of the animals were not significantly different for any of the dictionary sizes tested (p>0.5, t-test).

Figure 11 I. Performance (mean±s.e.m) of the Lasso using *Cntnap2*^{-/-} and WT mice glomerular data for the reduced test set using different sizes of dictionaries. Significance was calculated using a t-test.

Average glomerular responses were very similar between *Cntnap2*^{-/-} and WT mice. Although there was higher trial to trial variability in the odor responses in *Cntnap2*^{-/-} mice, our simulations show that it is unlikely that the increase glomerular variability would affect odor recognition in novel environments.

8. Is the fraction of trials for Miss in Fig. 7F significantly different between genotypes? Since the comparison between genotypes is the main point of Fig. 7F please plot the bar graph with the bars for the two genotypes next to each other and asterisks denoting statistical differences.

For both WT and *Cntnap2*^{-/-} mice that performed the reduced training set task, misses that is, animal not licking for the go stimulus, were the most common type of error when tested with novel background odors (see section ***Cntnap2*^{-/-} mice errors in novel environments were not caused by excessive licking** on page 19 and Figure 12 G-H).

Figure 12. *Cntnap2*^{-/-} mice target discrimination in novel backgrounds was selectively affected. **G.** Fraction of types of errors for WT and *Cntnap2*^{-/-} mice for standard background odors. Error bars are the 95% confidence intervals. P-values were calculated using the Fisher exact test. **H.** Types of errors for WT and *Cntnap2*^{-/-} mice for novel background odors.

Comparing the time of response for the asynchronous case between WT (line 267) and *Cntnap2*^{-/-} (line 337) it appears that the response for *Cntnap2*^{-/-} was slower. Is this correct?

The average response times for the *Cntnap2*^{-/-} mice were 13 ms faster than the WT mice trained with the full set as noted by the reviewer. However, the difference was not statistically significant. In the revised version of the manuscript, we have directly compared the reaction time between the *Cntnap2*^{-/-} mice and the 3 WT mice that performed the same asynchronous task with the reduced training set (page 20, lines 888-895). This comparison is more appropriate as both groups of animals were doing the same task. Although the reaction time to novel background odors was also faster for the *Cntnap2*^{-/-} mice compared to the WT mice, this difference was not significant (*Cntnap2*^{-/-} mice 553.0± 67.1 ms; WT mice , 621.5± 60.2 ms; p= 0.25).

Was there a difference between genotypes for the number of sniffs required for decision making?

We thank the reviewer for this question which has helped us provide a better characterization of the *Cntnap2*^{-/-} mice sniff responses to odors during olfactory behaviors. In the revised version of the manuscript we have substantially expanded the analysis of the sniff responses and reaction time in the *Cntnap2*^{-/-} mice compared to the WT mice. Interestingly, the *Cntnap2*^{-/-} mice did not increase their sniff rates when the target odors appeared in the standard condition, whereas WT mice in the asynchronous tasks elevated their sniff rates when the target appears both in the full training set (see Fig 6B) and the reduced training set (see **Cntnap2^{-/-} mice did not increase their sniff rate for target odors in known backgrounds** on page 19 and see Fig 12I).

Figure 6 B. Mean \pm s.e.m respiration signal for four WT mice for the first presentation of novel background odors compared to the presentation of the interleaved trials with standard background odors. **Figure 12 I.** Sniff responses for *Cntnap2*^{-/-} and WT mice for novel and standard background odors. Lines represent mean \pm s.e.m.

This lack of sniff response to the target is similar to the lack of sniff modulation observed in autistic subjects compared to neurotypical subjects (Rozenkrantz et al. 2015). However, when the novel odors appeared, the *Cntnap2*^{-/-} mice did modulate their sniff responses in an almost identical manner to the WT mice.

For the known background odors, because of the lack of increase in sniff rate in *Cntnap2*^{-/-} mice, the number of sniffs taken before a licking response was lower for *Cntnap2*^{-/-} mice compared to the WT mice. *Cntnap2*^{-/-} mice took 2.3 \pm 0.1 sniffs (mean \pm s.e.m, n= 462 lick responses, 4 mice) whereas WT mice took 2.5 \pm 0.1 sniffs (mean \pm s.e.m, n=496 lick responses, 3 mice, p= 0.008, Wilcoxon rank-sum test).

The number of sniffs required for decision making for novel background odors was not significantly different between genotypes. *Cntnap2*^{-/-} mice took 2.4 \pm 0.2 sniffs before licking in response to the target on the novel background odor (n = 49 lick responses, 4 *Cntnap2*^{-/-} mice) and WT mice took a similar number of sniffs (2.6 \pm 0.2, n=67 lick responses, 3 WT mice, p=0.65, Wilcoxon rank-sum test).

Minor comments

1. Please revise the English throughout.

We apologize to the reviewer. We have revised the manuscript.

2. For the paragraph starting in line 97 please reference a diagram that illustrates the experimental design.

We have added a diagram explaining the odor delivery and the licking contingencies with more detail in **Supplementary Figure 7**.

Supplementary Figure 7. Temporal profile of olfactory stimulus delivery and behavioral contingencies for the asynchronous task. Odor valves opened for 7 seconds to establish stable odor concentration at the input of the valves close to the animal's snout. After 7 seconds, a valve near the mouse opened and routed the contextual background odor flow to the mouse's snout. After 0.75 s, the novel background odor was delivered, followed after 0.75 s by the target odor. The mouse had to lick with a latency of >0.3 s from target onset to get rewarded. The target appeared concurrently with the background odors for 3 s. If the target was a go-odor and the animal licked during the response window (hit), the animal received water and 5 seconds were added to the intertrial interval. If the target odor was a no-go odor and the animal responded, it was a false alarm and 10 s were added to the intertrial interval. If the animal responded before the 0.3 s minimal latency from target onset (early lick), extra 10 s were added to the intertrial interval. If the animal did not lick in response to the odor, either as a miss or a correct rejection, 5 s were added to the intertrial interval. There was an extra random interval between 12 and 16 s that was always present in the intertrial interval.

3. Line 108. Please reference Slotnick since his publications were those that placed the go-no go task on the map for studying olfaction in rodents.

We now cite (Bodyak and Slotnick 1999) when we describe the task (page 2, line 79) and (Slotnick and Bisulco 2003) when we describe the redundancy of the olfactory bulb (page 22, line 950-952). We thank the reviewer for this suggestion.

4. The manuscript refers to "false rejections". This term is not often used in the literature. Please use the standard terms: Hit, Miss, Correct Rejection and False Alarm.

We have made the change and we use these terms across the manuscript.

5. Line 175: change to "quantified the animal's familiarity"

We have made the change.

6. Line 179: change “Consistently” to “consistent”

We have made the change.

7. Lines 199-200. Please provide the statistical basis for the statement that “the mice had an increase in false rejections”

We have performed a test and compared the number of misses during novel backgrounds with the number of misses with known backgrounds during the synchronous odor behavior.

We also found that there was a small but significant increase in the baseline of sniffing. We hypothesize that the synchronous condition requires higher alertness in the mice in order not to miss the target

Page 11, lines 446-459: “Compared to the asynchronous case, there was a small but significant increase in the number of misses in the synchronous case compared to the fraction of misses in the asynchronous case (from only 0.7% (12/1563) to 3.9% (46/1191), $p=2.1e-8$, Fisher exact test, see **Fig. 7C**). WT mice increased their baseline sniffing during synchronous odor presentation compared to the asynchronous case suggesting an enhanced level of alertness in order not to miss the target.”

“WT mice increased their baseline sniffing during synchronous odor presentation compared to the asynchronous case suggesting an enhanced level of alertness in order not to miss the target. The base rate of sniffing before odor onset was higher for WT mice that were doing the synchronous task (2.62 ± 0.04 sniffs per second, $n=1191$ trials, $n=4$ animals) compared to WT mice doing the asynchronous task (2.18 ± 0.03 , $n=1563$ trials, $p=6.9e-19$, t-test, $n=4$ WT mice). These differences were not due to differences in animal batches. The same WT mice that did the synchronous task had also significant lower baseline sniff rates (2.07 ± 0.04 sniffs per second, $n=1209$, $n=4$ WT mice) when they were performing the asynchronous task with known background odors during their training process, compared to when they performed the synchronous task ($p=5.99e-23$, t-test). The synchronous task seems to require increased baseline attention level given the unpredictable appearance of the target.”

8. Fig. 4E. Is this mean+SEM?

Yes, we have specified for this and all other graphs the meaning of the error bars.

9. Line 215. Please show the correlation of brief inhalation and higher performance.

We have included the supplementary figure 10 and a supplementary material.

Page 26: “Supplementary material 4: Briefer inhalation widths were correlated with increased WT mice performance in novel environments

WT mice increased their sniff rate and reduced the inhalation width on the second sniff after the onset of the novel background odor, as they acquired the information that a novel odor was present from the first inhalation and reacted to it on the second inhalation. We compared the second inhalation width after the onset of the novel background odor between correct and incorrect trials to determine if fast sniffing was correlated with better performance. This second

sniff width was significantly shorter ($p=0.014$, Wilcoxon rank sum test, single tailed) in the correct trials (91.3 ± 2.0 ms, 552 trials, 8 WT mice) compared to the incorrect trials (102.1 ± 4.5 ms, 131 trials). Interestingly, fast sniffing was not correlated with better performance for known backgrounds; the second inhalation width after (s)-(-)-limonene onset was not significantly shorter for known backgrounds ($p=0.97$, Wilcoxon rank sum test, single tailed), with a width of 101.6 ± 1.0 ms ($n=2816$ trials) for the correct trials and 94.6 ± 2.7 ms, ($n=331$ trials) for the incorrect trials.

We wondered if the correlation between the second sniff width and performance might be caused by slow fluctuations in animal alertness over the course of a behavioral session. If so, the trend between sniff rate and performance should already be noticeable from the first sniff after novel odor background onset. However, there was no difference in inhalation width ($p=0.50$, single tailed Wilcoxon sum test) between correct trials (110.2 ± 2.6 ms, 552 trials) and incorrect trials (109.6 ± 5.3 ms, 131 trials). There was also no difference ($p=0.85$, Wilcoxon rank sum test, single tailed) for the known backgrounds, with 105.3 ± 1.1 ms (2816 trials) for the correct trials and 100.7 ± 2.8 ms (331 trials) for the incorrect trials. Faster sniffing increased odor identification performance only for novel background odors in WT mice.”

Supplementary Figure 10

Brief inhalations in response to novel background odors were associated with improved performance **A**. Example trial of sniffing response to a novel background in which the WT mouse did not change their inhalation widths nor frequency (downward deflections) in response to the novel background odor. **B**. Example trial of sniffing response where the mouse produced a brief second inhalation after the onset of the novel background. **C**. The first inhalations in response to novel background and (s)-(-)-limonene were not significantly different between correct and incorrect trials. **D**. The second inhalation width reflects the animal reaction to novel background odors and it was significantly briefer in correct trials. There was not effect of briefer inhalation for known backgrounds. Error bars indicate s.e.m and the p-values of the reduction of the sniff widths were calculated using the single tailed Wilcoxon rank sum test.

10. Line 221 change “mice’s” to “mouse’s” here and in several other instances.

We apologize. We have corrected this and other typos.

11. Line 224. What is the basis for the statement that the linear classifier is less likely to be affected in mouse models of autism? Please clarify or remove the statement.

We have clarified our statement. We meant that the linear classifier implementation is very simple as it requires a small population of neurons that receives projections from multiple glomeruli from the olfactory bulb. Several structures in the olfactory pathway meet these requirements (piriform cortex, anterior olfactory nucleus, olfactory tubercle, and cortical amygdala) permitting a degree of redundancy that could compensate for circuit defects in any of these areas (page 7, line 262-271).

12. Line 416. Change “dids” to “did”

We have made the change.

13. Line 428. Was the 0.8% increase statistically significant?

This change produced by previous exposure to the background odors was not significant ($p=0.89$, Fisher exact test).

14. Line 451. Please provide statistics.

The baseline sniff rate was higher in animals doing the synchronous task. We also compared the sniff rates of the animals that performed the synchronous task when the same animals performed the asynchronous task as part of their training. Their base sniff rate was also elevated when they started doing the synchronous task. We have added the statistics.

Page 11, line 450-458: “The base rate of sniffing before odor onset was higher for WT mice that were doing the synchronous task (2.62 ± 0.04 sniffs per second, $n=1191$ trials, $n=4$ animals) compared to WT mice doing the asynchronous task (2.18 ± 0.03 , $n=1563$ trials, $p=6.9e-19$, t-test, $n=4$ WT mice). These differences were not due to differences in animal batches. The same WT mice that did the synchronous task had also significant lower baseline sniff rates (2.07 ± 0.04 sniffs per second, $n=1209$, $n=4$ WT mice) when they were performing the asynchronous task with known background odors during their training process, compared to when they performed the synchronous task ($p=5.99e-23$, t-test). The synchronous task seems to require increased baseline attention level given the unpredictable appearance of the target.”

15. There is a problem with the legend of the data in Fig. 7. Are the data in Figs. 7D and E for the full or the restricted training set? Please state the results of the statistical tests and give the number of animals and trials included in the data. What are the error bars?

All the data for WT mice on Fig 12 A-B (former 7D-E) are for the three WT mice that performed the task with the restricted set, which is the same task done as the *Cntnap2*^{-/-} mice. The error bars for behavioral performance are 95 percentile error, which we used for all the behavioral reports.

Figure 12. *Cntnap2*^{-/-} mice target discrimination in novel backgrounds was selectively affected. **A.** Performance for four *Cntnap2*^{-/-} mice and three WT mice on known backgrounds for the reduced training set. Symbols indicate average performance for individual animals and error bars are the 95% confidence interval. Significance across genotypes were calculated using the Fisher exact test. **B.** Performance in novel background for the reduced test set.

16. For Fig. 4C please plot the novel background bars next to the known background bars and show whether the differences are statistically significant. Was there correction for multiple comparisons?

We have changed the figure and the compared the differences in errors between novel and known background odors. We used the Bonferroni correction for multiple comparisons in the p-values calculated using the Fisher exact test as we did not have a prior hypothesis of what type of error rates would increase with novel background odors. Both the fraction of misses and early responses were significantly increased in the presence of novel backgrounds (see Figure 5B).

Figure 5 B. Types of error for known and novel background odors. P-values were calculated using the Fisher exact test adjusted using the Bonferroni correction

17. Please provide a diagram for the air-dilution machine and provide PID measurements of odor delivery time course.

We thank the reviewer for this helpful suggestion regarding the odor delivery time course. We have added a diagram of our odor machine including measured airflows (see page 28, Methods:

Odor delivery and Supplementary Figure 1). Our odor machine has a long delay and a short delay. The long delay is the time required for the odorized air to go from the odor vial to the valve next to the animal snout. The short delay is the time for the odorized air to go from the valve next to the animal snout to actually reach the animal snout. We calculated these delays based on the airflows, length of tubing, and tubing cross-sections (see page 28, Methods: **Odor machine latencies**). The calculated delays for the odor to reach from the odor vials to the valve close to the animal (3.4 s for the target and 2.6 s for the backgrounds) were much lower than the 7 seconds that we used for the establishment of odor near the animal's snout. The total delay between the opening of the valve next to the animal nose to the odor reaching the animal, including air transport delay and valve opening mechanical delay (15ms) was calculated to be 108 ms based on the airflows, length of tubing, and tubing cross-sections.

We confirmed our measurement of latencies and the reliability of the machine using a metal-oxide sensor as recently described by (Tariq et al. 2021) (see page 29, Methods: **Metal oxide measurements of odor machine latencies and Supplementary Figure 2**). The coefficient of variation was 3%. The delay from valve opening to odor delivery to the mouse snout was 121 ms. In the revised version of the manuscript we have corrected all of our mouse reaction times and odor sniff responses given the 121 ms delay. We do not have access to a PID.

Supplementary Figure 1

Diagram of the triple serial air dilution odor machine. Values indicate the calculated delay on each line based on the tubing cross-section (1/8 inch diameter, Teflon coated) and airflow. The values of the flowmeters (and calculated delays) correspond to the concentrations used during testing with novel background odors. The total concentration of the targets at the animal snout was 0.025% of saturated vapor and the concentration of the contextual background odors and the novel odors, and (s)-(-)-limonene was 0.1% of saturated vapor.

Supplementary Figure 2

Metal oxide sensor readings of the odor machine response time. **A.** Measurement of the delay between odor vial opening and odor arrival to the animal snout. The figure shows the superposition of 60 traces produced by 11.5 second long pulses of ethanol at 0.1% dilution connected to the contextual background line. To perform the measurement both the odor vial and the animal's snout valve opened synchronously. During the behavior and the imaging sessions, the odor vial valves opened at least 7 seconds before the animal snout valve to create a stable odor stream close to the animal snout. The calculated delay for this line to reach from the odor vial to the animal snout is 2.6 s. The actual measured delay was 2.4 s. We calculated a coefficient of variation of 3% over 60 repeats, by comparing the standard deviation with respect to the mean value of the signal averaged over the 4.5 seconds during the odor delivery. **B.** Measurement of the delay between animal snout valve opening and odor arrival to the animal snout. To perform this measurement, the odor vial valve opened 7 seconds before the animal snout valve opened to establish a constant odor concentration at the input of the animal snout valve. The calculated delay to reach from the animal snout valve to the animal snout is 108 ms. The actual measured delay was 121 ms.

18. What is shown in pseudocolor in Sup. Fig. 1?

We calculated the similarity of a pair of odors as the normalized dot product of the glomerular representation. Similarity of 1 indicates that odors have identical odor representations up to a scalar difference. In the revised version of the manuscript we used odor similarity matrices in **Figure 1 O-M-Q** to compare target odors and mixtures of odors. We also used odor similarity in **Figure 11** to compare odor representations between WT and *Cntnap2*^{-/-} mice.

19. What does the gray scale in Fig. 1A show? Are these negative z scores? Please show a gray scale with the z score range.

The dark colors represent negative z-scores. We have added a gray scale showing the z-score range.

Reviewer #3 (Remarks to the Author):

Detecting a relevant odor (figure) in the background of less relevant odors (ground) is a critical task that animals need to solve in a natural environment. Here Li et al. investigated if mice can solve this figure-ground separation problem using a go/no-go task. The major finding is that the mice can correctly perform the task even when the identity of a background odor was switched between a training and a test phase. They presented two quantitative models that can solve the same problem based on neural responses to odors acquired through intrinsic signal imaging; however, both fell short of explaining the actual behavior of mice in a challenging condition.

Finally, the authors report that the *Cntnap2*^{-/-} mouse model of autism failed to perform the task when the background odor was switched in a test phase.

Although the authors have revealed an aspect of olfactory behavior, this represents a rather modest conceptual advancement as - have already shown in a similar scheme that mice are capable of separating figure from ground odors; the main difference is limited to the use of a novel background odor during the test.

The lack of satisfactory behavioral model and the mere description of a performance of *Cntnap2*^{-/-} mice further dampen the enthusiasm for this manuscript.

We thank the reviewer for the very insightful comments. We have addressed them by doing more analysis and experiments to determine the crucial role played by odor novelty in the computational difficulty produced by background odors in a mouse model of autism, further clarifying the novelty of our approach. We show that the difficulty produced by a novel background odor for *Cntnap2*^{-/-} mice is not solely determined by the glomerular representation of the background as shown by Rokni et al. for WT mice, (Nat Neurosci, 2014). We also show that linear classifiers, which have been proposed as setting up an upper limit to WT mice performance by Mathis, Rokni et al (Neuron, 2016) failed to generalize when novel background odors and combinations are used. Our results show that a computational strategy that works for the simpler problem of targets in known backgrounds, does not translate for the more challenging situations involving novel background odors. It is this function that is selectively affected in the *Cntnap2*^{-/-} mouse model of autism.

We have also substantially expanded the analysis and the experiments done with *Cntnap2*^{-/-} mice to better establish their differences with WT mice. We determined that it was not the glomerular representation, nor learning problems, nor the detection of the novelty, nor differences in the sniffing response that caused the problems with novel background odors in *Cntnap2*^{-/-} mice but it was the novelty itself that affected the olfactory behavior of *Cntnap2*^{-/-} mice.

Major comments:

1. The authors initially frame the work (in the title, abstract, and the introduction) as an investigation into the function of *Cntnap2*^{-/-} mouse model of autism. However, this work is fundamentally about the olfactory behavior and physiology of wild type mice. This is misleading as examination of the *Cntnap2*^{-/-} mouse is very limited and remains very descriptive.

We were interested in developing a behavior that would capture the sensitivity to novel backgrounds in a mouse model of autism. The reviewer is right in that the groundbreaking article from Rokni et al had already shown that WT mice are able to detect target odors in the presence of well-known backgrounds and explored the role of the glomerular representation of the odors in determining the difficulty of the task. However, in Rokni et al paradigm animals learned this behavior over hundreds of trials. From previous work it was unclear a) whether WT mice would be able to detect odors in novel environments and b) whether a mouse model of autism would be affected compared to WT mice performance. The revised version of the manuscript includes a better explanation of our approach (**page 1, lines 41-50**), which highlights the novelty of our work.

By applying our novel behavior to *Cntnap2*^{-/-} mice, we were able to detect an olfactory deficit in *Cntnap2*^{-/-} mice that was not apparent from differences in glomerular responses (**see Figure 10 and 11**), nor from differences in learning rates (see **Figure 12D-F**). The deficit could be overcome by training with a background odor (see **Figure 12C**). *Cntnap2*^{-/-} mice were considered so far to have olfactory behavior that matched or exceeded WT mice in finding buried treats (Peñagarikano et al. 2011).

To further clarify the role of novelty in the olfactory deficits in the *Cntnap2*^{-/-} mouse model of autism we have substantially expanded the experiment and analysis of the *Cntnap2*^{-/-} mice, as follows:

- a) We have increased the number of WT and *Cntnap2*^{-/-} mice used for imaging, to perform a more thorough comparison of the glomerular responses of the *Cntnap2*^{-/-} mice with WT mice. We show that the magnitude of the glomerular responses (both in terms of average responses and fraction of activated glomeruli per odor) in WT mice are not significantly weaker in the *Cntnap2*^{-/-} mice (see **page 15, lines 657-673** and see **Figure 10 D-G**).

Figure 10. D. Average activation of the glomeruli produced by individual odors for the 5 *Cntnap2*^{-/-} mice plotted against average activation for 5 WT mice. **E.** Fraction of glomeruli activated per odor for the 5 *Cntnap2*^{-/-} mice plotted against fraction of glomeruli activated in 5 WT mice. Blue line represents the identity line. **F.** Average activation of glomeruli per odor for 5 *Cntnap2*^{-/-} mice and 5 WT mice. Each symbol is the average activation of one mouse. **G.** Fraction of glomeruli activated per odor.

b) We have analyzed the trial to trial variability of the WT mice and the *Cntnap2*^{-/-} mice (see page 3, line 103:131, Methods: Measurement of trial to trial variability in WT mice on page 32, and Methods: Measurement of trial to trial variability in *Cntnap2*^{-/-} mice on page 34 to see how the variability was estimated). Consistent with a previous report (M. Geramita and Urban 2017), there was increased trial-to-trial variability in the *Cntnap2*^{-/-} mice compared to WT mice (see Figure 1G-H and Figure 10B-C). The variability was also higher in the *Cntnap2*^{-/-} mice after subtracting the common variability across glomeruli (see Methods: Measurement of uncorrelated component of trial to trial variability in WT mice on page 33, Methods: Measurement of uncorrelated component of trial to trial variability in *Cntnap2*^{-/-} mice on page 34, and see Supplementary Figure 11).

Figure 1. G. Single trial responses for individual odors. **H.** Average z-score indicating the periods that were used to quantify the odor response. The air baseline period is also indicated as well as the z-score threshold (-0.46) used to detect glomerular responses. **Figure 10 B.** Examples responses as z-scores from individual glomeruli of a *Cntnap2*^{-/-} mouse on individual odor presentations and as averages. **Figure 10 C.** Average response per odor plotted against the standard deviation calculated over trials for 5 *Cntnap2*^{-/-} mice (red dots) and 5 WT mice (black dots). The magenta lines are the fitted functions for both genotypes used to estimate the coefficient of variation and imaging noise levels.

Supplementary Figure 11

Intrinsic glomerular response in awake *Cntnap2*^{-/-} mice had correlated fluctuations. **A.** Example of an image of average z-score responses to 22 presentations of ethyl butyrate

(0.1%) in an awake *Cntnap2*^{-/-} mouse. The odor was presented as a 9 second pulse. Odor responses are the average z-score calculated using the period between 2 and 9 seconds following odor onset and using the 5 seconds preceding the onset of the odor as the baseline. **B.** Example of trial by trial correlated odor evoked responses of two glomeruli located on different spots. **C.** Glomerular responses for 132 glomeruli for 2 different odor mixture presentations plotted against the average response over 22 odor presentations. Solid lines are the least square linear fit. **D.** We calculated the deviations from the best fitted line for 2980 ROI-odor pairs recorded from 5 *Cntnap2*^{-/-} mice and plotted in red the standard deviations of their deviation from the best fitted line (σ_{uncorr}) against the average response for that ROI-odor pair (μ). The black dots correspond to data from the WT mice. *Cntnap2*^{-/-} mice had higher uncorrelated trial-to-trial variability compared to WT mice.

- c) We have expanded the number of WT mice (5 mice) and *Cntnap2*^{-/-} mice (5 mice) used for imaging. The larger number of animals of both genotypes has permitted us to make comparisons between correlation coefficients of odor similarities between individual animals. Briefly, odor pairs that were similar for WT mice were also similar for *Cntnap2*^{-/-} mice. We have compared correlation coefficients within genotypes and across genotypes. Individual mice odor similarity matrix coefficient of correlation were equally correlated when compared within genotype as when compared across genotypes (see page 16, lines 674-699 and see Figure 11 D-G).

Figure 11. Pairs of odors that evoked similar activity patterns in WT mice also evoked similar patterns in *Cntnap2*^{-/-} mice. **D.** Patterns of glomerular activation using ROI from example individual animals of each genotype. **E.** Odor similarity matrices for the example *Cntnap2*^{-/-} mouse and WT mouse. **F.** Similarity between 190 odor pairs calculated using the example WT mouse data against the similarity from the example *Cntnap2*^{-/-} mouse. **G.** Distribution of linear correlation coefficients of odor similarities (r) for pairs of animals of the same genotype (WT vs WT, 10 animal pairs and *Cntnap2*^{-/-} vs *Cntnap2*^{-/-} 10 animal pairs) and for pairs of animals of different genotypes (WT vs *Cntnap2*^{-/-}, 25 animal pairs). Error bars represent the mean \pm s.e.m. Comparison between correlation coefficients were done using a t-test.

- d) We have compared the performances of the linear classifiers and the NNC using the glomerular data of the *Cntnap2*^{-/-} and the WT mice. The increased trial-to-trial variability of the *Cntnap2*^{-/-} mice glomerular responses was not large enough to affect the

performance of the linear classifiers, NNC, nor the Lasso deconvolution (see **Increased trial-to-trial variability in glomerular activity in *Cntnap2*^{-/-} did not significantly reduce performance of algorithms** on page 16 and see **Figure 11 H-I**).

Figure 11. H. Performance (mean±s.e.m) of SVM, logistic, and NNC classifiers calculated using *Cntnap2*^{-/-} and WT mice glomerular activation data for target detection in novel environments trained with the full training set and tested with the full training set. Symbols represent average performance per animal. **I.** Performance (mean±s.e.m) of the Lasso using *Cntnap2*^{-/-} and WT mice glomerular data for the reduced test set using different sizes of dictionaries.

- e) Although our imaging results indicate that odor identification in novel backgrounds should have been very similar between *Cntnap2*^{-/-} mice and WT mice, there was a significant behavioral deficit in the *Cntnap2*^{-/-} mice. We have directly tested that novelty of the background was the key element that affected performance in *Cntnap2*^{-/-} mice. We performed new experiments in a new cohort of *Cntnap2*^{-/-} mice and WT mice. We show that when a novel background odor is introduced gradually, performance is similar between the *Cntnap2*^{-/-} mice and the WT animals. However, the same background odor produced a much lower performance in the *Cntnap2*^{-/-} mice compared to WT mice when the background odor was introduced suddenly at a higher concentration as a novel background odor. This new experiment shows that it was not only the amount of glomerular representation overlap between a background odor and the target that determined the difficulty of a task for *Cntnap2*^{-/-} mice (see **Fig. 12 C** and **page 18, line 759-786**).

Figure 12. C. Performance of butyl propionate when it was used as a novel and as a standard background for different cohorts of *Cntnap2*^{-/-} mice and WT mice. The

performance of four *Cntnap2*^{-/-} mice performance was at chance level when butyl propionate was a novel background odor, but significantly increased for the two *Cntnap2*^{-/-} mice where butyl propionate was used instead of (s)-(-)-limonene as a standard background during training.

- f) We have also further analyzed the learning rates during training of the *Cntnap2*^{-/-} mice and compared them to WT mice. We compared the learning rates when *Cntnap2*^{-/-} and WT mice learned to discriminate the target odors without backgrounds (see **Figure 12D**). We also compared the learning rates when target odors were presented for the first time together with low concentration background odors (see Figure 12E), and when they learned to discriminate the target with the backgrounds at high concentration (see Figure 12F). Learning rates were not different between *Cntnap2*^{-/-} and WT mice (see section **Cntnap2^{-/-} mice odor learning rate is similar to WT mice** on page 18).

Figure 12. D. Learning curves for the second pair of go target and no-go target at a high concentration (0.1%) after animals had already learned to discriminate the first pair of odors at that concentration. **E.** Learning curves for the first exposure of the standard background odors at 0.025% (1/5 of the final concentration) with the targets at their final concentration (0.025%). **F.** Learning curves for the first exposure of the standard background odors at the final concentration of the background odors (0.1%).

The authors repeatedly note that they decided to focus on *Cntnap2*^{-/-} mice because they have reduced neural activity as well as inhibition, and these properties might compromise olfactory discrimination. However, the link remains weak; neither the effect of change in neural activity/inhibition on the behavior nor the state of neural activity/inhibition in *Cntnap2*^{-/-} mice in the target olfactory areas are tested. I would suggest to reframe the abstract and introduction.

We agree with the reviewer that there are potentially multiple, non-mutually exclusive circuit deficits that might account for the deficit we discovered in *Cntnap2*^{-/-} mice for novel background odors. We have moved the presentation of hypothesis regarding circuit mechanisms from the abstract and introduction to the discussion, page 23, lines 1025-1027.

“*Cntnap2*^{-/-} mice have multiple circuit deficits that might affect their capability to detect odors in the presence of strong novel background odors. The lack of *Cntnap2* causes broadening of the transmitted action potential resulting in enlarged neurotransmitter release (Scott et al. 2019) which might cause excessive activation by strong background odors, masking the target odors. However, *Cntnap2*^{-/-} mice differences in performance with respect to WT mice on novel

background odors became only apparent when novel background odors were presented as catch trials among trials with other well-rehearsed background odors and not when *Cntnap2*^{-/-} mice trained over days with the same background odor. The increased release of neurotransmitters would affect both situations when a background odor was novel and when it was a standard background odor. whereas the deficit in *Cntnap2*^{-/-} mice was selective for novel background odors.

Cntnap2^{-/-} mice could be using individual glomeruli or linear classifiers to identify target odors in known background odors, as these algorithms reached performances that matched *Cntnap2*^{-/-} mice behavior for known background odors but did not produce good behavior for novel background odors, similar to *Cntnap2*^{-/-} mice. These algorithms do not require inhibitory activity and its performance would not be affected in *Cntnap2*^{-/-} mice because of their reduced expression of GABA (Peñagarikano et al. 2011). In contrast, deconvolution algorithms that approximate the Lasso behavior require computations (Koulakov and Rinberg 2011; Grabska-Barwińska et al. 2017; Li and Hertz 2000; Otazu and Leibold 2011) including normalization (Carandini and Heeger 2012) which might make them more likely to be affected by reduced inhibition. Changes in cortical spine development (Anderson et al. 2012) and reduced local spine density in cortex (Lazaro et al. 2019) in *Cntnap2*^{-/-} mice might also affect the capability to integrate information over multiple glomeruli which is necessary to identify odors in novel background odors.”

2. The performance of a linear classifier is not correlated with that of the animal behavior, and the explanatory power of the other nearest neighbor classifier (NNC) is limited to a particular context (16 mixture testing and not 8 mixture testing). Negating models certainly narrows down the actual algorithms implemented in the brain, but one would obviously prefer to see a model that explains the experimental findings in a work.

We initially focused on linear classifiers because they had been proposed in the literature as setting an upper limit on WT mice performance. However, in our experiments, the linear classifiers were not the best descriptor of mice performance nor they could match the performance of WT mice when the diversity of the training data was reduced in half. We think that it was important to extensively test the linear classifiers as they were thought to represent an upper bound for WT mice behavior.

In the paper titled “Reading Out Olfactory Receptors: Feedforward Circuits Detect Odors in Mixtures without Demixing”, Mathis, Rokni et al (Neuron, 2016) conclude that “In this work, we have linked experimentally measured glomerular responses to behavioral data in an odor demixing task. Using realistic assumptions about neuronal noise, and nonlinear interactions for mixtures, we show that the information about odor mixture components at the level of olfactory receptors **is already linearly separable and does not require any preprocessing or inference algorithms that rely on prior information and feedback circuits**”.

In our manuscript, linear classifiers (SVM 70.1± 2.2%, logistic regression 70.1%± 2.5%) had similar performances as the nearest neighbor classifier (70.8±2.9%) and approximate the performance of the WT mice (74%, see **Linear classifiers and nearest neighbor classifier could identify target odors in novel background odors** on page 7 and see **Figure 3**).

Figure 3 Linear classifiers and nearest neighbor classifiers could identify odors in novel environments. **A.** Example of the average glomerular responses of a WT mouse to the 16 mixtures of the training set and responses to 16 mixtures of the test set with 4-methylanisole as the novel background. **B.** Estimated weights of SVM with linear kernel and logistic linear classifiers trained using the training set for the example. **C.** Output produced by multiplying the weights of the logistic linear classifier with glomeruli activation from the odor mixtures of the training set (blue squares)

and the test set (red squares) for the example. **D.** Performance of the logistic regressor calculated on data from 6 WT mice, 32 recording sessions on the test set, with the performance of each recording day using 100 instantiations of each of the responses of the test set using equation 1. Error bars are s.e.m. and p-values were calculated using a t-test. **E.** Same as **C** but using the SVM weights. **F.** Performance of the SVM for the data from 6 WT mice, 32 recording sessions. **G.** Example of a matrix of dot products between the training set and the test set used for calculating the nearest neighbor classifier (NNC). The red squares indicate the location of the most similar mixture from the training set for each mixture in the test set. If the valence (go or no-go) of the target odor in the test mixture matched the valence of the most similar mixture (Nearest Neighbor), the trial was considered correct. **H.** The performance of the NNC for novel background odors for 6 WT mice, 32 recording sessions. Error bars are s.e.m.

However linear classifiers were not as good at predicting WT mice performance of individual novel background odors compared the to the Nearest Neighbor Classifier (NNC), a non-linear classifier, when considering all available glomeruli (see **Classifiers that included more glomeruli had higher correlation with animal behavior** on page 12 and see **Figure 8G**).

Figure 8. G. Classifier’s correlation with behavior as a function of the number of ROI included. Error bars represent the standard deviation of the distribution of linear correlation coefficients (vertical) and number of glomeruli(horizontal) included calculated using a Montecarlo simulation with 500 repeats. The asterisks indicate the results of t-test comparing the distribution of correlations using all available glomeruli against the distribution of correlations using the reduced number of glomeruli.

Although there has been broad interest in generative models as potential mechanisms used by the olfactory system (Koulakov and Rinberg 2011; Grabska-Barwińska et al. 2017; Li and Hertz 2000), Mathis, Rokni et al concluded that “... mice cannot significantly benefit from a generative model by which they could task-independently decompose odor mixtures into their individual odor component.”. They based this statement in the results of an experiment where performance of WT mice trained with individual odors (go odors and no-go odors) failed to generalize and reach the performance of a linear classifier when challenged with odor mixtures of those odors (see **Figure 4 D from Mathis et al 2016**).

Figure 4 D from Mathis et al 2016 Average performance curves for OLE per target pair when trained only on 80% of the single odor stimuli (yellow, OLE₁) and on 80% of all the data (blue, OLE₁₄) based on 421 glomeruli. The red curve shows the average performance \pm SEM of five mice trained on single odors and subsequently tested on mixtures of 1, 3, 8, and 14 odors (Mice₁).

We do not know why mice in Mathis et al (2016) did not generalize beyond the performance of a linear classifier as our WT mice did (see **Figure 9C**). In our manuscript we speculate that by having our mice train with mixtures, the task contingency (go target odor present \rightarrow lick) was easier to be inferred by the mice permitting to uncover behavior that exceed the performance of linear classifiers as shown in **Figure 9C**. We have added a paragraph in the discussion describing previous research on mouse generalization and how our training approach uncovered behavior that exceeded the limits of the linear classifier (**page 22, lines 970-982**).

Figure 9C. Performance using imaging data of linear classifiers and NNC, when trained with the reduced training set and tested with the reduced test set were compared to the performance when trained with the full training set and tested with the full test set. Imaging data was obtained from 32 recording sessions using real mixtures (6 WT mice, 161.3 ± 39.8 glom per session). Classifiers were created using the average training set (full and reduced) and the performance was calculated for each recording session by generating 100 instantiations of the test set (full and reduced). Comparison between algorithm performances trained using the full and reduced, as well as comparisons of the reduced set performance against chance level were done using a t-test. Performance of a new cohort of three WT mice that were also trained with the reduced training set and tested with the reduced test set in the asynchronous task was compared to four WT mice that had trained with the full set and were evaluated with the full set. The p-value for the behavior comparison was calculated using the Fisher exact test.

When we reduced the diversity of the training set, WT mice performance was less affected compared to the linear classifiers and the NNC. In the revised version of the manuscript we have tested whether a representative deconvolution algorithm, the Lasso, would show improvement with respect to the linear classifiers or the NNC. The Lasso decomposes an observed signal into contributions selected from a large dictionary of known odors, while minimizing the number of odors used, permitting generalization to multiple combinations of dictionary odors. Although the large dictionary of odors, which represent the odors that the mice have stored, is hard to estimate, we have used multiple simulations with dictionaries of different sizes that had the same mean and covariance as measured odor responses, but that did not include the novel background odor.

All these implementations of the Lasso clearly outperformed linear and the nearest neighbor classifiers. Deconvolution algorithms constitute a plausible algorithm given our experimental data. Our study is the first one to have directly evaluated deconvolution algorithms with behavioral data (see section **A sparse deconvolution algorithm has better performance than NNC and linear classifier for reduced training data** on page 14 and **Figure 9D-F**).

Figure 9. D-F. Sparse representation algorithm, Lasso, identifies the odors present in a mixture. Mixture of a go target odor (isopropyl butyrate), contextual background odor (isoamyl acetate), and a novel background odor that was not part of the dictionary used by the Lasso. **E.** Concentrations assigned by the Lasso for a dictionary size of 500 elements. Bars correspond to the estimated concentrations of the four contextual background odors, the four target odors, and (s)-(-)-limonene. Stems correspond to estimated concentrations for the randomly generated dictionary elements. The maximum of the estimated concentration was larger for the two target go odors compared to the two target no-go odors, so this would be considered a correct identification. **F.** Performance of the Lasso calculated for different size of dictionaries was compared to the performances of the NNC, SVM, and logistic regression. Individual symbols represent average performances for the 5 WT mice. Error bars represent the 10-90% percentiles of the 13200 simulations performed per dictionary size and the SVM, logistic regression and NNC. Significance was calculated using a t-test and comparing the average performance per animal between the NNC, SVM and logistic regression against the Lasso performance for different dictionary sizes.

3. Linear classifier and NNC are trained using both virtual and actual odor responses. If actual odor responses are available, how come the main analyses in Figures 5 and 6 were conducted with virtual responses (There have been various reports on the linear and non-linear additivity of component responses in the bulb)? How did virtual and actual odor responses compare using intrinsic imaging?

We agree with the reviewer that actual odor responses better reflect the reality of the difficulty of the task that the mice are challenged with during the behavior, so we have conducted the analysis of Figures 8 and 9 (5 and 6 in the previous version of the manuscript) using the only the glomerular responses to odor mixtures.

In order to do that, we have performed additional imaging experiments where we have collected data from more WT mice using actual odor responses. We performed 32 recording sessions from 6 WT mice (161.3±39.8 glom per session, mean±s.d, 5163 glomeruli) where we recorded training set mixtures that included (s)-(-)-limonene and the corresponding mixtures that were part of the test set, where (s)-(-)-limonene was replaced by one of the 10 novel background odors.

Figure 8 also includes now the correlation between algorithms output and behavior, showing that although WT mice imaging data reached a plateau performance when using between 24-36 glomeruli, the correlation with behavior increased as more glomeruli were included.

○ Synchronous (4 WT mice)
 ○ Asynchronous (4 WT mice)

Figure 8. Classifiers that included more glomeruli had higher correlation with animal behavior. **A.** Performance of the NNC for 10 novel background odors was calculated using all the available glomeruli per imaging session (161 ± 40 ROI, mean \pm s.d, 32 imaging sessions, 6 WT mice). Black circles y coordinates represent the average behavioral performance on a novel background odor from four WT mice that performed the asynchronous task and red circle's y coordinate represent the average performance of the four WT mice that performed the synchronous task. The blue line is the linear correlation between imaging data and behavior. **B-C** Similar plot for logistic regression and SVM. **D-F. Reducing the number of glomeruli reduced correlation with behavior for all three classifiers.** **G.** Classifier's correlation with behavior as a function of the number of ROI included. Error bars represent the standard deviation of the distribution of linear correlation coefficients (vertical) and number of glomeruli (horizontal) included calculated using a Montecarlo simulation with 500 repeats. The asterisks indicate the results of t-test comparing the distribution of correlations using all available glomeruli against the distribution of correlations using the reduced number of glomeruli.

By using the real mixtures data, we show that linear classifiers and the NNC were more sensitive to the diversity of the training data compared to the WT mice. In fact, linear classifiers and NNC were at chance levels when trained with the restricted data using actual mixtures (see **Figure 9C** above).

4. There is a text saying that "ROI were drawn manually using ImageJ over activated glomeruli across all odors presented (line 706)", but there is no description on the definition of activated glomeruli, where trial-to-trial variability, area of ROIs, criteria for separating neighboring ROIs, etc are likely to be important. This needs to be reported.

The revised version of the manuscript includes a more thorough analysis of the ROIs and the associated trial to trial variability analysis. We have used the measured trial to trial variability of the glomerular responses to estimate the sources of glomerular variability (see **Figure 1 G-H** and **page 3, lines 111-138**). Briefly, there are two types of variability that affect our measurements. One type of variability is related to biological variability, which is proportional to the average glomerular response and appears as a coefficient of variation (CV). The second one is related to imaging noise (σ^2_{noise}) and is present even in the absence of an odor evoked response. Using the data from 2684 ROI-odor pairs from 3 WT mice (see **Figure 1I**), we estimated the coefficient of variation CV and the σ^2_{noise} by fitting the function $\sigma^2(\mu) = \sigma^2_{\text{noise}} + CV^2\mu^2$. The estimated coefficient of variation (CV) was 0.34 (95% CI: 0.30 - 0.37) which is similar to the value of 0.37 ± 0.07 (mean \pm SD) estimated using calcium imaging in anesthetized mice (Mathis et al. 2016).

Figure 1 G. Single trial responses for individual odors. **H.** Average z-score indicating the periods that were used to quantify the odor response. The air baseline period is also indicated as well as the z-score threshold (-0.46) used to detect glomerular responses.

Figure 1 I. Average odor response versus trial-to-trial variability. Trial-to-trial variability was the combination of a component that scaled with the average odor response plus a constant. Purple line indicates mean fitted function and dotted lines are the 95% confidence intervals of the fit.

This coefficient of variation includes both uncorrelated fluctuations of individual glomerular responses as well as fluctuations that are correlated across the whole glomerular population. WT mice performance might be mostly limited by the uncorrelated variability because, by having access to all set of glomeruli, mice could compensate for the correlated fluctuation of the whole population (Mathis et al. 2016). Therefore, we calculated a coefficient of variation CV_{uncorr} for the uncorrelated fluctuations (see **Supp. Fig. 3**) after subtracting the population response fluctuations. CV_{uncorr} was 0.25 (95% CI: 0.23 - 0.27). We have used the estimated CV_{uncorr} for creating noisy versions of single presentation that emulate the trial to trial variability that mice experience in order to evaluate different algorithms.

WT

Supplementary Figure 3

Intrinsic glomerular response in awake WT mice had correlated fluctuations. **A.** Example of an image of average z-score responses to 27 presentations of a mixture of ethyl caproate (0.1%), (s)-(-)-limonene (0.1%), and isobutyl propionate (0.025%) in an awake WT mouse. The odor was presented as a 9 second pulse. Odor responses are the average z-score calculated using the period between 2 and 9 seconds following odor onset and using the 5 seconds preceding the onset of the odor as the baseline. **B.** Example of trial by trial correlated odor evoked responses of two glomeruli. Linear correlation was significant ($R=0.66$, $p=0.0008$, linear correlation coefficient). **C.** Glomerular responses for 156 glomeruli for 2 different odor mixture presentations plotted against the average response over 27 odor presentations. Solid lines are the least square linear fits. Glomerular odor response variations had population correlated fluctuation and an uncorrelated fluctuation. The population correlated fluctuation is given by the least squares linear fits, whereas deviation from the least squares linear fits correspond to the uncorrelated fluctuation. **D.** We calculated the deviations from the least squares fitted line for 2684 ROI-odor pairs recorded from 3 WT mice and plotted the standard deviations of their deviation from the best fitted line (σ_{uncorr}) against the average response for that ROI-odor pair (μ). The total variance σ_{uncorr}^2 is given by the sum of an odor independent variance $\sigma_{\text{uncorr noise}}^2$ and a response dependent variance. The purple line is the fitted line and the dotted lines are the 95% confidence interval of the fit.

To determine the threshold for considering a glomerulus to be activated, we used the imaging noise (σ_{noise}^2) as well as n , the number of odor presentations to determine the threshold of activation for a mean glomerular response to be considered different from zero.

Page 4, lines 132-138: “The estimated trial to trial variability associated to the imaging noise σ_{noise} was 1.59 z-score (95% CI: 1.51 - 1.66). We averaged responses over n trials, so this noise would appear as a standard error of the mean around zero of $\sigma_{\text{noise}}/\sqrt{n}$. This s.e.m determines our threshold to detect mean glomeruli activation that was reliably different from zero. We used at least $n=16$ trials so the standard error of the mean was <0.42 . Glomeruli-odor responses that had average z score of -0.42 or larger were considered non-responsive. This threshold was smaller than the responsive ROI-odor responses (-1.37 ± 1.17 z-score, mean \pm s.d., $n=6477$ ROI-odor pairs, 5 WT mice). “

We have further confirmed that the weak glomerular activity that we measured using intrinsic imaging also propagated postsynaptically using the Thy1- GCaMP6F mouse (Dana et al. 2014) that expresses GCaMP6F in the mitral and tufted cells. Glomeruli identified with intrinsic imaging colocalized to glomeruli identified using the fluorescence signal (see **Supp. Fig. 4**).

Supplementary Figure 4

Glomerular activation measured with intrinsic signal correlate with increases in fluorescence in Thy1-GCaMP6f mouse. We implanted a C57BL/6J-Tg(Thy1-GCaMP6f) GP5.11Dkim/J (Dana et al. 2014) mouse with a window over the olfactory bulb as described above. **A**. We presented 20 odors using 9 second odor pulses at the concentrations used for the behavior in an awake mouse and recorded intrinsic signal using 780 nm infrared light. **B**. We also used a 470 nm blue light and recorded the green fluorescent responses that correspond to mitral and tufted cells, the output neurons of the olfactory bulb. **C,D**. Examples of average z-score of the intrinsic imaging averaged over 2 to 9 seconds from odor onset. The negative deflections in odor evoked glomerular responses in the intrinsic signal colocalized with odor evoked positive deflections in the fluorescence signal. The fluorescent signal had larger magnitude compared to the intrinsic signal. **E**. Examples of time course responses from individual glomeruli to 9 second odor pulses that start at 0 s. Even intrinsic signals with relatively small z-scores (see ROI 46 in response to ethyl

propionate) resulted in increases in the fluorescence signal. **F.** Relationship between the ROI-odor intrinsic signal, defined in the intrinsic signal, and the fluorescence signal. Black dots represent individual ROI-odor pairs and circles are the mean of data binned according to the z-score of the intrinsic imaging. The error bars are the standard deviation of the fluorescence signal of the ROI-odor pair in a bin. We compared the fluorescence signal of the binned data according to the z-score with the distribution of fluorescence in the absence of odor responses (noise distribution). The responses were significantly different from the noise distribution for responses with a mean z-score response of intrinsic imaging of -0.3 ($p < 0.02$) or larger.

Furthermore, the effect of neglecting the glomeruli that did not pass the criteria (in other words, the effect of noise) on the classification should be investigated.

We have also tested whether changing the z-score threshold affected the performance of the linear classifiers or the NNC. Perturbations of the z-score threshold around the established $z = -0.42$ did not affect the performance of the algorithms as the threshold (-0.42) was much smaller than most ROI-odor responses (average -1.37 ± 1.17 z-score, mean \pm s.d., $n = 6477$ ROI-odor pairs, 5 WT mice, see **Supp. Fig. 6**).

Supplementary Figure 6

Performance of linear classifiers and NNC for odor identification in novel background odors were robust to changes in the threshold of glomerular responses using intrinsic imaging. **A.** Average number of glomeruli-odor responses included per recording session as a function of the threshold of z-score responses. Error bars are the s.e.m for 32 recording sessions (6 WT mice, 5163 ROI). The number of ROI-odor responses fluctuated between -16% and $+17\%$ as the threshold for z-score responses was changed between -0.1800 (most permissive) to -0.66 (most restrictive). The reference z-score threshold was -0.42 which resulted in 72885 odor-ROI responses. **B.** Performance on individual recording sessions of the linear classifiers and NNC as a function of the z-score threshold. Error bars are s.e.m. The performance of the NNC, and the linear SVM and logistic regression were not affected by changes in the threshold of the z-score.

We also have added supplementary material 1 regarding physical description of the drawn ROIs and a test we performed to determine whether neighboring ROIs included signals from common glomeruli.

Supplementary material 1: Physical characteristics of drawn ROIs

The average area of the drawn ROIs was $1030.7 \pm 7.9 \mu\text{m}^2$ (mean \pm s.e.m, n=5590 drawn ROIs, 11 WT mice). To determine whether the responses of single ROIs included mixed signals originating from multiple glomeruli, we analyzed the responses from nearby ROIs. Out of 35594 simultaneously recorded ROI pairs, there were 351 ROI pairs whose centers were closer than 50 μm or less. These neighbor ROIs could potentially include an overlap of signals originating from neighbor glomeruli. However, these neighbor ROIs responded to different sets of odors, with each pair of ROIs having 7.6 ± 3.1 odors (mean \pm s.d, 351 ROI pairs) with significant responses (z_score of -0.42 or less) for one ROI and not the other. There was only 1 pair of neighbor ROIs that were responsive to the same set of odors. Responses of neighbor ROIs were different, indicating that their signals originated from different glomeruli.

5. The text should be proof read. There were so many typos, grammatical errors etc.

We have proofread the manuscript. We apologize to the reviewer.

Minor comments

6. How did animal and model performance differ depending on the identity of target, non-target, and novel background odors? Did neural responses to these odors explain the behavior? (All the figures lump the results for all the odors)

For this manuscript, we wanted to characterize the effect of the presence of novel background odors in target identification. As we have used the same go-target odors, no-go target odors, and contextual background odors, performance differences could be ascribed to the identity of the novel background odors.

Because we are studying novelty, the number of odor presentations that include a novel background odor is limited to 8 per animal. The number of possible mixtures that a novel background odor could appear is 4 target odors * 4 contextual background odors, resulting in 16 possible mixtures. As we have recorded behavioral responses from 8 WT mice, and each novel odor was presented 8 times, there are at most, $8 \times 8 / 16 = 4$ presentations to estimate the performance of individual elements of the mixture, which is not sufficient.

Although the interaction between the known background odors and novel background odors is an interesting question, it would require the use of known odors that produce a larger range of behavioral performances. From analyzing the performances for the training mixtures, the range of performances produced by the identity of the contextual background odor used was limited to at most 6% (see **page 8, lines 343-348** for the asynchronous case and **page 10, line 437-443** for the synchronous case).

7. Line 183. "Surprisingly, there was no increase in performance even when WT mice recognized the novel background odor,...". Please present the data showing the relationship between the sniff and the animal's performance.

In the revised version of the manuscript **Figure 5C** shows that the performance of the WT mice is above chance on the first presentation of a novel background odor on the asynchronous case and does not increase with repeated (up to 8) presentation of a novel background odor. In contrast, the sniff rate in response to the novel odors (**Figure 6C**) shows that the increase in sniff rate produced by the novel background odor returns to baseline after the first exposure to the odors, indicating some sort of recognition of the background odor. The same relationship is present for the synchronous case (**Figure 7E and F**).

Figure 5C. Performance of the novel background odor as a function of the novel background odor presentation (8 presentation total over two days). Symbols represent individual animals. P-value was calculated for the first presentation to determine whether performance was different from 50% (chance) using a binomial test. **Figure 6C.** Mean±s.e.m of the increase in sniff rate for novel odor compared to preceding trial with (s)-(-)-limonene as a function of the novel background odor presentation (8 presentation total over two days). P-values were calculated using the Bonferroni corrected t-test. **Figure 7E.** Performance of the novel background odor as a function of the novel background odor presentation **F.** Increase in sniff rate for novel background odor compared to preceding trial with (s)-(-)-limonene as a function of the novel background odor presentation.

8. Line 273, 303 etc. How come only recurrent computation is discussed? For example, temporal integration can equally be possible.

We have added temporal integration as another possibility for a strategy used by the animals to identify odors in novel environments in the discussion (page 23, line 997-999).

9. Line 355. "..., showing that adaptation was not necessary in novel background odors." The statement should be softened as adaptation was not confirmed using imaging.

The reviewer is right as we do not have imaging data of the neural responses across the olfactory bulb/olfactory cortex to determine the degree of adaptation along the olfactory pathway. We have rephrased the wording to reflect that previous exposure to the odor is not necessary to identify odors in novel environments (see **page 12, lines 495-497** and **page 22, lines 936-938**).

10. Line 360. “*Cntnap2*^{-/-} mice might rely more on sniffing induced suppression as a mechanism to identify odors in novel backgrounds”. What is the rationale behind this?

We have removed the statement. In the revised version of the manuscript we have done a more thorough comparison of *Cntnap2*^{-/-} mice sniffing responses compared to WT. Although *Cntnap2*^{-/-} mice do not elevate their sniffing responses in response to the target odors as WT mice do for known background odors, their sniffing modulation produced by novel background odors is identical as the WT mice. This shows that fast sniffing of novel background odors is not sufficient in *Cntnap2*^{-/-} mice to match the performance of the WT mice (see section ***Cntnap2*^{-/-} mice did not increase their sniff rate for target odors in known backgrounds** on page 19, section ***Cntnap2*^{-/-} mice increase in sniffing rate for novel background odors was similar to WT mice** on page 20, and see Fig 12I).

Figure 12 I. Sniff responses for *Cntnap2*^{-/-} and WT mice for novel and standard background odors. Lines represent mean±s.e.m.

11. Line 367. “WT mice might achieve faster reaction times using glomeruli with higher affinity for the presented odors that might be located in the ventral surface of the olfactory bulb”. Again, what is the rationale behind this statement? In general, there are too much speculative descriptions in the Discussion.

In the revised version of the manuscript we have removed references to the role of the ventral glomeruli in reaction time.

12. Line 378-384. If the olfactory tubercle was performing linear computations in case of discriminating known odor mixtures, what is the possible mechanism that allows the switching of processing between the tubercle and the piriform cortex? Of the authors were to touch upon the involvement of the tubercle, it might be useful to describe some hypotheses.

We think that the activation of mitral cells, which are sensitive for novel odors might recruit circuits that could perform recurrent computations.

Page 23, line 1002-1005: "The additional mitral cell activity produced by novel odors (Kato et al. 2012) might recruit piriform cortex, which given their association fiber systems, might implement recurrent computations(Haberly and Price 1978) required for identifying odors in novel backgrounds."

REVIEWER COMMENTS

Reviewer #1 (Remarks to the Author):

The authors have addressed my key comments well and improved on the presentation. The new data and in particular extended methodological descriptions are appreciated.

My only comment at this point would be that the authors should revisit their references including (more) work that has appeared recently, since their original submission. This includes (to name only a few) e.g. Burton et al 2021 (PMID: 35861321) on glomerular representation, Lebovich et al (PMID: 34871306) on background-foreground separation, Ackels et al 2021 (PMID: 33953395) in the context of odor source separation, or Nakayama et al 2022 (PMID: 35784646) as a recent novel olfactory paradigm.

Reviewer #2 (Remarks to the Author):

The manuscript by Li and co-workers is vastly improved and makes a solid contribution to the understanding of odorant signal processing. The comparing the performance of mice identifying odorants in novel olfactory environments and the performance of algorithms decoding odorant identification is informative. The finding that *Cntnap2*^{-/-} mice are not able to identify odorants in a novel environment is interesting. I have a few suggestions for edits.

1. Line 30. Please change “Glomeruli activation” to “Glomerular activation”
2. Line 46. Change “ a mouse model of autism” to “ the *Cntnap2*^{-/-} mouse model of autism”
3. I like your new analysis of decoding performance with different number of ROIs in Fig. 4F. Would it be better to show the x axis on a log scale? Could you test the statistical difference between nearest neighbor and the other decoding schemes?

Reviewer #3 (Remarks to the Author):

Major comments:

In a revised manuscript, the authors report the result of additional data analyses on the WT glomerular activity and imaging of odor-evoked glomerular activity in *Cntnap2*^{-/-} mice. A major progress is the identification of a sparse deconvolution model (Lasso) that can explain the behavioral performance of WT mice in novel background. However, an important missing analysis is the application of Lasso to glomerular activity in *Cntnap2*^{-/-} mice. If the model outperformed the actual behavior of *Cntnap2*^{-/-} mice, this would suggest that the model does not likely reflect the computation employed by the brain. Therefore, this needs to be addressed.

The authors found that the only difference between glomerular responses to odors in WT and *Cntnap2*^{-/-} mice is the trial-to-trial variability, and this has not been linked, even using a model, to poorer behavioral performance in the mutant mice.

Minor comments:

The text still contains many incorrect and awkward expressions that need to be revised.

Please explain Figures 2E, 3C, 3E in more detail. What do different rows correspond to? What is perf?

More detailed explanation of Lasso analysis is warranted (Including explanation of Figures 9D, 9E).

Panels in Figure 11 are too small to read.

REVIEWER COMMENTS

Reviewer #1 (Remarks to the Author):

The authors have addressed my key comments well and improved on the presentation. The new data and in particular extended methodological descriptions are appreciated.

My only comment at this point would be that the authors should revisit their references including (more) work that has appeared recently, since their original submission. This includes (to name only a few) e.g. Burton et al 2021 (PMID: 35861321) on glomerular representation, Lebovich et al (PMID: 34871306) on background-foreground separation, Ackels et al 2021 (PMID: 33953395) in the context of odor source separation, or Nakayama et al 2022 (PMID: 35784646) as a recent novel olfactory paradigm.

We thank the reviewer for the suggestions regarding the recent and very relevant literature. We have included and discussed these new developments in regard of our work as follows:

- (page 23, line 986) A single glomerulus strategy was not effective at the micromolar concentrations that we used, where there was great overlap between the target and the backgrounds. Burton et al showed that at picomolar or lower odor concentrations, there is little overlap between glomerular representations for different odors. A single glomerulus strategy for odor identification in backgrounds might be plausible at these very low concentrations.
- (page 24, line 1027) Lebovich et al determined that the effect of background odors in decision making could be reflected as increased noise in a drift diffusion model. This would result in reduced performance with paradoxical reduction in reaction times. Interestingly, we found that the *Cntnap2*^{-/-} mice glomerular responses have increased variability in the glomerular responses. This neural variability was not large enough to affect performance. However, increased variability in other downstream circuits produced by absence of CNTNAP2 might result in the lower performance and reduced reaction time observed in *Cntnap2*^{-/-} mice.
- (page 22, line 948) Ackels et al showed that mice are sensitive to fast differences in temporal profile between odor streams which could be used to separate odor sources. We have not explored the role of these fluctuations extensively for odor identification in novel environments. The only temporal difference that we used was a 750 ms onset difference between the novel background odor and the target in the asynchronous task. This temporal difference did not produce a significant increase in performance compared to when the odors started synchronously.
- (page 6, line 222) Nakayama et al showed that the presence of background odors also increases the perceptual similarity between the mixtures potentially increasing the task difficulty.

Reviewer #2 (Remarks to the Author):

The manuscript by Li and co-workers is vastly improved and makes a solid contribution to the understanding of odorant signal processing. The comparing the performance of mice identifying odorants in novel olfactory environments and the performance of algorithms decoding odorant identification is informative. The finding that *Cntnap2*^{-/-} mice are not able to identify odorants in a novel environment is interesting. I have a few suggestions for edits.

We thank the reviewer for the evaluation of our work.

1. Line 30. Please change “Glomeruli activation” to “Glomerular activation”

We have made the change.

2. Line 46. Change “ a mouse model of autism” to “ the *Cntnap2*^{-/-} mouse model of autism”

We have made the change.

3. I like your new analysis of decoding performance with different number of ROIs in Fig. 4F. Would it be better to show the x axis on a log scale?

We have changed the x axis to a log scale.

Could you test the statistical difference between nearest neighbor and the other decoding schemes?

We changed the number of glomeruli of the nearest neighbor by selecting glomeruli based on the auROC calculated for the training set. For the linear classifiers, we used a sparseness constraint. As the number of glomeruli included do not match between the two methods, it is difficult to do a statistical comparison. However, from the log scale, it seems that the linear classifiers have higher performance with lower number of glomeruli compared to the nearest neighbor classifier.

Reviewer #3 (Remarks to the Author):

Major comments:

In a revised manuscript, the authors report the result of additional data analyses on the WT glomerular activity and imaging of odor-evoked glomerular activity in *Cntnap2*^{-/-} mice. A major progress is the identification of a sparse deconvolution model (Lasso) that can explain the behavioral performance of WT mice in novel background. However, an important missing analysis is the application of Lasso to glomerular activity in *Cntnap2*^{-/-} mice. If the model outperformed the actual behavior of *Cntnap2*^{-/-} mice, this would suggest that the model does not likely reflect the computation employed by the brain. Therefore, this needs to be addressed.

We thank the reviewer for the encouraging comments. In the revised version of the manuscript we have used the Lasso with the imaging data from five *Cntnap2*^{-/-} mice. The performance of the Lasso using *Cntnap2*^{-/-} mice imaging data in the reduced test set was $92.6 \pm 0.5\%$ (mean \pm s.d., 10 tested sizes of dictionary between 100 and 1000 elements) and it was very similar to the performance calculated for the WT mice ($93.2 \pm 0.9\%$, see Fig. 11I). For each dictionary size, we compared the performance of the Lasso using the 5 WT mice imaging data with the 5 *Cntnap2*^{-/-} mice. The performances of the animals were not significantly different for any of the dictionary sizes tested ($p > 0.5$, t-test).

Figure 11 I. Performance (mean \pm s.e.m) of the Lasso using *Cntnap2*^{-/-} and WT mice glomerular data for the reduced test set using different sizes of dictionaries. Significance was calculated using a t-test.

The authors found that the only difference between glomerular responses to odors in WT and *Cntnap2*^{-/-} mice is the trial-to-trial variability, and this has not been linked, even using a model, to poorer behavioral performance in the mutant mice.

The reviewer is right that the only difference in the glomerular response between the *Cntnap2*^{-/-} mice and the WT mice was the increased trial-to-trial variability in *Cntnap2*^{-/-} mice ($CV_{uncorr} = 0.44$) compared to WT mice ($CV_{uncorr} = 0.25$). In principle, the increased variability in the

glomerular responses should have been detrimental for the performance of the algorithms tested. However, none of the tests using linear models, NNC, nor the Lasso were affected by the enhanced variability of the *Cntnap2*^{-/-} mice (see **Supp. Fig. 13A-C**).

In order to determine the level of variability that could have affected the performance of the algorithms, we have performed simulations using the WT mice data with increasing levels of trial-to-trial variability. The performance of the algorithms on odor identification in novel environments was significantly affected by larger values of CV_{uncorr} ($CV_{uncorr} > 0.85$ for SVM and logistic and $CV_{uncorr} > 1.05$ for NNC, see **Supp. Fig. 13D-F**) compared the value observed in *Cntnap2*^{-/-} mice ($CV_{uncorr} = 0.44$).

Supplementary Figure 13

Performances of the linear SVM, logistic, and NNC on novel background odors were not affected by the higher trial to trial variability seen in *Cntnap2*^{-/-} mice. **A.** Performance of linear SVM classifier in the low variability (CV=0.25, WT like) and high variability regime (CV=0.44, *Cntnap2*^{-/-} like). The performance for each recording session was calculated by creating 100 instantiations of the test set (16 mixtures that included a novel background odor) as shown in Figure 3. To simulate the *Cntnap2*^{-/-} mice data, we used the higher trial to trial variability ($CV_{uncorr} = 0.44$) and the lower z-score threshold (z-score<-0.46) whereas to simulate the WT mice we used the lower variability ($CV_{uncorr} = 0.25$) and the higher z-score threshold (z_score<-0.42). The p-values were calculated using a t-test comparing the high variability and the low variability results (n=32 recording sessions, 6 WT mice data). **B.** Similar plot for the logistic model. **C.** The higher trial-to-trial variability also did not affect performance of the NNC. **D.** Performance of the linear SVM as a function of the coefficient of variation, which ranged between 0.25 (value measured in WT mice) to 2.95. The asterisks mark significant differences (p<0.05, t-test, n=32 recording

sessions) between the performance of the WT mice data (CV=0.25) with the simulations with higher coefficients of variation. SVM performance decayed significantly when the CV was 0.85 or higher, which is almost two-fold the CV measured in *Cntnap2*^{-/-} mice (CV=0.44). **D, E.** Similar plots for the logistic regression and the NNC. Logistic regression required a CV of 0.85 or higher for significant reduction in performance and NNC required a CV of 1.05 or higher to affect performance.

Minor comments:

The text still contains many incorrect and awkward expressions that need to be revised.

WE apologize for the reviewer. We have revised the text.

Please explain Figures 2E, 3C, 3E in more detail. What do different rows correspond to? What is perf?

Perf. is performance for the test and training set. For Figure 2E performance was determined as the fraction of test set and training set z-score responses for the best ROI that were correctly discriminated by the threshold determined from the training set. For Figure 3C, 3E, performance was calculated using the value output produced by the linear classifier (logistic regression for 3C and SVM for 3E) in response to the test set mixture. Correct responses were positive for go-mixtures and negative for no-go mixtures. We have changed the figure to make this more explicit.

We have expanded the figure captions for figures 2E, 3C, and 3E as follows:

Figure 2E. Example of the z-score responses of the best ROI, determined by the training set, to the 16 mixtures of the training set and the 16 mixtures of the test set. Each row represents the z-score of the best ROI for a given combination of target odor and contextual background odor from the training set (blue square) or the test set (red square). The blue vertical line represents the optimal threshold (z-score=-0.018) calculated from the training set. Z-score responses that exceed this threshold corresponded to no-go stimuli and responses below the threshold

corresponded to go-stimuli. The performance for the test set was determined by the fraction of test set responses that were correctly discriminated by this threshold.

Figure 3C-E. Output produced by multiplying the weights w_i of the logistic linear classifier and adding the constant bias term w_0 with the glomeruli activation. Each row represents the output produced for a given combination of target odor and contextual background odor from the training set (blue squares) or the test set (red squares). Correct performance consisted of positive responses for go stimuli and negative responses for no-go stimuli. **E.** Same as **C** but using the SVM weights.

More detailed explanation of Lasso analysis is warranted (Including explanation of Figures 9D, 9E).

We have added the following explanation to the main text:

Page 14, line 604: “The Lasso finds the combination of elements of a dictionary that could reconstruct the observed pattern of glomerular activation in the least square error sense. The reconstruction produced by the Lasso minimizes the sum of the absolute value (or L1 norm) contribution of the dictionary elements weighted by a regularization constant λ that is:

$$Cost = \sum_{i=1}^n (s_i - \sum_{j=1}^m c_j d_{i,j})^2 + \lambda \sum_{j=1}^m |c_j|$$

where n is the number of glomeruli, m is the number of elements in the dictionary, s_i is the observed activation of the i -th glomerulus, $d_{i,j}$ is the i -th glomerular activation of the j -th dictionary element, c_j is the concentration estimated and λ is the sparseness constrain. “

We have also changed the figure legends of Figure 9D and E:

D. Representation of 20 possible odors used to create odor mixtures. The yellow squares mark the odors selected for an example mixture consisting of a go target odor (isopropyl butyrate), a contextual background odor (isoamyl acetate), and a novel background odor (butyl propionate) that was not part of the dictionary used by the Lasso. **E.** Concentrations assigned by the Lasso for a dictionary size of 500 elements. The dictionary included 9 elements known to the animal: two go target odors, two no-go target odors, four contextual backgrounds, and (s)-(-)-limonene. It also included 491 randomly generated dictionary elements. Bars correspond to the estimated concentrations of the four contextual background odors, the four target odors, and (s)-(-)-limonene. Stems correspond to estimated concentrations for the randomly generated dictionary elements. The maximum of the estimated concentration was larger for the two target go odors compared to the two target no-go odors, so this would be considered a correct identification.

Panels in Figure 11 are too small to read.

We have reorganized and enlarged the panels to make them easier to read.

REVIEWER COMMENTS

Reviewer #3 (Remarks to the Author):

I have nothing to add.